# An Empirical Study into Clustering of Unseen Datasets with Self-Supervised Encoders

## Abstract

Can pretrained models generalize to new datasets without any retraining? We deploy pretrained image models on datasets they were not trained for, and investigate whether their embeddings form meaningful clusters. Our suite of benchmarking experiments use encoders pretrained solely on ImageNet-1k with either supervised or self-supervised training techniques, deployed on image datasets that were not seen during training, and clustered with conventional clustering algorithms. This evaluation provides new insights into the embeddings of self-supervised models, which prioritize different features to supervised models. We find evidence that supervised encoders offer more utility than SSL encoders within the training domain, and vice-versa far outside of it. However, fine-tuning SSL encoders for ImageNet-1k classification results in the opposite behaviour, with better performance than supervised-only models on in-domain and decreased performance on far out of domain data—worse at far-OOD than either SSL-only or supervised-only models. Clustering provides a way to evaluate the utility of self-supervised learnt representations orthogonal to existing feature quality estimation methods. Additionally, we find the silhouette score when measured in a UMAP-reduced space is highly correlated with clustering performance, and can therefore be used as a proxy for clustering performance on data with no ground truth labels.

## 1 Introduction

Self-supervised learning (SSL) has attracted great interest in recent years across almost every machine learning sub-field, due to the promise of being able to harness large quantities of unlabelled data and obtaining generic feature embeddings useful for a variety of downstream tasks (Balestriero et al., 2023). This has, for example, led to the development of impressive large language models (Brown et al., 2020) and computer vision systems trained on 1 billion images (Goyal et al., 2021). However, while the embeddings from an SSL-trained encoder can perform well on downstream tasks after fine-tuning the network, there has been less investigation into the utility of the embeddings without fine-tuning. Prior methods have primarily focused on metrics based on the individual representations, such as the use of discriminative features (Kalibhat et al., 2024), the effective rank of embeddings (Garrido et al., 2023), and the eigenspectrum decay of the feature covariance matrix (Agrawal et al., 2022). Prior work (Vaze et al., 2022; Zhou & Zhang, 2022) suggests SSL feature encoders generate embeddings suitable for clustering, but nonetheless adjust the feature encoders through fine-tuning. Yet, widespread interest in the application of large pretrained models on custom datasets, combined with prohibitive cost of compute, make this question important and increasingly urgent.

We find that to date there has been no investigation into whether SSL-trained feature encoders can serve as a foundation for clustering, yielding informative groupings of embeddings on real-world datasets that were totally unseen to the encoder during its training. Vaze et al. (2023) showed that features from SSL encoders are typically biased toward shape features and not colour, texture, or count when clustered using K-Means. However, this was conducted using a synthetic dataset, where very specific object attributes could be disentangled. Lu et al. (2023) proposed an SSL evaluation metric, CLID, where the combination of *Cluster Learnability* and Intrinsic Dimension of the SSL representations was determined to be a good estimator of 1-NN probing performance. However, this study was conducted with a focus on estimating probing performance on in-domain data, with little analysis into the clusterability of the SSL feature representation across target

domains. Cole et al. (2022) conducted a controlled study of the conditions under which contrastive pretraining succeeds, varying the quantity, domain, quality, and task granularity of the pretraining data. They showed that contrastive representations perform better when the downstream task is in-domain, and though they can capture coarse-grained semantic structure they lag substantially behind supervised baselines when the downstream task is more fine-grained. However, their analysis is restricted to a single SSL method and relies on supervised methods (linear probing and fine-tuning) to read out how much information is captured in the representations. Our work is complementary to this; we hold the pretraining data fixed and use label-free clustering to investigate how well SSL models represent unseen data.

In contrast, in this work we perform a *zero-shot transfer-learning task*, evaluating the performance of a suite of SSL-trained feature encoders across a diverse set of datasets, using various classical clustering methods, yielding the following contributions. We:

- Conduct the first (to our knowledge) in-depth investigation of clustering of SSL feature encoders outside their training domain, finding SSL encoders can produce meaningful clusters across a variety of unseen datasets without per-dataset parameter tuning. These results are complimentary to established results on SSL classification performance (Cole et al., 2022; Lu et al., 2023), which focus on other evaluation methods, within-domain classification, and/or narrower ranges of SSL models.

- Establish a comprehensive suite of benchmark evaluations for clustering unseen image datasets using models pretrained exclusively on ImageNet-1k.

- Show that clusterings can be further investigated on multi-labelled datasets to identify which stimulus attributes the encoder prioritizes.

- Demonstrate that fine-tuning SSL models for classification tasks on the same domain as their pretraining results in better performance on in-domain data, but reduces clustering performance on far-OOD data.

- Discover that the representations of SSL-pretrained models are more heavily impacted by background-foreground disparity than supervised pretrained models.

- Corroborate, through label-free clustering rather than supervised probing, the finding of Cole et al. (2022) that SSL encoders degrade monotonically with finer label granularity, and extend this finding to additional SSL paradigms and architectures.

- Find manifold-based reduction of embeddings is essential for performant clustering.

- Find that Agglomerative Clustering clusters embeddings best, statistically significant though the effect size is small.

- Find that the silhouette score is strongly correlated with the adjusted mutual information score, especially when the silhouette is measured in UMAP-reduced space, and hence can be a strong proxy of clustering performance without access to ground-truth labels.

## 2  Background

Our work builds upon two broad fields of research: self-supervised learning for computer vision applications, and clustering. We give a general overview of each field.

**Self-Supervised Learning** (SSL) has recently received an increasing amount of interest from the computer vision domain, in part due to its promising results in natural language processing (Brown et al., 2020). Whilst SSL has a long history of research, currently dominant methods can be divided into four general categories (Balestriero et al., 2023): (1) Contrastive Learning approaches, which build on metric learning, in which embeddings of multiple views of the same instance are brought together and embeddings from different instances are pushed apart (Chopra et al., 2005; Song et al., 2016; Sohn, 2016; Chen et al., 2020; He et al., 2020; Chen et al., 2021); (2) Self-Distillation approaches, where a student and teacher encoder process an input image with distinct transforms applied, and the student is tasked with predicting embeddings of the teacher (Grill et al., 2020; Chen & He, 2021; Caron et al., 2021; Zhou et al., 2022; Oquab et al., 2024); (3) Canonical Correlation Analysis approaches, where feature embeddings are analyzed in terms of

the cross-covariance matrix, through mechanisms such as minimizing covariance across feature dimensions and minimizing correlation across feature embeddings for different inputs (Caron et al., 2020; Zbontar et al., 2021; Ermolov et al., 2021; Bardes et al., 2022); (4) Masked Image Modelling approaches, where large parts of the input image are masked out and have to be reconstructed in image-space (Pathak et al., 2016; He et al., 2022; Bao et al., 2022; Xie et al., 2022).

Delimitation: In this paper we explicitly focus on SSL backbones pretrained on the ImageNet-1K dataset, in order to ensure that the pretrained backbones can be accurately compared on domain shifted data. This limits the range of models available for us to explore. State-of-the-art models are typically trained on larger, non-public datasets such as LVD-142M which was constructed to contain a much broader range of data whose domain overlaps with all commonly used evaluation datasets (Oquab et al., 2024). This makes it infeasible to evaluate the model on domains unseen during training; consequently we do not include the DINOv2 backbone by Oquab et al. (2024).

**Clustering** is one of the most common tasks in a large variety of applications and can be defined as the task of finding local structures that are homogeneous and separated without explicit label supervision (Everitt et al., 2011). This problem has been studied for centuries resulting in methods using clustering criteria based on partitioning (Lloyd, 1982; Arthur & Vassilvitskii, 2007), fuzzy theory (Bezdek et al., 1984), graph theory (Yu & Shi, 2003; Frey & Dueck, 2007), density (Ester et al., 1996; Ankerst et al., 1999; McInnes & Healy, 2017), hierarchies (Sokal & Michener, 1958; Ward, 1963), and many more (Xu & Tian, 2015). These methods have traditionally necessitated a disjointed processing pipeline, as the clustering algorithms have been optimized independently of the feature generators. However, in recent years several methods have been proposed to jointly learn feature extractors and clustering processes (Ronen et al., 2022; Caron et al., 2018; Tapaswi et al., 2019; Pakman et al., 2020; Yang et al., 2017; Van Gansbeke et al., 2020; Millán Arias et al., 2022; Adaloglou et al., 2023). We focus this paper on classical lightweight clustering methods, which can be easily attached to any backbone. This means that recent advances within deep clustering methods such as SCAN (Van Gansbeke et al., 2020) and TEMI (Adaloglou et al., 2023) are not included. We leave it open as future work to conduct a similar study into learned clustering methods.

## 3 Experimental Design

We consider the task of **zero-shot clustering** of feature embeddings obtained from pretrained encoders. The aim of this task is to cluster the feature embeddings from various as-yet unseen datasets, in a way such that the clusters are intrinsically well-defined and, ideally, match the ground-truth (GT) label assignments if available, through the transfer of pretraining knowledge and without any domain-adaptation. Our feature encoders and clustering methods are only tuned on data from a single dataset, the commonly used ImageNet-1k (IN-1k) (Russakovsky et al., 2015). The clustering methods are then deployed on all test datasets without re-tuning any of the parameters, allowing us to cluster novel datasets without utilizing any training data for the transfer datasets.

### 3.1 Feature Encoders

In order to capture the diverse methodologies within the self-supervised learning field, we compare methods from the major self-supervised paradigms within computer vision (Balestriero et al., 2023). We choose one representative method per paradigm, and compare the clusterability of their features against those of a model pretrained with cross-entropy supervision (X-Ent.) using the IN-1k labels. The SSL models selected are as follows:

- **Contrastive Learning**: MoCo-v3 (Chen et al., 2021)

- **Self-Distillation**: DINO (Caron et al., 2021)

- **Canonical Correlation Analysis**: VICReg (Bardes et al., 2022)

- **Masked Image Modelling**: MAE (He et al., 2022)

Table 1: **Dataset overview.** We evaluate on a diverse set of experiments of differing levels of task granularity, number of classes and samples, domain shift, and class imbalance. We report the number of samples and GT classes contained in the subset of the dataset that was clustered; where possible this was the publicly available test partition (see Appendix H for more details). The class imbalance, $\rho$, is the ratio between the most and least frequent classes.

| Type | Dataset | Reference | # Sample | # Class | $\rho$ | Description |
|---|---|---|---|---|---|---|
| In-Domain | ImageNet-1k | Russakovsky et al. (2015) | 50 000 | 1 000 | 1.00 | Diverse general objects |
| | ImageNet-v2 | Recht et al. (2019) | 10 000 | 1 000 | 1.00 | Diverse general objects |
| | CIFAR-10 | Krizhevsky (2009) | 10 000 | 10 | 1.00 | Diverse general objects |
| | CIFAR-100 | Krizhevsky (2009) | 10 000 | 100 | 1.00 | Diverse general objects |
| | ImageNet-9 originals | Xiao et al. (2020) | 4 050 | 9 | 1.00 | Diverse general objects |
| Domain-shift | ImageNet-9 FG-only | Xiao et al. (2020) | 4 050 | 9 | 1.00 | Isolated foregrounds |
| | ImageNet-9 MixRand | Xiao et al. (2020) | 4 050 | 9 | 1.00 | Remixed fore/background |
| | ImageNet-R | Hendrycks et al. (2021a) | 30 000 | 200 | 8.43 | Art/sculptures of objects |
| | ImageNet-Sketch | Wang et al. (2019) | 50 889 | 1 000 | 1.02 | Sketches of objects |
| Near-OOD | ImageNet-O | Hendrycks et al. (2021b) | 2 000 | 200 | 6.00 | Diverse general objects |
| | LSUN | Yu et al. (2015) | 10 000 | 10 | 1.00 | Urban/indoor scenes |
| | Places365 | Zhou et al. (2018) | 36 500 | 365 | 1.00 | Scenes |
| Fine-grained | FGVC Aircraft | Maji et al. (2013) | 3 333 | 100 | 1.03 | Aircraft variants |
| | Stanford Cars | Krause et al. (2013) | 8 041 | 196 | 2.83 | Car variants |
| | Oxford Flowers | Nilsback & Zisserman (2008) | 6 149 | 102 | 11.90 | Flower variants |
| | NABirds | Van Horn et al. (2015) | 24 633 | 555 | 6.67 | Bird species |
| | BIOSCAN-1M | Gharaee et al. (2023) | 24 799 | 2 688 | 782.50 | Insect species |
| | iNaturalist-2021 | Van Horn et al. (2021) | 100 000 | 10 000 | 1.00 | Plant & animal species |
| Far-OOD | CelebA | Liu et al. (2015) | 19 962 | 1 000 | 32.00 | Human faces (identity) |
| | UTKFace | Zhang et al. (2017) | 5 925 | 101 | 549.00 | Human faces (age) |
| | BreakHis | Spanhol et al. (2016) | 3 164 | 32 | 8.60 | Tumor tissue microscopy |
| | DTD | Cimpoi et al. (2014) | 1 880 | 47 | 1.00 | Texture descriptions |
| | EuroSAT | Helber et al. (2019) | 4 050 | 10 | 1.50 | Satellite RGB images |
| | MNIST | LeCun et al. (1998) | 10 000 | 10 | 1.27 | Handwritten digits |
| | Fashion MNIST | Xiao et al. (2017) | 10 000 | 10 | 1.00 | Clothing articles |
| | SVHN | Netzer et al. (2011) | 26 032 | 10 | 3.20 | House numbers |

For each method we consider two common backbone architectures, ResNet-50 (He et al., 2016) and ViT-B (Dosovitskiy et al., 2021), using publicly available checkpoints trained on the IN-1k dataset. However, (1) MAE only supports transformer architectures and hence lacks a ResNet-50 checkpoint; (2) VICReg did not have a pretrained ViT-B checkpoint available. We also investigated using embeddings from randomized ResNet-50 and ViT-B networks, or using the raw image pixels, but across all datasets the performance of these was negligible and did not serve as a worthwhile baseline comparator (see Appendix H).

## 3.2 Clustering Methods

In order to cluster the feature embeddings, we considered several classical clustering methods: *K-Means* (Lloyd, 1982) with K-Means++ init. (Arthur & Vassilvitskii, 2007), *Spectral Clustering* (Yu & Shi, 2003), *Agglomerative Clustering* (AC) (Everitt et al., 2011), *Affinity Propagation* (AP) (Frey & Dueck, 2007), and *HDBSCAN* (McInnes & Healy, 2017). These clustering methods were chosen because they have few parameters to tune, cover several clustering paradigms (partition, hierarchical, graph-theory, and density), and include both parametric and non-parametric methods. As K-Means and Spectral require the number of clusters in order to run, we assume that this is known *a priori*. In contrast, AC, AP, and HDBSCAN automatically determine the number of clusters in the data. AC can either operate with the number of clusters given or inferred, and we consider both configurations ("AC w/ C" and "AC w/o C", respectively). HDBSCAN can identify samples which belong to *no* cluster (noise/background samples). Unless stated otherwise, we consider the noise class to be its own class when computing the AMI (see Equation 2). This sets HDBSCAN at a disadvantage, since the samples it identifies as noise are typically distributed across all

GT classes, but is fairer than ignoring samples it identifies as noise since that would evaluate it only on easier samples.

We excluded neural clustering methods, such as Neural Clustering Processes (Pakman et al., 2020) or DeepCluster (Caron et al., 2018), as they jointly learn the feature encoder and clustering step, which is outside our scope. In this work, we focus on evaluating the clusterings of feature embeddings from pretrained self-supervised encoders.

### 3.3   Datasets

We evaluated the different permutations of feature encoders and clustering methods on a diverse set of datasets, detailed in Table 1. These datasets span tasks with differing levels of label granularity, number of classes and samples, domain shifts, and degree of class imbalance. Out of all these datasets, only the IN-1k training split was present during training of the feature encoders and used to optimize the parameters of the clustering methods. No other datasets have been observed by the networks, and the methodology was not tuned on them. We divided the datasets into five groups as follows:

- **In-domain (ID).** Images and class labels lie within the IN-1k domain.
- **Domain-shifted (DS).** Class labels are aligned with IN-1k, but the images are changed *e.g.* background removed or replaced; images of artwork representing the class.
- **Near-out-of-domain (Near-OOD).** Images look like IN-1k images, and the classification task is similar but with new classes and distributional shift.
- **Fine-grained near-out-of-domain (FG).** Natural images resembling a subdomain of IN-1k, but labelled at a much finer-level of granularity *e.g.* plant species.
- **Far-out-of-domain (Far-OOD).** Images which lie outside the domain of IN-1k, with especially different objectives *e.g.* textures, text, faces, microscopy slides.

### 3.4   Evaluation Metrics

We evaluated the performance of a clustering using two metrics: adjusted mutual information (AMI) (Vinh et al., 2010) and silhouette score (Rousseeuw, 1987), defined in Appendix C. AMI measures the agreement between the constructed clusters and the GT labels whilst correcting for chance-level agreement. The silhouette score measures how well-defined the clusters are intrinsically, without reference to a GT clustering. AMI was chosen over the commonly used Normalized Mutual Information (NMI) metric (Zhou et al., 2024), as it corrects for chance agreements in clusterings. We use AMI instead of adjusted Rand index as AMI works better in the regime of unbalanced GT clusters (Romano et al., 2016), common in real-world data scenarios and true of half our evaluation datasets, but our findings would be unchanged otherwise (Table 8).

### 3.5   Clustering Parameter Search

In order to maximize performance of each permutation of the feature encoder and clustering methods, we conducted a staggered sweep over relevant clustering parameters. This was conducted using subsets of training splits of IN-1k, Imagenette, and Imagewoof (Howard, 2019). Imagenette and Imagewoof are coarse- and fine-grained subsets of IN-1k, resp., with 10 classes each. These datasets were selected to find parameters robust against changing the number of classes and their granularity, whilst *only* optimizing clustering performance on data within the encoder's original training set. For each of these three, we created a validation set as a class-stratified random subset of the training set with the same number of samples as in the datasets' test set (50 000, 3 925, and 3 929 resp.). The same split was used across all encoders, clusterers, and stages of the parameter search.

As the curse of dimensionality can negatively affect performance of the considered clustering methods (Bellman et al., 1957), we searched for an appropriate dim. reduction process to apply before clustering. We considered using PCA (controlled either by number of reduced dimensions, or fraction of variance explained) (Pearson, 1901), UMAP (McInnes et al., 2018), and PaCMAP (Wang et al., 2021), and compared the performance to

using the original (unreduced) embeddings. We found that raw images and embeddings through randomized (untrained) networks were typically best clustered when reduced with PCA. Embeddings with pretrained networks were typically best clustered with some form of manifold-based reduction. For Spectral clustering, a manifold-based reduction step is already included in its method and it benefitted from seeing the original or PCA-reduced embeddings for this process. For others, clustering was best with UMAP-reduction, and the number of reduced dimensions was unimportant across the range 5–200 dims. The findings are consistent with the idea that neural networks embed stimuli onto a low-dimensional, non-linear manifold within their embedding space. For further details on the parameter search and its outcomes, see Appendix E.

### 3.6 Experimental Methodology

For each test dataset (see Table 1), we preprocessed the images by resizing the shortest side to 224 pixels and taking a centered square 224×224 crop. Greyscale images were converted to RGB by replicating the grey channel. For each encoder, the image was standardized using the RGB mean and standard deviation used to train the encoder, then passed through the encoder to create an embedding. The embeddings are 2048-d (ResNet-50) or 768-d (ViT-B). For each encoder, we clustered the generated embeddings with each clusterer, using parameters fit on IN-1k training data (see §3.5). When using UMAP or PCA for dim reduction, this was fit separately for each test dataset.

## 4 Experimental Results

We report the clustering capabilities of the considered encoders and clusterers measured by AMI, with ResNet-50 and ViT-B backbones on datasets of varying distances from the training domain.

### 4.1 Comparison of Clustering Methods

We compared the performance of the clustering methods by ranking each clusterer for each combination of pretrained encoder and dataset, shown in Figure 7. The results show that AC w/ C performs best ($p < 0.05$; Wilcoxon signed-rank test versus each other clusterer). Spectral, K-Means, AC w/o C, and AP all perform similarly. HDBSCAN performed worst ($p < 10^{-33}$), due to its use of a noise class instead of trying to place every sample in a cluster. Although this is a legitimate and principled methodology (McInnes, 2016), it puts HDBSCAN at a disadvantage here; we found HDBSCAN often placed half the samples in the noise class (see Table 16). When considering non-parametric clusterers, AC w/o C and AP were in a statistical tie. The trends across encoders and datasets were similar, irrespective of the clusterer used (see Appendix L). For subsequent analysis, we thus present the average over clusterers.

### 4.2 Comparison of SSL Encoders

For each dataset described in §3.3, we measured the AMI between the clustered embeddings of each pretrained encoder and the GT labels (averaged over clusterers). Using the IN-1k supervised encoder as a baseline, we took the difference between its AMI and that of the SSL encoders, then took the average within each group of datasets.

As shown in Figure 1, the performance of the SSL encoders is lower than that of the supervised network on in-domain, domain-shifted, and near-OOD datasets; though the effect size is often large (in the order of 10 p.p.), the difference is generally not significant for individual models due to the limited number of datasets in each category and the variance between them (see starred bars on Figure 1 for significant effects for individual models). The MAE encoder (either using the embedding of the CLS token, or the average embedding of the image patch tokens) performed especially poorly (significantly worse than supervised ViT-B on DS and Near-OOD; $p < 0.05$). This finding is congruent with the observation that MAE-trained models possess details about the pixel-level contents of the stimulus, but need fine-tuning to perform well at whole-image classification (He et al., 2022). For the SSL models overall (excluding MAE from the group due to its known limitation of requiring fine-tuning), the reduction is significant for in-domain (paired $t$-test over 5 datasets for 5 models: $n = 25$, $p < 10^{-4}$) averaging $-9.8$ p.p.; domain-shift ($-7.5$ p.p., $n = 20$, $p < 10^{-3}$); and near-OOD ($-10.7$ p.p., $n = 15$, $p < 0.01$).

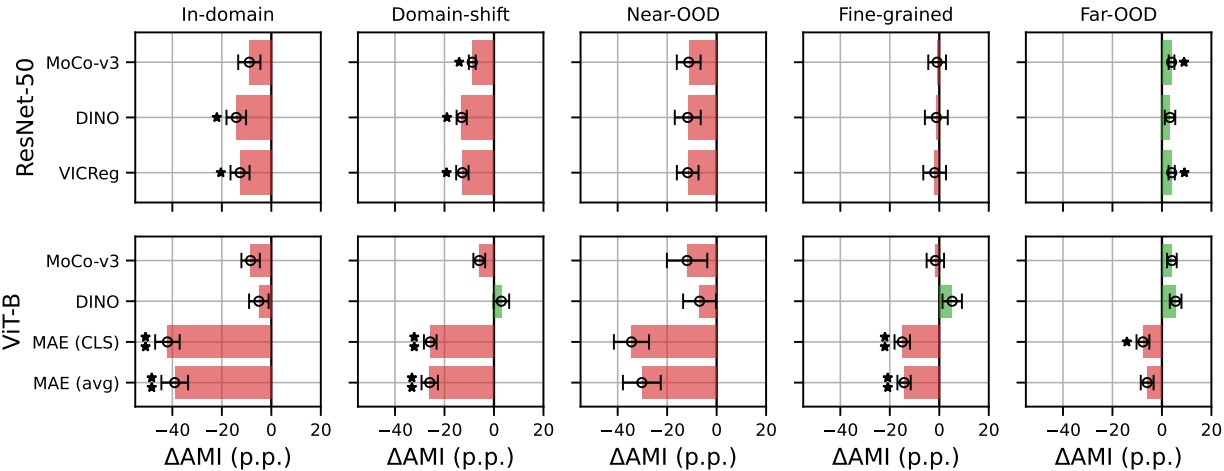

Figure 1: **Percentage-point (p.p.) difference in AMI between clusters formed from SSL encoder embeddings versus supervised encoder embeddings.** We compare the quality of clustering of each dataset (mean AMI over 6 clusterers) using SSL encoder embeddings against that of encoders trained with cross-entropy on IN-1k. We present the mean across datasets in each group (error bars: $\pm 1$ stderr; $3 \leq N \leq 8$ datasets; $\star$: $p < 0.05$; $\star\star$: $p < 0.01$).

For FG datasets, the overall results show SSL encoders are not significantly different in performance compared to supervised (except MAE, with perf. lower than sup., $p < 0.05$, test as above), but when we explore the results on a per-dataset basis, we find supervised encoders perform best on Stanford Cars and NABirds by a reasonable margin, whilst SSL encoders perform best on Aircraft and Oxford Flowers datasets (see Appendix H for details). We speculate this difference between the FG datasets may be caused by their respective (dis)similarity with IN-1k imagery. When we consider Far-OOD datasets, we find SSL-encoders outperform supervised networks (except MAE, which is not well-aligned with whole-image classification), with a significant overall increase averaging +4.1 p.p. ($n = 40$, $p < 10^{-4}$).

Taken together, these results demonstrate that supervised encoders perform better at clustering unseen datasets similar to the training data, but as the data moves further from the training dataset, performance of supervised networks decreases and SSL encoders increases such that they become better. Comparing within the SSL encoders, DINO produced the best SSL encoder when using a ViT-B architecture, but was the worst SSL encoder for ResNet-50. We believe this is because the DINO training process, unlike other SSL methods, is able to take advantage of the ViT's attention mechanism to focus solely on the subject, which we explore further in §4.5.

### 4.3 Effect of Dataset Granularity

Furthermore, we observe that the overall level of performance on FG datasets varies greatly. While seemingly arbitrary, we find that performance correlates with how fine-grained the datasets are when considering the proposed granularity measure from Cui et al. (2019). Specifically, we find that FGVC Aircraft is the most challenging dataset, matching the finding by Cui et al. (2019) that it is the most fine-grained dataset of the ones considered, while NABirds and Oxford Flowers gradually become more coarse-grained, and easier to correctly cluster. Similarly, we find that the large scale iNaturalist-21 dataset is in general a very hard dataset. These observations echo the recent results from Cole et al. (2022), where it was determined that current SSL methods are not suitable for fine-grained tasks. Using the iNaturalist-21 and BIOSCAN-1M datasets we can vary the labels from coarse to fine-grained using the 7 taxonomic levels available for each data point, see Figure 2. For these results we use an alternative AMI measure, $\mathrm{AMI_{true}}$, which reports the proportion of entropy in the true labels which is explained by the clustering (see Equation 3). On the iNaturalist-21 dataset, we find that for all methods the $\mathrm{AMI_{true}}$ score is high at for coarse-level ranks (kingdom, phylum, and class), while it dramatically decreases for finer-grained ranks, with a max performance of 10% $\mathrm{AMI_{true}}$

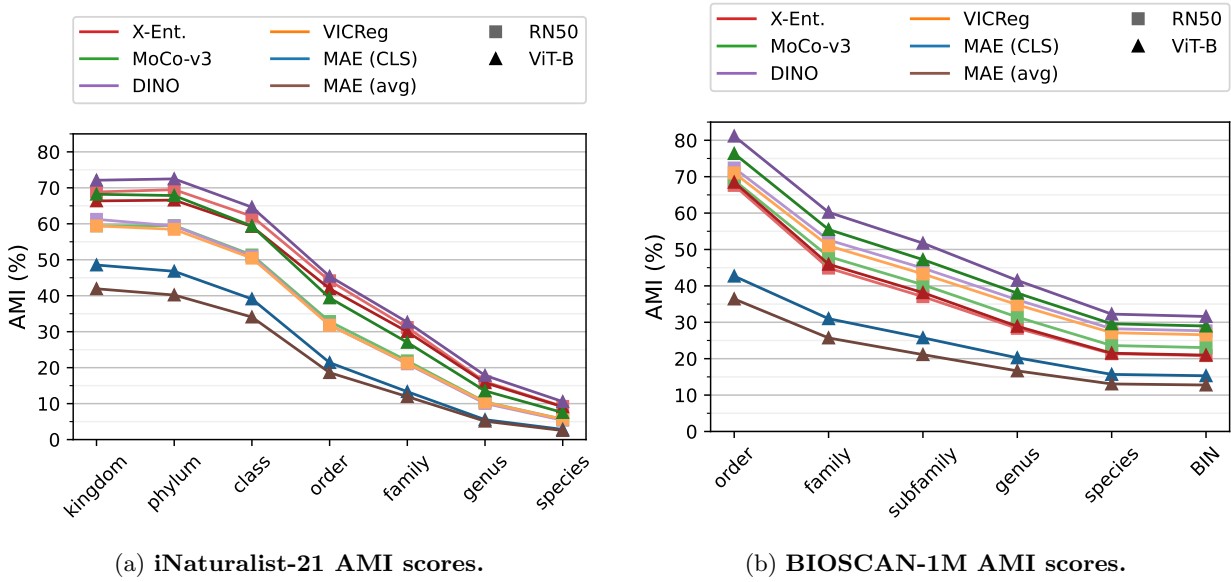

(a) **iNaturalist-21 AMI scores.**  (b) **BIOSCAN-1M AMI scores.**

Figure 2: **AMI$_{\text{true}}$ scores across taxonomic levels.** We measure the AMI$_{\text{true}}$ score at each of the 7 taxonomic levels of the iNaturalist-21 dataset and from order to species level as well as when using the Barcode Index Number (BIN) as a proxy for subspecies labels for the BIOSCAN-1M dataset. The scores are reported for each encoder, averaged over the tested clustering methods. Paler lines with squares: ResNet-50; darker lines with triangles: ViT-B.

at a species level. Similarly, we find that there is a peak at the order level when using the BIOSCAN-1M dataset, and suffers a less drastic performance drop when using species and BIN labels. This aligns with the findings of Cole et al. (2022), who found that the accuracy of SSL encoders decreases monotonically as one moves down the label hierarchy when evaluating with a linear probe.

## 4.4 Comparison of Fine-Tuned SSL Encoders

There is often more than one way in which a collection of images can be legitimately grouped together, depending on which high-level properties within the images are prioritized. Thus, although machine learning datasets are typically only annotated once with one set of GT labels, other valid groupings may exist. We considered that clustering the embeddings produced by the SSL-pretrained encoders may sometimes result in "legitimate" clusterings that are consistent with particular semantic features of the images, just not aligned with categorization used for the GT annotations. For example, we qualitatively found SVHN clusters corresponded more to the colour and font of the digits than the identity of the center digit (the classification target). Moreover, previous work has shown that MAE requires fine-tuning (FT) to be able to be able to perform whole-frame classification (He et al., 2022). Consequently, we investigated whether fine-tuning the pretrained encoders on an IN-1k classification task would make their embeddings more aligned with the classification typically employed in machine learning tasks. We fine-tuned each of the SSL-pretrained encoders on IN-1k following the methodology of He et al. (2022), repeated the clustering parameter search for the FT encoders, then clustered their embeddings of each test dataset.

As shown in Figure 3, we found fine-tuning unsurprisingly increases performance on in-domain and domain-shifted datasets, where target classes are the same as (or subset of) the IN-1k classes used for the FT task. The gain in performance was sufficient that SSL-encoders tended to beat the supervised network on these datasets. Overall the 6 models beat the supervised-only baseline on in-domain data by an average of $+1.2$ p.p., which was significant (paired $t$-test over 5 datasets for 6 models: $n=30$, $p<10^{-3}$). Furthermore, with a ResNet-50 backbone we find weak evidence that FT SSL-encoders beat the supervised baseline on Near-OOD data $(+1.9$ p.p., $n=9$, n.s.), whilst with a ViT-B backbone the FT SSL-encoders beat the supervised baseline significantly on FG datasets $(+2.4$ p.p., $n=18$, $p<0.01)$.

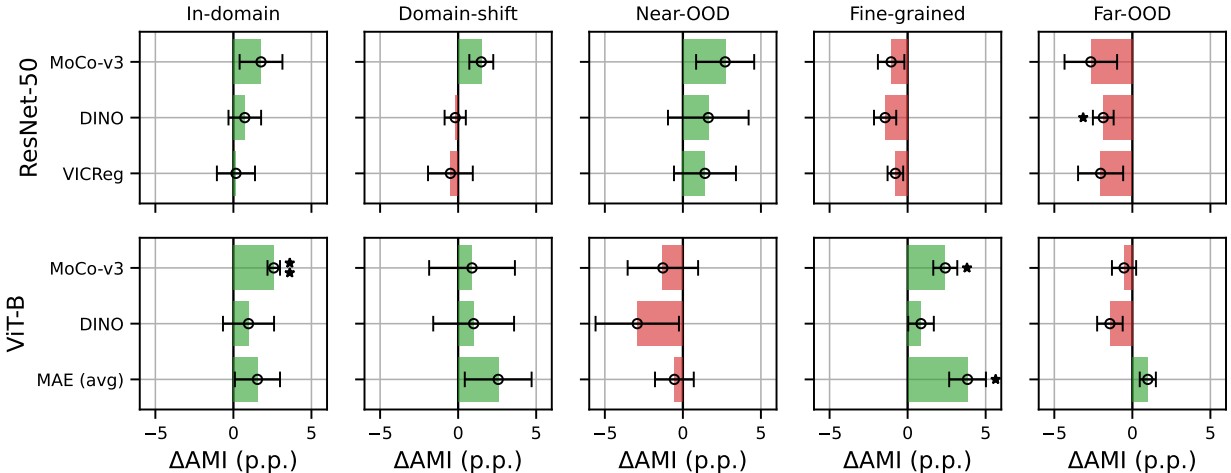

Figure 3: **Percentage-point (p.p.) difference in AMI between clusters formed from embeddings of SSL-pretrained networks fine-tuned on IN-1k versus fully-supervised networks.** We measure the difference in AMI (mean over 6 clusterers) with fine-tuned SSL encoders as compared to encoders trained with cross-entropy on IN-1k (error bars: $\pm 1$ stderr; $3 \leq N \leq 8$ datasets; $\star$: $p < 0.05$; $\star\star$: $p < 0.01$). *Note:* The x-scale differs from that used in Figure 1, but the baseline (0 values) are the same.

Excluding MAE as an outlier which requires fine-tuning to be performant on any task, we found that fine-tuning the SSL models significantly increased their in-domain clustering ($n = 25$; $p < 10^{-5}$) by an average of +11.1 p.p. For domain-shifted datasets (+8.1 p.p., $n = 20$; $p < 0.01$), and near-OOD (+11.0 p.p., $n = 15$; $p < 0.01$), the increase was also significant. There was no significant change for fine-grained data (+0.06 p.p., $n = 30$; n.s.). These differences are visualized in Appendix K.

However, the performance on Far-OOD datasets declined significantly post-FT (avg. exc. MAE: $-5.8$ p.p., $n = 40$, $p < 10^{-5}$), enough that the performance of SSL-encoders became worse than supervised encoders (avg inc. MAE: $-1.3$ p.p., $n = 48$, $p < 0.02$). The only exception to this was MAE, which greatly increased its performance on all types of dataset. Across the supervised and FT encoders, MAE was the best performing encoder on every group of datasets, though its performance of Far-OOD data was still below that of the non-FT SSL-encoders.

### 4.5 ImageNet-9 Background Challenge

To investigate whether SSL encoders natively focus more on foreground or background contents of images, we analyzed the amount of information about ImageNet-9 variants (Xiao et al., 2020), tabulated in Table 2. We present the AMI when clustering the original images (OG), foreground only (FG), foreground replaced with black ($FG^C$), background only (bounding box replaced with bg texture; BG), mixed-same (fg overlaid on the bg of a sample of the same class; MS), and mixed-random (fg overlaid on the bg of a random sample; MR). Illustrative examples of these are shown in Figure 9. We also show the difference between MS and MR performance ("BG-Gap"; Xiao et al., 2020) and the difference relative to MS ("BG-Gap Rel").

SSL and supervised encoders yielded similar quality to each other when clustering the original images (OG), or when clustering the foreground-only images (FG). Supervised and fine-tuned networks consistently had more information about the background of the images ($FG^C$ and BG), congruent with the widely held belief that supervised networks learn to exploit information in the background of images. Surprisingly then, we find SSL-encoders have nearly twice as large a BG-gap than their supervised counterparts. Despite the fact that SSL embeddings possess less information about the image backgrounds, using a background that is incongruent with the foreground induces much more "confusion" in the SSL-encoders. We hypothesize that this is because supervised networks are better able to prioritize foreground over background information when creating their embeddings, whereas SSL-encoders are typically unable to distinguish foreground from background and thus their embeddings are always a combination of the two. This is in keeping with their

Table 2: **ImageNet-9 breakdown.** We show the AMI (%) when clustering variants of the ImageNet-9 dataset, averaged over 6 clusterers. See §4.5 for descriptions of the variants. **Bold**: highest scoring encoder per dataset. Underlined: highest scoring encoder per backbone. Background: ranges from the median value (white) to maximum (blue/red) per dataset. FT: fine-tuned with cross-entropy (x-ent.) on IN-1k.

| Encoder | FT | OG | FG | FG$^C$ | BG | MS | MR | BG-Gap | BG-Gap Rel |
|---|---|---|---|---|---|---|---|---|---|
| **RN50** — X-Ent. | | 69 | **70** | 47 | 26 | 71 | 60 | 11 | 14 |
| MoCo-v3 | | 70 | 61 | 35 | 17 | 60 | 48 | 12 | 19 |
| DINO | | 70 | 64 | 32 | 22 | 59 | 43 | 16 | 26 |
| VICReg | | 69 | 63 | 30 | 19 | 58 | 40 | 18 | 30 |
| MoCo-v3 | ✓ | **77** | **70** | 48 | 25 | **76** | **64** | 12 | 15 |
| DINO | ✓ | 75 | 68 | 49 | 25 | **76** | 62 | 14 | 18 |
| VICReg | ✓ | 75 | 67 | 47 | 24 | **76** | **64** | 12 | 14 |
| **ViT-B** — X-Ent. | | 61 | 61 | 52 | 27 | 66 | 51 | 15 | 22 |
| MoCo-v3 | | 62 | 62 | 41 | 23 | 65 | 44 | **21** | 31 |
| DINO | | 72 | 68 | 43 | 25 | 73 | 61 | 11 | 15 |
| MAE (CLS) | | 38 | 39 | 21 | 10 | 29 | 18 | 11 | 41 |
| MAE (avg) | | 44 | 41 | 22 | 9 | 25 | 15 | 10 | **46** |
| MoCo-v3 | ✓ | 64 | 53 | 52 | 27 | 65 | 52 | 14 | 19 |
| DINO | ✓ | 65 | 53 | **55** | **28** | 71 | 53 | 18 | 24 |
| MAE (avg) | ✓ | 66 | 57 | 52 | 25 | 69 | 54 | 16 | 21 |

training task, which is to give every unique stimulus its own unique embedding (instance learning), and the stimulus is comprised of both its foreground *and* its background.

The only exception to this pattern was the DINO ViT-B encoder, which had the lowest BG-gap of all ViT-B encoders, with the exception of MAE. MAE's BG-Gap is lower only because it performs so poorly on the IN9-MS task to begin with and it still has a large relative reduction. We speculate that DINO has such a low BG-gap because it learnt to attend to foreground objects as an emergent outcome of its training process (Caron et al., 2021). This is possible with the ViT backbone, but not the ResNet as it lacks attention mechanisms and must attend to the whole stimulus, hence the DINO ResNet-50 encoder performs the same as MoCo-v3 and VICReg. The behaviour is not replicated for MoCo-v3 ViT-B since its training loss incentivizes it to attend to all features that are common across multiple views of the same sample and differ between samples, including background features.

We provide similar breakdowns for the information encoded by clusterings about different label types for ImageNet-Rendition (Appendix O) and BreakHis (Appendix P).

### 4.6 Correlation between AMI and Silhouette Score

So far, we focused on the AMI metric, which measures clustering quality by comparing the predicted clusters with the GT labels. However, in the context of SSL this can be problematic as there may be no GT available. Therefore, we considered whether the intrinsic silhouette metric (see §C.2), $S$, calculated from just the predicted clusters would be valuable for evaluation of SSL encoders.

We compared the AMI and $S$ for each clusterer across encoders and datasets (see Figure 4 where we compare the rank of the values, and Figure 11 where the raw values are compared) and computed the Spearman's rank correlation coefficient, $\rho$, using the silhouette score in either the original embedding space, or the UMAP-reduced space. We find that AMI and $S$ are strongly correlated, with high silhouette scores having correspondingly high AMI scores. The correlation is increased in UMAP-reduced space, where a wider range of $S$ values are observed. Our findings are contrary to previous work by Xu et al. (2022) which found $S$ to be an inconsistent metric. Nonetheless, we conclude silhouette score can be a good proxy for cluster quality when GT labels are unavailable.

## 5 Practical Guidance

Throughout this paper we have so far focused on analyzing and elucidating the performance of various SSL backbones and clustering methods combinations. In this section, we offer recommendations on how to apply zero-shot clustering (ZSC) on new datasets and domains based on our findings.

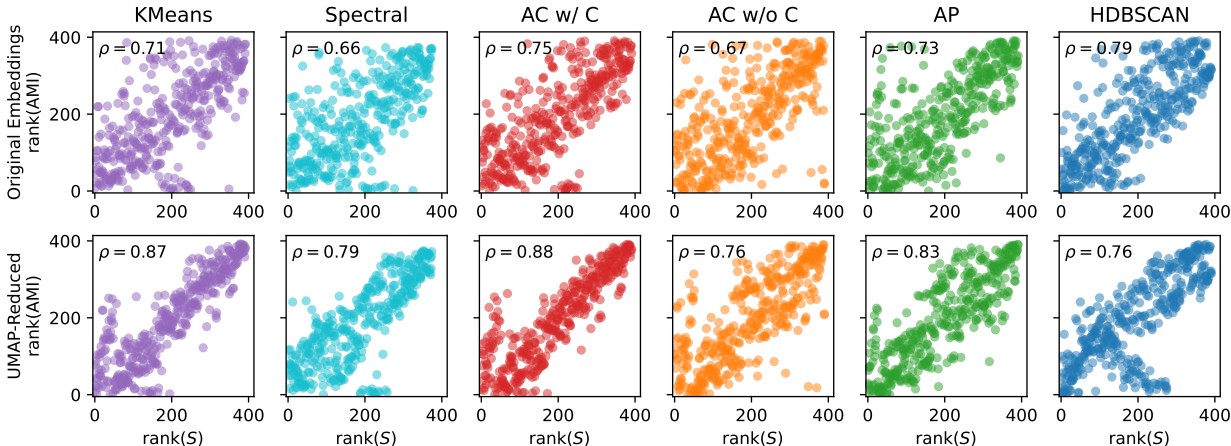

Figure 4: **Ranked AMI–Silhouette scatter plots.** The ranked AMI and silhouette score ($S$) per clusterer, across datasets and encoders (higher is better). The silhouette scores are measured in the original (top) and UMAP-reduced 50-d (bottom) feature spaces. We indicate the per-clustering-method Spearman's rank correlation ($\rho$).

## 5.1 Dimensionality Reduction

Through our hyperparameter search (Appendix E) it was clear that UMAP (with 30 neighbours and distance threshold of 0.0) is the best choice for dimensionality reduction, yielding the best performance across all trained models[1]. In our hyperparameter search, we found that 5–200 dimensions worked well, and recommend using a 50-dimensional UMAP space as the default setting in novel applications of ZSC. For large datasets, practitioners may wish to reduce the number of dimensions further to save on memory consumption—we recommend using at least 5 dimensions in this case.

## 5.2 Choice of Clustering Method

In §4.1 we found that Agglomerative Clustering (AC) with known number of clusters performed significantly better than all other clustering approaches. Using AC when not knowing the number of clusters *a prior* was found to perform similarly to the other clustering methods. Therefore, we directly recommend that all novel applications of ZSC start out by using AC. Based on our hyperparameter search (see Appendix E) we found the L2 metric with either Ward (best for supervised) or Average (best for SSL) linkage to be strong default hyperparameters. When the number of clusters is unknown, we recommend selecting the distance threshold which maximizes the silhouette score in UMAP-reduced space (as originally proposed by Rousseeuw, 1987), with an initial distance threshold of 2.0.

For larger datasets, AC can be infeasible due to its $\mathcal{O}(n^2)$ memory scaling, in which case K-Means, $\mathcal{O}(nd)$, should be used. If this is still OOM, we recommend fitting the centroids on a subset of the data.

## 5.3 Applicability of Silhouette Scores

In §4.6 it was determined that the silhouette score of a clustering is highly correlated with the AMI score, especially when the silhouette score is computed in UMAP-reduced space. Since silhouette score is a metric which can be evaluated without ground truth labels, this means it can be a valuable proxy for how well a model can represent and cluster unlabelled data.

We envision that this can be an important tool during the construction of datasets, as it can give an indication of model performance before the potentially costly and time-consuming data annotation process. However, we note that such as an approach is currently unvalidated, so special care should be taken in ensuring the

---

[1]Except when using Spectral clustering, which includes its own manifold-based dimensionality reduction step as part of the clusterer.

correctness of the performance estimates by *e.g.*comparing with AMI scores on a smaller representative sample set which has been annotated. Additionally, we found evidence that silhouette scores do not relate to AMI well for randomly initialized networks, in agreement with Xu et al. (2022)—there is a transition during training from embeddings which are Gaussian-distributed to ones with structure lying on low-dimensional manifold—thus this metric is not meaningful at the start of training.

## 6 Conclusion

We empirically investigated how well the embeddings produced by pretrained networks can be clustered for data unseen during training. We considered two architectures trained on IN-1k using one of 5 methodologies (1 supervised, 4 self-supervised), deployed on 26 datasets using 5 distinct types of clusterer. Analyzing the performance of SSL encoders with classical clustering enables us to investigate what the embeddings represent in-of-themselves, without imposing a training objective aligned with the evaluation.

To cluster embeddings of a novel dataset, we suggest dimensionality reduction with UMAP (5–100 dim, we chose 50d), then use AC with L2, Ward, on the reduced embeddings. UMAP-reduction works best for all clusterers except Spectral, despite not being distance-preserving. We also show promising results that silhouette score can be used to evaluate SSL methods when no GT is available, especially when applied on UMAP-reduced embeddings. These results are indicative of the embedded dataset lying on a low-dimensional, non-linear manifold in the embedding space.

For datasets far outside the original IN-1k training domain, SSL encoders provide clustering in best agreement with the data annotations. For images near the training domain, SSL encoders fine-tuned on class-labels from the training domain perform best, but this gain in performance comes at a cost, greatly reducing the performance on Far-OOD data. Our work emphasizes the importance of the alignment between the model's training task and the downstream task its embeddings are applied on.

We hope this work will serve as an important baseline for future work toward methods of learning to cluster images with deep networks. Our work focused solely on encoders pretrained on ImageNet-1k, for clustering other image datasets, and thus we can only draw strong conclusions in this regime. We leave for future work investigation of the reverse transfer—taking encoders pretrained on other image datasets and performing clustering of ImageNet-1k—and the applicability of clustering in other domains.

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

## Appendices

## A    Impact Statement

In this paper we analyze self-supervised encoders from the perspective of clustering. While the main goal of the paper is to advance our collective understanding of self-supervised learning, we acknowledge that the clustering process may lead to the construction of clusters which amplify stereotypical or biased groupings. While we do test on face and medical datasets, we do want to caution against deploying unverified clustering approaches in these highly sensitive areas, and that the medical dataset results are not clinically validated.

## B    Limitations

While our evaluation has spanned a broad range of test datasets, we have only considered models pretrained on ImageNet-1k. Consequently, we have only studied the ability of models trained on ImageNet-1k to generalize to other datasets. While we anticipate that our findings would generalize to models trained on other datasets (with the unseen datasets being in- and out-domain changed to reflect the new training domain), this assumption has not been verified.

An aspect of changing the training data that is more likely to impact our findings is the diversity of the training data. Whilst models which are trained on a larger dataset will have a larger in-domain space, some data will still be out-of-domain and thus our considerations will be meaningful. However, the ability of models to generalize from larger datasets could be impacted differently depending on the pretraining paradigm.

Our primary evaluations use the difference in AMI between two conditions (e.g. supervised vs an SSL model). Although we do report absolute AMI values for individual models in Appendix H, it is not clear which results are "good" and which are "bad"—each evaluation dataset has a different intrinsic difficulty and we do not calibrate for this.

Our work was constrained to only one data modality: vision. While we anticipate that our findings would generalize to other modalities provided the pretraining paradigms are comparable, this is yet to be verified.

The clusterings we have performed were on the embeddings of fully-trained networks. The behaviour of untrained networks (and to a lesser extent, MAE-trained networks without a whole-stimulus target) was not consistent with that of the trained networks in some regards, as we note in §E.2 and as previously observed by (Xu et al., 2022). Consequently, our finding that the intrinsic silhouette score of clustered UMAP-reduced embeddings is correlated with the performance of the encoder on the target dataset may not be applicable to measuring performance in the middle of training, while the feature space is still transitioning from an amorphous distribution to a structured manifold subspace.

We considered only one type of supervised pretraining, cross-entropy. We assume that other supervised loss functions would produce a similar outcome.

In our work, we explored the effect of fine-tuning the self-supervised pretrained encoders on ImageNet-1k and found their behaviour was similar to models trained from scratch with cross-entropy. However, we did not investigate the behaviour of the encoder during training. Consequently, it is unclear when the transition from the SSL-pretrained encoder behaviour to supervised training occurs.

While we considered two architectures in this paper (ResNet-50 and ViT-B), they are not of similar capacities and so it is not possible for us to draw conclusions about which architecture generalizes best outside its training domain. Consequently, we make no claims in this regard.

## C    Evaluation Metrics Details

### C.1    Adjusted Mutual Information

Since we are evaluating the clustering on annotated datasets, we evaluated a candidate clustering assignment against the "ground-truth" cluster labels, from an information theoretic perspective. The Normalized Mutual

Information (NMI) between two label assignments $V$ an $U$ is defined as

$$\text{NMI}(U, V) = \frac{\text{MI}(U, V)}{\text{mean}(\text{H}(U) + \text{H}(V))}, \tag{1}$$

where $\text{MI}(U, V)$ is the mutual information between label assignments $V$ an $U$, and $\text{H}(\cdot)$ is the Shannon entropy of the considered label assignment (Zhou et al., 2024). NMI is a relative measure of the amount of information between two label sets, and hence is bounded between 0 and 1, with 1 occurring for a perfect match and 0 occurring when there is absolutely no mutual information between the label assignments. NMI has commonly been used to evaluate deep-learning based clustering methods, together with the clustering accuracy (Zhou et al., 2024).

However, NMI is not corrected for chance so its value can increase merely by increasing the number of clusters used (Vinh et al., 2010). In order to account for this, we use the adjusted mutual information metric proposed by Vinh et al. (2010), defined as

$$\text{AMI}(U, V) = \frac{\text{MI}(U, V) - \mathbb{E}[\text{MI}(U, V)]}{\text{mean}(\text{H}(U) + \text{H}(V)) - \mathbb{E}[\text{MI}(U, V)]}, \tag{2}$$

where $\mathbb{E}[\text{MI}(U, V)]$ is the expected value of the mutual information between the considered label assignments. Similar to NMI, an AMI of 1 represents a perfect agreement between label assignments, but a score of 0 indicates the typical score for a completely random label assignment (negative AMI scores are possible).

In addition to this, for some evaluations we used an alternative formulation of AMI where the denominator is based on the entropy of the true distribution only,

$$\text{AMI}_{\text{true}}(U, V) = \frac{\text{MI}(U, V) - \mathbb{E}[\text{MI}(U, V)]}{\text{H}(U) - \mathbb{E}[\text{MI}(U, V)]}. \tag{3}$$

Unlike the standard definition of AMI, this version corresponds to the fraction of entropy in the true labels, $U$, which is explained by observing clustering $V$. The stability in the comparison is helpful when comparing performance across experiments with different numbers of clusters.

## C.2 Silhouette Score

The silhouette score, $S$, is a clustering measure based on the intrinsic structure of the created clusters (Rousseeuw, 1987), defined as

$$S = \frac{1}{N} \sum_{i}^{N} \frac{a_i - b_i}{\max(a_i, b_i)}, \tag{4}$$

where $N$ is the total number of data points, $a_i$ is the average distance between data point $i$ and all other points assigned in the same cluster, and $b_i$ is the average distance from $i$ to all points in the next nearest cluster. $S$ is bounded between $-1$ and 1. A score near 0 indicates that clusters are overlapping, as the data points are equally close to several clusters. A score of 1 indicates that the clusters are dense with little within-cluster distance, and thereby well-clustered. Negative values may indicate an inaccurate clustering. Since $S$ is defined based on the relative distances of data points, it can be computed without reference to a set of ground-truth cluster assignments.

## D   Encoder Training Details

The supervised encoders were obtained from the `torchvision` library. We use the weights defined in the following enums:

- ResNet-50 [recipe][2]: `torchvision.models.ResNet50_Weights.IMAGENET1K_V2`

---

[2]https://github.com/pytorch/vision/issues/3995#issuecomment-1013906621

- ViT-B [recipe][3]: `torchvision.models.ViT_B_16_Weights.IMAGENET1K_V1`

For the training details of each of the SSL encoders, please refer to their respective papers, cited accordingly in §3.1.

For our fine-tuning step, we used the method defined by He et al. (2022). When fine-tuning the ResNet architectures, we modified the method to omit the per-layer LR scaling.

## E  Clustering Parameter Search Details

As described in §3.5, we conducted our clustering parameter search on subsets of the train partition for ImageNet-1k, Imagenette, and Imagewoof. In this section, we provide further details on the parameter search process.

For the full array of selected clustering parameters, see Table 3 and Table 4.

### E.1  Preliminary Configuration

In our initial explorations, we found that the performance of most clusterers worked reasonably well with their default parameters, and thus initialized our search using the default parameters for the clusterers. There were three exceptions to this. (1) For Spectral Clustering, the default affinity matrix computation method was using a radial basis function which could not scale to the size of the data. We thus changed the affinity calculation method to use a graph of nearest neighbors instead, which scales better with dimensionality and number of samples. An initial search over the number of neighbors to use indicated the method would perform well across a range of values. Additionally, we changed the default label assignment method from `kmeans` to `cluster_qr`, as the latter has no tuning parameters and is known to perform consistently well. (2) For Affinity Propagation, we found that although PCA-reduced embeddings were insensitive to the choice of damping, we found that UMAP- and PaCMAP-reduced embeddings would not converge when using the default damping value of 0.5. An initial search over the damping using 20-d reduced embeddings with PCA, UMAP, and PaCMAP indicated that a damping value of 0.9 would give robust performance across all dim-reduction methods and all pretrained encoders, hence we adopted this as our default value. Furthermore, we increased the maximum number of iterations for K-Means and Affinity Propagation to 1 000 (from 300 and 200), to help ensure convergence of the algorithms. (3) For HDBSCAN, we noticed that for some encoders the clusterer would select very few clusters for Imagenette and Imagewoof, which reduced its performance. We verified, by clustering the full embeddings, that decreasing the maximum cluster size mitigated this problem. We thus set the maximum cluster size to be a generous 20% of the number of samples throughout the remainder of the search and subsequent experiments, so as to ensure HDBSCAN would not produce a degenerate solution but without forcing it to produce a certain number of clusters.

Our parameter search was conducted as a series of line-searches, in which we optimized one clustering parameter at a time. Once a parameter was optimized, it was frozen and we progressed to optimizing the next parameter. To begin the search, we used the default parameters of the clustering methods as defined in SCIKIT-LEARN, except for the maximum number of iterations (1 000) and the Affinity Propagation damping (0.9). For K-Means and AC, we provided the number of annotated classes within the dataset (1 000 or 10) as number of clusters to produce. Unless stated otherwise, throughout each stage of the search we took the weighted-average AMI over the three datasets, weighting ImageNet-1k twice as much as the two 10-class subsets, and selected the parameter value which yielded the highest weighted-average AMI. For AP, it was infeasible to conduct this search on IN-1k due to its compute and memory scaling w.r.t. number of samples; hence we optimized its parameters using Imagenette and Imagewoof only.

The random seed (for random, numpy, dimensionality reducer, and clusterer) was held fixed at one value (100) throughout the parameter search, and then changed to a different value (1) for the final experiments.

We used SCIKIT-LEARN (sklearn) version 1.3.1 for our search and experiments. The initial parameters used for each clusterer were as follows:

---

[3]https://github.com/pytorch/vision/tree/806dba6/references/classification#vit_b_16

- K-Means
  - `n_clusters = [number of GT classes]`
  - `algorithm = "lloyd"`
  - `init = "k-means++"`
  - `n_init = 1`
  - `tol = 0.0001`
  - `max_iter = 1000`
  - `random_state = 100`

- Spectral Clustering
  - `n_clusters = [number of GT classes]`
  - `n_components = n_clusters`
  - `affinity = "nearest_neighbors"`
  - `assign_labels = "cluster_qr"`
  - `eigen_solver = "arpack"`
  - `eigen_tol = 0.0`
  - `n_components = None`
  - `n_neighbors = 10`
  - `random_state = 100`

- Agglomerative Clustering
  - `n_clusters = [number of GT classes]`
  - `distance_threshold = None`
  - `metric = "euclidean"`
  - `linkage = "ward"`
  - `compute_full_tree = "auto"`

- Affinity Propagation
  - `damping = 0.9`
  - `convergence_iter = 15`
  - `affinity = "euclidean"`
  - `max_iter = 1000`
  - `random_state = 100`

- HDBSCAN
  - `min_cluster_size = 5`
  - `min_samples = min_cluster_size`
  - `max_cluster_size = [20% of the number of samples]`
  - `metric = "euclidean"`
  - `cluster_selection_method = "eom"`
  - `cluster_selection_epsilon = 0.0`
  - `alpha = 1.0`
  - `algorithm = "auto"`
  - `leaf_size = 40`
  - `allow_single_cluster = False`

Table 3: **Clustering parameters for raw images and ResNet-50 encoders.** We present the parameters discovered by our search on ImageNet train data, and subsequently used throughout our main experiments as presented in the paper. Some parameters are specific to particular clusterers and hence do not have a value for the other clusterers. The dimension reduction value indicates the number of reduced dimensions if larger than 1, or the target variance explained if less than 1. The parameters for AC w/ C were the same as for AC w/o C, except the distance threshold was not specified, instead being automatically determined from the target number of clusters. Continued in Table 4.

| | | | | | | | Agg. Clustering | | Spectral | Aff. Prop. |
|---|---|---|---|---|---|---|---|---|---|---|
| Arch. | Encoder | FT | Clusterer | Dim Reduction | | Metric | Linkage | Dist. Thr. | # Neigh. | Damping |
| — | Raw image | | K-Means | PCA | 0.90 | – | – | – | – | – |
| | | | Spectral | None | | – | – | – | 10 | – |
| | | | AC w/o C | PCA | 200 | cosine | average | 0.71 | – | – |
| | | | Affinity Prop | PCA | 0.80 | – | – | – | – | 0.85 |
| | | | HDBSCAN | UMAP | 50 | $L2$ | – | – | – | – |
| RN50 | Rand. | | K-Means | PCA | 0.95 | – | – | – | – | – |
| | | | Spectral | PCA | 200 | – | – | – | 50 | – |
| | | | AC w/o C | PCA | 200 | $L\infty$ | average | 10.00 | – | – |
| | | | Affinity Prop | PCA | 0.90 | – | – | – | – | 0.90 |
| | | | HDBSCAN | UMAP | 50 | $L1$ | – | – | – | – |
| | X-Ent. | | K-Means | UMAP | 50 | – | – | – | – | – |
| | | | Spectral | None | | – | – | – | 20 | – |
| | | | AC w/o C | UMAP | 50 | $L2$ | ward | 2.00 | – | – |
| | | | Affinity Prop | UMAP | 50 | – | – | – | – | 0.90 |
| | | | HDBSCAN | UMAP | 50 | $L2$ | – | – | – | – |
| | MoCo-v3 | | K-Means | UMAP | 50 | – | – | – | – | – |
| | | | Spectral | None | | – | – | – | 30 | – |
| | | | AC w/o C | UMAP | 50 | $L2$ | ward | 10.00 | – | – |
| | | | Affinity Prop | UMAP | 50 | – | – | – | – | 0.75 |
| | | | HDBSCAN | UMAP | 50 | $L2$ | – | – | – | – |
| | DINO | | K-Means | UMAP | 50 | – | – | – | – | – |
| | | | Spectral | PCA | 0.80 | – | – | – | 10 | – |
| | | | AC w/o C | UMAP | 50 | $L2$ | average | 0.50 | – | – |
| | | | Affinity Prop | UMAP | 50 | – | – | – | – | 0.90 |
| | | | HDBSCAN | UMAP | 50 | $L1$ | – | – | – | – |
| | VICReg | | K-Means | UMAP | 50 | – | – | – | – | – |
| | | | Spectral | None | | – | – | – | 10 | – |
| | | | AC w/o C | UMAP | 50 | $L2$ | average | 0.50 | – | – |
| | | | Affinity Prop | UMAP | 50 | – | – | – | – | 0.80 |
| | | | HDBSCAN | UMAP | 50 | $L1$ | – | – | – | – |
| | MoCo-v3 | ✓ | K-Means | UMAP | 50 | – | – | – | – | – |
| | | ✓ | Spectral | None | | – | – | – | 30 | – |
| | | ✓ | AC w/o C | UMAP | 50 | $L2$ | ward | 2.00 | – | – |
| | | ✓ | Affinity Prop | UMAP | 50 | – | – | – | – | 0.95 |
| | | ✓ | HDBSCAN | UMAP | 50 | $L\infty$ | – | – | – | – |
| | DINO | ✓ | K-Means | UMAP | 50 | – | – | – | – | – |
| | | ✓ | Spectral | PCA | 0.80 | – | – | – | 20 | – |
| | | ✓ | AC w/o C | UMAP | 50 | $L2$ | ward | 2.00 | – | – |
| | | ✓ | Affinity Prop | UMAP | 50 | – | – | – | – | 0.90 |
| | | ✓ | HDBSCAN | UMAP | 50 | $L1$ | – | – | – | – |
| | VICReg | ✓ | K-Means | UMAP | 50 | – | – | – | – | – |
| | | ✓ | Spectral | None | | – | – | – | 20 | – |
| | | ✓ | AC w/o C | UMAP | 50 | $L2$ | ward | 2.00 | – | – |
| | | ✓ | Affinity Prop | UMAP | 50 | – | – | – | – | 0.90 |
| | | ✓ | HDBSCAN | UMAP | 50 | $L2$ | – | – | – | – |

Table 4: **Clustering parameters for ViT-B encoders.** Continues Table 3 to show parameters for ViT-B encoders. For MAE with Spectral Clustering, we found standardizing the data with a z-score (and not applying PCA) yielded the best performance.

| Arch. | Encoder | FT | Clusterer | Dim Reduction | | Metric | Agg. Clustering | | Spectral | Aff. Prop. |
|---|---|---|---|---|---|---|---|---|---|---|
| | | | | | | | Linkage | Dist. Thr. | # Neigh. | Damping |
| ViT-B | Rand. | | K-Means | PCA | 100 | – | – | – | – | – |
| | | | Spectral | PCA | 0.95 | – | – | – | 50 | – |
| | | | AC w/o C | PCA | 0.85 | $L\infty$ | average | 2.00 | – | – |
| | | | Affinity Prop | PCA | 0.90 | – | – | – | – | 0.95 |
| | | | HDBSCAN | UMAP | 50 | $L1$ | – | – | – | – |
| | X-Ent. | | K-Means | UMAP | 50 | – | – | – | – | – |
| | | | Spectral | PCA | 0.70 | – | – | – | 30 | – |
| | | | AC w/o C | UMAP | 50 | $L2$ | ward | 2.00 | – | – |
| | | | Affinity Prop | UMAP | 50 | – | – | – | – | 0.90 |
| | | | HDBSCAN | UMAP | 50 | $L\infty$ | – | – | – | – |
| | MoCo-v3 | | K-Means | UMAP | 50 | – | – | – | – | – |
| | | | Spectral | PCA | 0.85 | – | – | – | 50 | – |
| | | | AC w/o C | UMAP | 50 | $L\infty$ | average | 1.00 | – | – |
| | | | Affinity Prop | UMAP | 50 | – | – | – | – | 0.75 |
| | | | HDBSCAN | UMAP | 50 | $L2$ | – | – | – | – |
| | DINO | | K-Means | UMAP | 50 | – | – | – | – | – |
| | | | Spectral | PCA | 0.90 | – | – | – | 10 | – |
| | | | AC w/o C | UMAP | 50 | $L2$ | average | 0.20 | – | – |
| | | | Affinity Prop | UMAP | 50 | – | – | – | – | 0.85 |
| | | | HDBSCAN | UMAP | 50 | $L1$ | – | – | – | – |
| | MAE (CLS) | | K-Means | PCA | 0.95 | – | – | – | – | – |
| | | | Spectral | z-score only | | – | – | – | 10 | – |
| | | | AC w/o C | PCA | 0.90 | cosine | average | 0.71 | – | – |
| | | | Affinity Prop | PCA | 200 | – | – | – | – | 0.60 |
| | | | HDBSCAN | PCA | 0.95 | $L2$ | – | – | – | – |
| | MAE (avg) | | K-Means | PCA | 0.90 | – | – | – | – | – |
| | | | Spectral | z-score only | | – | – | – | 30 | – |
| | | | AC w/o C | PCA | 0.85 | cosine | average | 0.71 | – | – |
| | | | Affinity Prop | UMAP | 50 | – | – | – | – | 0.60 |
| | | | HDBSCAN | UMAP | 50 | $L1$ | – | – | – | – |
| | MoCo-v3 | ✓ | K-Means | UMAP | 50 | – | – | – | – | – |
| | | ✓ | Spectral | PCA | 0.95 | – | – | – | 50 | – |
| | | ✓ | AC w/o C | UMAP | 50 | $L2$ | ward | 2.00 | – | – |
| | | ✓ | Affinity Prop | UMAP | 50 | – | – | – | – | 0.95 |
| | | ✓ | HDBSCAN | UMAP | 50 | $L\infty$ | – | – | – | – |
| | DINO | ✓ | K-Means | UMAP | 50 | – | – | – | – | – |
| | | ✓ | Spectral | PCA | 0.90 | – | – | – | 50 | – |
| | | ✓ | AC w/o C | UMAP | 50 | $L2$ | ward | 2.00 | – | – |
| | | ✓ | Affinity Prop | UMAP | 50 | – | – | – | – | 0.90 |
| | | ✓ | HDBSCAN | UMAP | 50 | $L\infty$ | – | – | – | – |
| | MAE (avg) | ✓ | K-Means | UMAP | 50 | – | – | – | – | – |
| | | ✓ | Spectral | PCA | 0.75 | – | – | – | 50 | – |
| | | ✓ | AC w/o C | UMAP | 50 | $L2$ | ward | 2.00 | – | – |
| | | ✓ | Affinity Prop | UMAP | 50 | – | – | – | – | 0.90 |
| | | ✓ | HDBSCAN | UMAP | 50 | $L\infty$ | – | – | – | – |

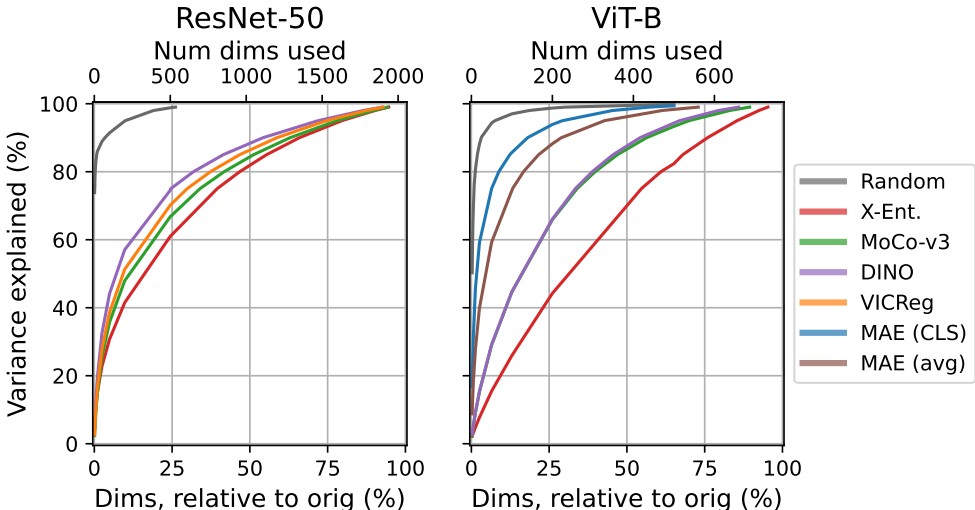

Figure 5: **Percentage of variance explained by PCA-reduced embeddings**. We show the fraction of the total variance of the data which is explained by the first $N$ PCA dimensions. The number of dimensions included is represented both in absolute terms (upper x-axes) and relative to the number of dimensions of the original embeddings (lower x-axes).

## E.2 Dimensionality Reduction

First, as the curse of dimensionality can negatively affect the performance of the considered clustering methods (Bellman et al., 1957), we searched for an appropriate dimensionality reduction process. We compared the performance of using the original un-reduced feature embedding space (2048-d for ResNet-50, 768-d for ViT-B) against applying PCA (Pearson, 1901), UMAP (McInnes et al., 2018), or PaCMAP (Wang et al., 2021) to reduce the number of dimensions. Specifically, we considered reducing the feature embeddings to [2, 5, 10, 20, 50, 100, 200] with either PCA, UMAP, or PaCMAP. We also considered using PCA to reduce the number of dimensions to capture a target fraction of total variance of the data [0.75, 0.8, 0.85, 0.9, 0.95]; this differs from using a fixed number of dimensions as the method may select a different number of dimensions for each of the three datasets.

To perform PCA, we first took the z-score of each dimension and then used the default parameters of scikit-learn (Pedregosa et al., 2011), without whitening the data.

To perform UMAP, we set the number of neighbours considered to 30 (increased from the default of 15) and set the minimum distance to 0.0 (decreased from the default of 0.1), following the recommendations of McInnes (2018); we otherwise used the default parameters of umap (McInnes et al., 2018). The dimensionality reduction distance metric was always set to euclidean ($L_2$), irrespective of the distance metric used by the downstream clusterer.

To perform PaCMAP, used the authors' implementation, disabling the PCA-reduction preprocessing step but otherwise using the default parameters. The default number of neighbors was automatically determined from the size of the dataset as $10 + \max(0, 15 \, (\log_{10}(N) - 4))$. We found the performance of PaCMAP was consistently worse than UMAP, and so we also considered setting the number of neighbours to 30 to match UMAP; however this did not lead to a significant change in its performance.

For raw images and randomly initialized (untrained) networks, we found that PCA reduction typically performed best (see Table 3 and Table 4), and was optimal with relatively large number of dimensions (at least 100), as a large number of dimensions was needed to capture the majority of the variance of the data (shown in Figure 5). However, the majority of trained encoders performed best with UMAP-reduced embeddings and were insensitive to the choice of dimension, with minimal change in mean AMI across the range 5 to 200. Thus for consistency, we selected a 50-dim UMAP reduction for all encoders/clusterers where

UMAP performed best. The MAE-trained ViT-B encoder bucked this trend and performed poorly with UMAP reduction across all clusterers (and all three datasets), yielding better performance when using PCA instead. This was true for both the CLS token embedding (which was not connected to any loss during the training of the network), and when taking the average over all the embeddings of patch tokens (for some but not all clusterers).

For Spectral Clustering, we found using PCA-reduced or unreduced embeddings as the input to the clusterer yielded better performance than using UMAP-reduced embeddings. This is because the Spectral Clustering methodology already includes a manifold-based dimensionality reduction step (taking the eigenvalues of the neighborhood-based affinity matrix) as part of its pipeline. Performing both UMAP and Spectral dimensionality reduction reduced the performance, as UMAP is not distance-preserving.

These results emphasize that the output of untrained networks are distributed amorphously, as is the case for the raw stimuli in pixel-space, whereas the output of encoders trained on a whole-stimulus task lie on a low-dimensional manifold which can be discovered by manifold-based dimensionality reduction methods such as UMAP or Spectral Clustering, but not linear methods such as PCA. Hence using manifold-based dimensionality reduction provides the best clustering results, even for clusterers which rely on distance metrics between pairs of samples and despite the fact these distance metrics are not preserved by the dimensionality reduction. Meanwhile, for encoders trained on a local-feature task—MAE (avg)—the output is somewhere in between the two, with PCA and UMAP reduced embeddings giving comparable performance.

In the subsequent stages of the parameter search, we iterated over the per-method specific parameters, whilst using the dimensionality reductions per encoder selected in this stage.

### E.3    K-Means

For K-Means, we did not optimize any parameters other than the dimensionality reduction method. We used the `kmeans++` initialization (Arthur & Vassilvitskii, 2007) throughout our experiments, with 1 initialization per clustering.

### E.4    Spectral Clustering

For Spectral Clustering, we optimized the number of neighbors used when building the affinity matrix over the search space [5, 10, 20, 30, 50, 100]. We found the performance was not very sensitive to the neighborhood size, with optimal values in the range 10–50 depending on the encoder.

After fixing the neighborhood size, we investigated the effect of the number of eigenvectors (components). We found the number of components which yielded the best performance varied greatly between Imagenette/Imagewoof and ImageNet-1k (10, and 100–1 000, respectively). As this was around the same as the target number of clusters, which was the default parameter, we retained the default behaviour.

### E.5    Affinity Propagation

For Affinity Propagation, we optimized the damping parameter over the search space [0.5, 0.6, 0.7, 0.75, 0.8, 0.85, 0.9, 0.95, 0.98]. Then, after freezing the amount of damping, we investigated the effect of the convergence stopping threshold over the search space [5, 8, 10, 15, 20, 25, 30]. We found the performance was insensitive to the stopping threshold, and so froze it at the default value of 15 for all encoders.

### E.6    HDBSCAN

For HDBSCAN, we investigated the effect of the distance metric and cluster selection method jointly. We considered distance metrics $\{L1, L2, L\infty\}$, and both the excess of mass (eom) and leaf selection methods. We found the eom selection method universally outperformed leaf in terms of AMI, and there was minimal effect from the choice of distance metric.

We used the default minimum cluster size of 5 throughout our search and consequently also for the majority of our experiments. However, for BIOSCAN-1M, CelebA, and UTKFace (where some classes have only 1 or 2 occurrences in the test set) we reduced the minimum cluster size to 2.

### E.7 Agglomerative Clustering

For AC, continuing to use the "ground-truth" number of classes as the number of clusters, we evaluated all combinations of distance metric {$L1$, $L2$, $L\infty$, cosine} and linkage method {ward ($L2$ only), complete, average, single}, for 13 options in total. For each encoder, we selected the metric and linkage which yielded the best weighted-average AMI over the three datasets. This selection completed the parameter options to use for AC w/ C.

Finally, for AC w/o C, we selected the distance threshold to use for each encoder. The distance threshold provides an alternative stopping criteria for AC so it does not need to know the number of clusters *a priori*. To make the distance threshold more likely to be comparable across embeddings from different datasets, after dimensionality reduction we standardized the embeddings by subtracting the mean of each dimension and dividing by the average standard-deviation across all dimensions. This spherically rescales the distances of the space without stretching dimensions relative to each other and thus without changing the relative importance of each dimension toward the distance between samples. We also divided by the number of dimensions for encoders where the $L1$ metric was selected, or by the square-root of the number of dimensions for encoders where the $L2$ metric was selected. This process keeps the expected distance between samples similar even if the dimensionality differed between reduced embeddings.

For each encoder, we fit the clusterer on each of the 3 datasets for 21 distance thresholds sampled logarithmically from 0.001 to 5000.0. For each of the three datasets, we scaled the values across the distance thresholds relative to the maximum AMI to make them more comparable—since the AMI falls to 0 if the distance threshold is too high (only one cluster) or too low (every sample in its own cluster), rescaling the AMI in this way gives each dataset the same dynamic range. We then selected the distance threshold which yielded the highest weighted-average relative-AMI.

We found that embeddings which had been reduced with UMAP had a broad curve for the distance threshold, but PCA-reduced embeddings were highly sensitive to the distance threshold with a narrow peak across only a pair of values in our search grid. Because of this, we refined the search for the distance threshold on PCA-reduced embeddings at twice the resolution before picking the best distance threshold value.

## F Clustering Raw Images

To cluster the raw images, we used an image size of 32×32×3 throughout our parameter search, reduced by resizing the shortest side to 32 pixels and cropping to a square. For the final experiments, we used the same process, except for MNIST and Fashion-MNIST which have smaller images than this and hence we dimensionality-reduced them starting from 28×28×3 images.

## G Computational Resource Requirements

In this section, we describe the computational requirements of our experiments. All experiments were performed on a compute cluster with the job utilizing two CPU cores (2x Intel Xeon Gold 6148 CPU @ 2.40GHz).

The amount of memory used per job varied depending on the demands of the clusterer and the size of the dataset. An upper-bound for the memory requirements of each experiment is shown in Table 5.

The total runtime of our parameter search was 4.9 years. The total runtime of the clustering results shown in the main figures (1, 3, etc) and tables (7–18) was 351 days. Including auxiliary results, preliminary experiments, and otherwise discarded experiments, the total runtime of the CPU-only clustering steps for this project was 7.6 years. Typical runtimes for each clusterer and dataset are shown in Table 6.

Table 5: **Memory requirements (GB).** We indicate an upper-bound on the amount of RAM required, in GB, to cluster the test set of each dataset using each clustering method.

| Dataset | K-Means | Spectral | Agglom. | Affinity Prop | HDBSCAN |
|---|---|---|---|---|---|
| Imagenette | 1 | 2 | 2 | 1 | 1 |
| Imagewoof | 1 | 2 | 2 | 1 | 1 |
| ImageNet-1k | 4 | 20 | 20 | 72 | 4 |
| ImageNet-v2 | 2 | 6 | 6 | 6 | 2 |
| CIFAR-10 | 2 | 6 | 6 | 6 | 2 |
| CIFAR-100 | 2 | 6 | 6 | 6 | 2 |
| ImageNet-9 (all var.) | 2 | 4 | 4 | 2 | 2 |
| ImageNet-R | 4 | 16 | 16 | 48 | 4 |
| ImageNet-S | 4 | 20 | 20 | 72 | 4 |
| ImageNet-O | 1 | 2 | 2 | 1 | 1 |
| LSUN | 2 | 6 | 6 | 6 | 2 |
| Places365 | 4 | 16 | 16 | 48 | 4 |
| FGVC Aircraft | 1 | 2 | 2 | 1 | 1 |
| Stanford Cars | 2 | 6 | 6 | 6 | 2 |
| Oxford Flowers 102 | 2 | 4 | 4 | 2 | 2 |
| BIOSCAN-1M | 4 | 16 | 16 | 48 | 4 |
| NABirds | 4 | 16 | 16 | 48 | 4 |
| iNaturalist-2021 | 6 | 72 | 72 | 292 | 6 |
| CelebA | 4 | 12 | 12 | 12 | 4 |
| UTKFace | 2 | 4 | 4 | 2 | 2 |
| BreakHis | 1 | 2 | 2 | 1 | 1 |
| DTD | 1 | 2 | 2 | 1 | 1 |
| EuroSAT | 2 | 4 | 4 | 2 | 2 |
| MNIST | 2 | 6 | 6 | 6 | 2 |
| FashionMNIST | 2 | 6 | 6 | 6 | 2 |
| SVHN | 4 | 16 | 16 | 48 | 4 |

The fine-tuning of the SSL encoders in §4.4 was conducted on two Nvidia A40 GPUs following the MAE fine-tuning schedule from He et al. (2022). Each training run took approximately 43 hours, resulting in a total of 11 GPU compute days.

Table 6: **Clustering job runtime.** For each clustering method, we show the runtime of the clustering process (including dimensionality reduction, as applicable) on each dataset in seconds, minutes, or hours. We take the median value across all encoders, excluding raw pixel and randomized (untrained) networks. See Appendix H for dataset abbreviations. Background: from fastest (white) to slowest (red) per dataset. DNF: did not complete within 48 h job allocation.

| Clusterer | In-domain | | | | | Domain-shift | | | | Near-OOD | | | Fine-grained | | | | | | Far-OOD | | | | | | | |
| --- | --- | --- | --- | --- | --- | --- | --- | --- | --- | --- | --- | --- | --- | --- | --- | --- | --- | --- | --- | --- | --- | --- | --- | --- | --- | --- |
| | IN1k | INv2 | C10 | C100 | IN9 | 9-FG | 9-MR | IN-R | IN-S | IN-O | LSU | P365 | Air | Cars | F102 | Bio | Birds | iNat | CelA | UTKF | BHis | DTD | ESAT | MNST | Fash | SVHN |
| K-Means | 9.0 h | 19.2 min | 20.0 min | 19.4 min | 3.4 min | 3.2 min | 4.6 min | 4.1 h | 11.2 h | 1.0 min | 1.9 min | 5.3 h | 2.7 min | 11.3 min | 9.1 min | 2.9 h | 2.5 h | 31.6 h | 1.6 h | 7.1 min | 2.0 min | 62.7 s | 3.1 min | 16.3 min | 18.5 min | 2.4 h |
| Spectral | 22.2 h | 49.0 min | 37.0 min | 55.5 min | 6.8 min | 6.6 min | 7.3 min | 9.0 h | 28.4 h | 1.6 min | 3.3 min | 18.7 h | 5.1 min | 27.0 min | 13.6 min | 8.0 h | 5.1 h | DNF | 4.4 h | 14.5 min | 2.5 min | 90.9 s | 7.2 min | 25.8 min | 25.6 min | 4.7 h |
| AC w/ C | 7.8 h | 18.7 min | 18.8 min | 18.0 min | 3.7 min | 3.1 min | 3.4 min | 3.7 h | 8.3 h | 1.1 min | 1.7 min | 5.3 h | 2.4 min | 9.0 min | 8.7 min | 2.0 h | 1.9 h | 18.7 h | 1.3 h | 6.1 min | 1.8 min | 54.9 s | 2.8 min | 14.3 min | 18.3 min | 2.2 h |
| AC w/o C | 9.2 h | 17.3 min | 17.2 min | 14.8 min | 3.5 min | 3.4 min | 4.1 min | 3.0 h | 9.5 h | 1.1 min | 2.4 min | 5.5 h | 2.3 min | 12.4 min | 8.1 min | 2.5 h | 2.0 h | 27.3 h | 1.4 h | 6.0 min | 1.9 min | 62.6 s | 3.0 min | 18.9 min | 18.0 min | 2.3 h |
| Affinity Prop | 10.4 h | 18.6 min | 22.1 min | 19.7 min | 5.4 min | 4.0 min | 3.5 min | 3.8 h | 11.3 h | 1.0 min | 2.3 min | 5.5 h | 2.6 min | 15.3 min | 9.0 min | 2.5 h | 2.5 h | 43.8 h | 1.2 h | 7.9 min | 2.3 min | 67.6 s | 3.8 min | 23.2 min | 23.4 min | 1.9 h |
| HDBSCAN | 5.3 h | 15.9 min | 11.3 min | 11.6 min | 4.3 min | 3.0 min | 3.5 min | 2.2 h | 6.4 h | 1.0 min | 1.8 min | 3.6 h | 1.6 min | 10.0 min | 6.9 min | 1.9 h | 1.3 h | 14.9 h | 1.0 h | 7.7 min | 1.5 min | 66.3 s | 2.6 min | 12.7 min | 11.3 min | 1.0 h |

# H AMI Results for Individual Datasets

In this section, we tabulate the results for clustering the embeddings of each of the test datasets used in the main results (26 datasets) with each of the encoders (raw images, 2 random networks, 2 supervised encoders, 7 SSL encoders, 6 SSL+FT encoders; for 18 encoders total), using each of the clustering methods (6; counting both AC w/ and w/o C). This yields a total of 2 808 clustering results.

For each dataset, we clustered the images from the test partition only. In cases where there is no public test partition, but there is a public validation partition (e.g. ImageNet-1k), we evaluated the clustering on the validation partition. For BIOSCAN-1M, we use the splits from Gong et al. (2025) and evaluate on the union of their key partitions and test partitions. For some datasets, no partitioning is indicated in the dataset release, and we partitioned these as follows. For BreakHis, we used a random test split of 40% of the data, stratified on the joint distribution of tumor type and image magnification. For EuroSAT, we used a stratified random test split of 15% of the data. For UTKFace, we use a random test split of 25% of the data, stratified over age and approximately stratified over age and gender within this.

The datasets used in the experiments, described in Table 1, are abbreviated here as follows:

- In-domain
    - IN1k: ImageNet-1k (ILSVRC 2012)
    - INv2: ImageNet-v2
    - C10: CIFAR-10
    - C100: CIFAR-100
    - IN9: ImageNet-9 – Original images

- Domain-shift
    - 9-FG: ImageNet-9 – Foreground-only
    - 9-MR: ImageNet-9 – Mixed-random
    - IN-R: ImageNet-Rendition
    - IN-S: ImageNet-Sketch

- Near-OOD
    - IN-O: ImageNet-O
    - LSU: Large-scale Scene Understanding (LSUN)
    - P365: Places365

- Fine-grained
    - Air: FGVC Aircraft
    - Cars: Stanford Cars
    - F102: Oxford Flowers 102
    - Bio: BIOSCAN-1M
    - Birds: NABirds
    - iNat: iNaturalist-2021

- Far-OOD
    - CelA: CelebA
    - UTKF: UTKFace
    - BHis: BreakHis
    - DTD: Describable Textures Dataset
    - ESAT: EuroSAT
    - MNST: Modified National Institute of Standards and Technology database (MNIST)
    - Fash: Fashion-MNIST
    - SVHN: Street View House Numbers

Table 7: **Adjusted mutual information, (AMI; %) averaged over clusterers.** These results are used to create the figures of the main paper. The AMI shown is averaged over K-Means, Spectral, AC w/ C, AC w/o C, AP, and HDBSCAN results (except iNat, which excludes Spectral). See Tables 9–14 for individual clusterers. See Appendix H for dataset abbreviations. **Bold**: best encoder per dataset (or within 0.5 of best). Underlined: best encoder per architecture (or within 0.5 best). Background: from median AMI (white) to max (blue) per dataset.

| Encoder | FT | In-domain | | | | | Domain-shift | | | | Near-OOD | | | Fine-grained | | | | | | Far-OOD | | | | | | | |
|---|---|---|---|---|---|---|---|---|---|---|---|---|---|---|---|---|---|---|---|---|---|---|---|---|---|---|---|
| | | IN1k | INv2 | C10 | C100 | IN9 | 9-FG | 9-MR | IN-R | IN-S | IN-O | LSU | P365 | Air | Cars | F102 | Bio | Birds | iNat | CelA | UTKF | BHis | DTD | ESAT | MNST | Fash | SVHN |
| Raw image | | 2 | 1 | 9 | 10 | 5 | 5 | 1 | 2 | 9 | 6 | 6 | 4 | 4 | 2 | 16 | 8 | 3 | 1 | 4 | 3 | 13 | 5 | 21 | 54 | 48 | 2 |
| **RN50** — Rand. | | 2 | 1 | 4 | 7 | 3 | 6 | 2 | 2 | 9 | 4 | 4 | 3 | 3 | 2 | 13 | 9 | 2 | 0 | 2 | 1 | 14 | 4 | 22 | 37 | 37 | 1 |
| X-Ent. | | 70 | 45 | 57 | 50 | 69 | 70 | 60 | 34 | 40 | 60 | 58 | 37 | 15 | 22 | 62 | 21 | 40 | 12 | 8 | 10 | 26 | 47 | 66 | 71 | 63 | 6 |
| MoCo-v3 | | 47 | 26 | 57 | 48 | 70 | 61 | 48 | 26 | 36 | 37 | 52 | 32 | 19 | 11 | 76 | 24 | 29 | 8 | 11 | 11 | 30 | 49 | 68 | 82 | 64 | 12 |
| DINO | | 44 | 24 | 43 | 40 | 70 | 64 | 43 | 18 | 28 | 36 | 50 | 34 | 16 | 13 | 78 | 29 | 21 | 7 | 11 | 11 | 43 | 51 | 74 | 70 | 61 | 3 |
| VICReg | | 45 | 26 | 46 | 43 | 69 | 63 | 40 | 20 | 31 | 38 | 50 | 32 | 14 | 12 | 78 | 28 | 21 | 8 | 11 | 11 | 36 | 52 | 76 | 75 | 64 | 5 |
| MoCo-v3 | ✓ | 69 | 46 | 59 | 50 | 77 | 70 | 64 | 35 | 42 | 67 | 60 | 37 | 17 | 22 | 59 | 17 | 40 | 11 | 8 | 10 | 22 | 49 | 60 | 58 | 63 | 7 |
| DINO | ✓ | 69 | 46 | 57 | 50 | 75 | 68 | 62 | 34 | 41 | 68 | 56 | 36 | 16 | 21 | 59 | 17 | 40 | 11 | 8 | 9 | 22 | 47 | 62 | 67 | 62 | 5 |
| VICReg | ✓ | 68 | 45 | 57 | 49 | 75 | 67 | 64 | 33 | 39 | 66 | 57 | 36 | 15 | 22 | 62 | 18 | 40 | 11 | 8 | 10 | 22 | 50 | 61 | 60 | 64 | 6 |
| **ViT-B** — Rand. | | 3 | 1 | 9 | 9 | 6 | 4 | 1 | 3 | 8 | 6 | 8 | 5 | 3 | 2 | 18 | 12 | 3 | 1 | 3 | 2 | 15 | 8 | 25 | 14 | 20 | 0 |
| X-Ent. | | 75 | 54 | 76 | 63 | 61 | 61 | 51 | 38 | 43 | 67 | 62 | 39 | 18 | 24 | 67 | 21 | 38 | 12 | 8 | 9 | 25 | 45 | 61 | 73 | 63 | 3 |
| MoCo-v3 | | 57 | 35 | 72 | 60 | 62 | 62 | 44 | 26 | 36 | 36 | 60 | 37 | 14 | 11 | 78 | 29 | 29 | 10 | 10 | 11 | 37 | 55 | 70 | 75 | 58 | 3 |
| DINO | | 64 | 42 | 66 | 60 | 72 | 68 | 61 | 33 | 42 | 44 | 63 | 40 | 20 | 13 | 88 | 32 | 44 | 14 | 11 | 10 | 40 | 58 | 74 | 74 | 63 | 2 |
| MAE (CLS) | | 21 | 11 | 26 | 24 | 38 | 39 | 18 | 10 | 23 | 17 | 29 | 18 | 9 | 5 | 45 | 15 | 12 | 4 | 7 | 6 | 28 | 29 | 49 | 56 | 50 | 2 |
| MAE (avg) | | 23 | 11 | 29 | 27 | 44 | 41 | 15 | 10 | 23 | 19 | 36 | 22 | 8 | 6 | 54 | 13 | 11 | 3 | 8 | 8 | 30 | 37 | 52 | 56 | 48 | 1 |
| MoCo-v3 | ✓ | 77 | 57 | 77 | 67 | 64 | 53 | 52 | 44 | 48 | 61 | 64 | 40 | 19 | 27 | 70 | 21 | 44 | 14 | 9 | 10 | 24 | 48 | 57 | 70 | 64 | 2 |
| DINO | ✓ | 79 | 58 | 71 | 61 | 65 | 53 | 53 | 43 | 47 | 58 | 63 | 39 | 19 | 26 | 66 | 19 | 42 | 14 | 9 | 9 | 24 | 45 | 57 | 67 | 63 | 2 |
| MAE (avg) | ✓ | 78 | 58 | 73 | 61 | 66 | 57 | 54 | 44 | 48 | 64 | 63 | 40 | 18 | 28 | 75 | 22 | 46 | 15 | 10 | 10 | 26 | 49 | 63 | 71 | 64 | 3 |

Table 8: **Adjusted Rand index (ARI; %), averaged over clusterers.** The ARI reported is averaged over K-Means, Spectral, AC w/ C, AC w/o C, AP, and HDBSCAN results (except iNat, which excludes Spectral). The magnitude of the values differs from the AMI reported in Table 7, but the trends across encoders is the same.

| Encoder | FT | In-domain | | | | | Domain-shift | | | | Near-OOD | | | Fine-grained | | | | | | Far-OOD | | | | | | | |
|---|---|---|---|---|---|---|---|---|---|---|---|---|---|---|---|---|---|---|---|---|---|---|---|---|---|---|---|
| | | IN1k | INv2 | C10 | C100 | IN9 | 9-FG | 9-MR | IN-R | IN-S | IN-O | LSU | P365 | Air | Cars | F102 | Bio | Birds | iNat | CelA | UTKF | BHis | DTD | ESAT | MNST | Fash | SVHN |
| Raw image | | 0 | 0 | 3 | 1 | 1 | 2 | 0 | 0 | 1 | 1 | 2 | 0 | 1 | 0 | 3 | 1 | 0 | 0 | 0 | 1 | 3 | 1 | 9 | 35 | 26 | 0 |
| **RN50** — Rand. | | 0 | 0 | 1 | 1 | 1 | 2 | 0 | 0 | 1 | 1 | 1 | 0 | 0 | 0 | 2 | 1 | 0 | 0 | 0 | 0 | 3 | 1 | 8 | 18 | 16 | −0 |
| X-Ent. | | 33 | 19 | 39 | 21 | 42 | 46 | 35 | 12 | 11 | 35 | 41 | 11 | 4 | 6 | 36 | 5 | 12 | 2 | 2 | 3 | 6 | 23 | 49 | 54 | 43 | 2 |
| MoCo-v3 | | 15 | 8 | 39 | 18 | 50 | 38 | 23 | 6 | 8 | 16 | 34 | 7 | 5 | 2 | 51 | 6 | 6 | 1 | 2 | 3 | 9 | 23 | 58 | 74 | 48 | 4 |
| DINO | | 11 | 7 | 25 | 13 | 49 | 41 | 19 | 3 | 4 | 16 | 31 | 7 | 4 | 2 | 51 | 9 | 3 | 1 | 2 | 5 | 16 | 26 | 65 | 56 | 43 | 1 |
| VICReg | | 11 | 7 | 28 | 15 | 47 | 40 | 19 | 4 | 5 | 16 | 33 | 7 | 3 | 2 | 53 | 9 | 3 | 1 | 2 | 5 | 12 | 26 | 67 | 65 | 48 | 1 |
| MoCo-v3 | ✓ | 32 | 19 | 41 | 21 | 55 | 49 | 44 | 12 | 12 | 41 | 43 | 10 | 5 | 6 | 33 | 4 | 12 | 1 | 1 | 3 | 5 | 26 | 41 | 39 | 44 | 2 |
| DINO | ✓ | 31 | 18 | 38 | 21 | 53 | 45 | 39 | 11 | 11 | 43 | 36 | 10 | 4 | 6 | 34 | 4 | 11 | 1 | 2 | 3 | 5 | 24 | 44 | 51 | 41 | 2 |
| VICReg | ✓ | 30 | 18 | 37 | 21 | 53 | 43 | 42 | 10 | 10 | 40 | 40 | 10 | 4 | 6 | 37 | 4 | 11 | 1 | 2 | 3 | 5 | 27 | 44 | 41 | 47 | 2 |
| **ViT-B** — Rand. | | 0 | 0 | 3 | 1 | 2 | 1 | 0 | 0 | 0 | 1 | 3 | 0 | 0 | 0 | 4 | 2 | 0 | 0 | 0 | 1 | 2 | 2 | 10 | 4 | 9 | 0 |
| X-Ent. | | 41 | 25 | 65 | 33 | 27 | 33 | 24 | 13 | 12 | 46 | 48 | 10 | 5 | 6 | 42 | 5 | 11 | 1 | 1 | 3 | 6 | 23 | 43 | 58 | 45 | 0 |
| MoCo-v3 | | 19 | 13 | 58 | 29 | 37 | 36 | 18 | 6 | 8 | 15 | 45 | 10 | 3 | 2 | 54 | 9 | 6 | 1 | 2 | 5 | 11 | 31 | 55 | 61 | 44 | 1 |
| DINO | | 23 | 15 | 46 | 28 | 47 | 42 | 31 | 8 | 10 | 22 | 49 | 11 | 5 | 3 | 72 | 11 | 11 | 1 | 2 | 4 | 12 | 34 | 61 | 57 | 44 | 0 |
| MAE (CLS) | | 4 | 3 | 13 | 6 | 19 | 18 | 8 | 1 | 3 | 6 | 14 | 3 | 2 | 1 | 20 | 4 | 2 | 0 | 1 | 2 | 8 | 11 | 29 | 33 | 27 | 0 |
| MAE (avg) | | 5 | 3 | 14 | 7 | 22 | 19 | 6 | 1 | 3 | 6 | 20 | 3 | 1 | 1 | 29 | 2 | 1 | 0 | 1 | 3 | 9 | 14 | 33 | 39 | 31 | 0 |
| MoCo-v3 | ✓ | 42 | 27 | 67 | 39 | 30 | 17 | 25 | 17 | 15 | 39 | 51 | 12 | 5 | 7 | 45 | 5 | 14 | 2 | 2 | 3 | 6 | 25 | 38 | 53 | 45 | 1 |
| DINO | ✓ | 45 | 28 | 59 | 32 | 32 | 19 | 27 | 17 | 16 | 35 | 48 | 11 | 5 | 7 | 41 | 5 | 13 | 2 | 2 | 3 | 6 | 23 | 39 | 48 | 45 | 1 |
| MAE (avg) | ✓ | 44 | 28 | 62 | 32 | 35 | 21 | 26 | 18 | 16 | 42 | 49 | 12 | 4 | 8 | 50 | 6 | 14 | 2 | 2 | 3 | 7 | 25 | 45 | 56 | 45 | 1 |

Table 9: **AMI score (%) using K-Means.**

| Encoder | FT | In-domain | | | | | Domain-shift | | | | Near-OOD | | | Fine-grained | | | | | | Far-OOD | | | | | | | |
|---|---|---|---|---|---|---|---|---|---|---|---|---|---|---|---|---|---|---|---|---|---|---|---|---|---|---|---|
| | | IN1k | INv2 | C10 | C100 | IN9 | 9-FG | 9-MR | IN-R | IN-S | IN-O | LSU | P365 | Air | Cars | F102 | Bio | Birds | iNat | CelA | UTKF | BHis | DTD | ESAT | MNST | Fash | SVHN |
| Raw image | | 2 | 1 | 8 | 13 | 5 | 5 | 1 | 4 | 11 | 5 | 5 | 4 | 4 | 3 | 19 | 7 | 3 | 0 | 4 | 3 | 16 | 6 | 25 | 44 | 50 | 0 |
| **RN50** — Rand. | | 2 | 1 | 2 | 9 | 2 | 6 | 2 | 3 | 11 | 4 | 3 | 3 | 3 | 1 | 15 | 9 | 2 | 0 | 2 | 2 | 16 | 5 | 23 | 34 | 41 | 0 |
| X-Ent. | | 73 | 48 | 68 | 51 | 84 | 80 | 72 | 34 | 42 | 63 | 63 | 39 | 15 | 23 | 64 | 20 | 39 | 9 | 8 | 11 | 28 | 48 | 76 | 81 | 69 | 5 |
| MoCo-v3 | | 48 | 25 | 64 | 51 | 78 | 68 | 49 | 27 | 38 | 41 | 57 | 33 | 21 | 12 | 80 | 23 | 28 | 4 | 10 | 12 | 33 | 52 | 70 | 86 | 71 | 11 |
| DINO | | 44 | 22 | 49 | 42 | 78 | 70 | 47 | 18 | 29 | 37 | 57 | 35 | 18 | 13 | 82 | 27 | 18 | 4 | 9 | 12 | 44 | 52 | 79 | 74 | 64 | 1 |
| VICReg | | 46 | 24 | 53 | 45 | 81 | 71 | 45 | 21 | 33 | 38 | 55 | 33 | 16 | 12 | 81 | 26 | 18 | 4 | 10 | 12 | 38 | 52 | 80 | 80 | 70 | 3 |
| MoCo-v3 | ✓ | 73 | 49 | 66 | 51 | 88 | 80 | 77 | 36 | 44 | 72 | 65 | 38 | 17 | 22 | 61 | 17 | 39 | 8 | 7 | 11 | 23 | 50 | 69 | 65 | 71 | 7 |
| DINO | ✓ | 72 | 49 | 67 | 52 | 86 | 75 | 75 | 34 | 42 | 74 | 63 | 38 | 15 | 22 | 60 | 16 | 39 | 8 | 7 | 11 | 24 | 49 | 71 | 76 | 70 | 5 |
| VICReg | ✓ | 71 | 48 | 66 | 51 | 86 | 73 | 73 | 33 | 41 | 71 | 62 | 38 | 15 | 22 | 63 | 17 | 38 | 8 | 7 | 11 | 23 | 51 | 68 | 71 | 71 | 5 |
| **ViT-B** — Rand. | | 2 | 1 | 9 | 10 | 6 | 3 | 1 | 3 | 11 | 5 | 8 | 5 | 3 | 2 | 19 | 14 | 3 | 0 | 2 | 3 | 17 | 8 | 25 | 13 | 22 | 0 |
| X-Ent. | | 79 | 59 | 83 | 65 | 64 | 73 | 61 | 39 | 45 | 71 | 67 | 39 | 18 | 25 | 68 | 21 | 38 | 8 | 7 | 10 | 26 | 45 | 65 | 80 | 70 | 1 |
| MoCo-v3 | | 60 | 36 | 79 | 62 | 79 | 69 | 45 | 26 | 38 | 37 | 64 | 39 | 15 | 12 | 81 | 28 | 27 | 6 | 9 | 12 | 40 | 55 | 79 | 83 | 71 | 1 |
| DINO | | 67 | 44 | 77 | 62 | 88 | 81 | 68 | 34 | 43 | 45 | 66 | 41 | 21 | 13 | 89 | 31 | 44 | 9 | 10 | 11 | 43 | 60 | 84 | 81 | 69 | 1 |
| MAE (CLS) | | 21 | 9 | 29 | 29 | 42 | 38 | 17 | 11 | 25 | 17 | 35 | 21 | 10 | 5 | 47 | 15 | 10 | 1 | 6 | 7 | 31 | 31 | 53 | 45 | 55 | 1 |
| MAE (avg) | | 22 | 9 | 32 | 29 | 41 | 31 | 1 | 12 | 23 | 19 | 44 | 23 | 8 | 6 | 53 | 12 | 9 | 1 | 7 | 8 | 30 | 37 | 49 | 42 | 55 | 1 |
| MoCo-v3 | ✓ | 81 | 61 | 81 | 69 | 60 | 48 | 47 | 44 | 49 | 66 | 67 | 41 | 19 | 26 | 72 | 20 | 43 | 10 | 8 | 11 | 24 | 49 | 63 | 78 | 71 | 1 |
| DINO | ✓ | 82 | 63 | 80 | 62 | 71 | 47 | 53 | 43 | 49 | 62 | 68 | 40 | 18 | 26 | 67 | 18 | 41 | 10 | 8 | 11 | 25 | 45 | 63 | 78 | 70 | 2 |
| MAE (avg) | ✓ | 82 | 63 | 79 | 63 | 76 | 57 | 55 | 45 | 51 | 68 | 67 | 41 | 18 | 28 | 77 | 21 | 44 | 10 | 9 | 11 | 27 | 48 | 68 | 82 | 71 | 2 |

Table 10: **AMI score (%) using Spectral Clustering.**

| Encoder | FT | In-domain | | | | | Domain-shift | | | | Near-OOD | | | Fine-grained | | | | | | Far-OOD | | | | | | | |
|---|---|---|---|---|---|---|---|---|---|---|---|---|---|---|---|---|---|---|---|---|---|---|---|---|---|---|---|
| | | IN1k | INv2 | C10 | C100 | IN9 | 9-FG | 9-MR | IN-R | IN-S | IN-O | LSU | P365 | Air | Cars | F102 | Bio | Birds | iNat | CelA | UTKF | BHis | DTD | ESAT | MNST | Fash | SVHN |
| Raw image | | 2 | 1 | 9 | 12 | 6 | 6 | 1 | 4 | 11 | 4 | 8 | 5 | 5 | 2 | 18 | 6 | 3 | – | 4 | 3 | 15 | 5 | 27 | 70 | 60 | 1 |
| **RN50** — Rand. | | 2 | 0 | 3 | 10 | 3 | 7 | 1 | 3 | 10 | 4 | 3 | 4 | 3 | 2 | 16 | 7 | 2 | – | 2 | 2 | 17 | 6 | 25 | 53 | 50 | 0 |
| X-Ent. | | 73 | 48 | 64 | 52 | 59 | 75 | 53 | 37 | 42 | 64 | 63 | 39 | 16 | 21 | 62 | 18 | 40 | – | 9 | 11 | 25 | 48 | 66 | 70 | 67 | 5 |
| MoCo-v3 | | 51 | 28 | 61 | 51 | 65 | 63 | 48 | 30 | 39 | 40 | 55 | 35 | 20 | 13 | 75 | 21 | 29 | – | 11 | 11 | 30 | 51 | 66 | 83 | 69 | 12 |
| DINO | | 47 | 25 | 47 | 45 | 71 | 68 | 47 | 23 | 33 | 39 | 54 | 37 | 20 | 14 | 76 | 24 | 23 | – | 10 | 10 | 39 | 52 | 72 | 74 | 66 | 2 |
| VICReg | | 48 | 26 | 53 | 46 | 63 | 59 | 40 | 24 | 35 | 41 | 54 | 35 | 17 | 12 | 77 | 23 | 22 | – | 12 | 11 | 34 | 53 | 73 | 75 | 68 | 5 |
| MoCo-v3 | ✓ | 72 | 49 | 67 | 52 | 84 | 75 | 62 | 37 | 43 | 71 | 62 | 38 | 15 | 21 | 58 | 15 | 40 | – | 8 | 11 | 23 | 50 | 59 | 54 | 69 | 7 |
| DINO | ✓ | 72 | 49 | 65 | 54 | 76 | 69 | 60 | 37 | 42 | 72 | 58 | 38 | 14 | 20 | 58 | 15 | 38 | – | 7 | 9 | 22 | 47 | 61 | 67 | 69 | 6 |
| VICReg | ✓ | 71 | 48 | 64 | 51 | 79 | 71 | 67 | 35 | 41 | 70 | 59 | 38 | 16 | 22 | 61 | 16 | 39 | – | 8 | 11 | 22 | 51 | 59 | 57 | 68 | 5 |
| **ViT-B** — Rand. | | 2 | 1 | 10 | 10 | 6 | 4 | 1 | 4 | 10 | 5 | 9 | 5 | 3 | 2 | 20 | 11 | 2 | – | 2 | 3 | 19 | 9 | 30 | 19 | 25 | 2 |
| X-Ent. | | 79 | 56 | 79 | 64 | 44 | 47 | 36 | 39 | 42 | 70 | 63 | 39 | 18 | 22 | 65 | 17 | 38 | – | 8 | 10 | 24 | 49 | 70 | 68 | 65 | 2 |
| MoCo-v3 | | 61 | 38 | 77 | 62 | 67 | 72 | 51 | 29 | 39 | 38 | 61 | 40 | 14 | 11 | 73 | 24 | 27 | – | 10 | 12 | 33 | 55 | 75 | 70 | 69 | 3 |
| DINO | | 67 | 46 | 69 | 62 | 59 | 61 | 54 | 38 | 44 | 48 | 63 | 42 | 23 | 16 | 88 | 28 | 46 | – | 13 | 11 | 37 | 57 | 74 | 76 | 68 | 1 |
| MAE (CLS) | | 27 | 12 | 36 | 34 | 51 | 56 | 28 | 14 | 28 | 19 | 36 | 24 | 11 | 5 | 55 | 15 | 12 | – | 7 | 7 | 34 | 36 | 57 | 81 | 66 | 0 |
| MAE (avg) | | 27 | 12 | 34 | 34 | 54 | 49 | 25 | 14 | 25 | 20 | 45 | 25 | 8 | 6 | 59 | 12 | 9 | – | 8 | 8 | 31 | 40 | 58 | 68 | 64 | 1 |
| MoCo-v3 | ✓ | 81 | 59 | 79 | 68 | 66 | 47 | 45 | 44 | 51 | 60 | 64 | 40 | 18 | 25 | 67 | 19 | 45 | – | 9 | 11 | 23 | 50 | 56 | 66 | 70 | 2 |
| DINO | ✓ | 82 | 60 | 73 | 63 | 63 | 47 | 48 | 44 | 50 | 58 | 65 | 40 | 17 | 25 | 65 | 17 | 43 | – | 9 | 11 | 24 | 48 | 57 | 60 | 65 | 1 |
| MAE (avg) | ✓ | 82 | 58 | 77 | 63 | 65 | 49 | 43 | 43 | 48 | 62 | 68 | 41 | 18 | 26 | 73 | 18 | 44 | – | 10 | 11 | 25 | 52 | 69 | 58 | 69 | 3 |

Table 11: **AMI score (%) using Agglomerative Clustering with number of clusters given (AC w/ C).** In this configuration, the target number of clusters is provided to AC (set to the number of classes in the GT annotations) and the distance threshold is automatically selected to split the hierarchy into the target number of clusters.

| Encoder | FT | In-domain | | | | | Domain-shift | | | | Near-OOD | | | Fine-grained | | | | | | Far-OOD | | | | | | | |
|---|---|---|---|---|---|---|---|---|---|---|---|---|---|---|---|---|---|---|---|---|---|---|---|---|---|---|---|
| | | IN1k | INv2 | C10 | C100 | IN9 | 9-FG | 9-MR | IN-R | IN-S | IN-O | LSU | P365 | Air | Cars | F102 | Bio | Birds | iNat | CelA | UTKF | BHis | DTD | ESAT | MNST | Fash | SVHN |
| Raw image | | 4 | 3 | 7 | 8 | 3 | 4 | 0 | 1 | 7 | 8 | 6 | 5 | 4 | 2 | 15 | 11 | 3 | 2 | 7 | 3 | 9 | 3 | 15 | 51 | 45 | 0 |
| **RN50** — Rand. | | 2 | 1 | 2 | 6 | 3 | 6 | 1 | 2 | 9 | 4 | 3 | 4 | 3 | 2 | 13 | 11 | 3 | 0 | 3 | 1 | 9 | 3 | 21 | 23 | 24 | 0 |
| X-Ent. | | 73 | 49 | 67 | 52 | 83 | 81 | 70 | 34 | 43 | 65 | 64 | 39 | 15 | 23 | 64 | 20 | 39 | 9 | 8 | 11 | 28 | 48 | 75 | 82 | 69 | 4 |
| MoCo-v3 | | 49 | 26 | 64 | 51 | 78 | 66 | 54 | 27 | 39 | 41 | 57 | 33 | 20 | 12 | 81 | 24 | 28 | 5 | 10 | 12 | 33 | 52 | 72 | 87 | 70 | 10 |
| DINO | | 48 | 27 | 48 | 42 | 76 | 67 | 47 | 18 | 31 | 38 | 57 | 38 | 19 | 15 | 82 | 34 | 21 | 7 | 12 | 13 | 46 | 52 | 78 | 74 | 67 | 1 |
| VICReg | | 49 | 28 | 53 | 46 | 79 | 69 | 38 | 20 | 34 | 40 | 55 | 35 | 16 | 13 | 82 | 33 | 23 | 7 | 13 | 13 | 40 | 52 | 81 | 81 | 69 | 2 |
| MoCo-v3 | ✓ | 73 | 50 | 67 | 52 | 88 | 80 | 79 | 35 | 45 | 72 | 63 | 38 | 16 | 23 | 61 | 17 | 40 | 8 | 7 | 11 | 23 | 50 | 66 | 66 | 70 | 7 |
| DINO | ✓ | 72 | 49 | 66 | 52 | 86 | 80 | 73 | 34 | 43 | 74 | 63 | 38 | 16 | 23 | 61 | 16 | 39 | 8 | 7 | 11 | 23 | 48 | 68 | 76 | 69 | 5 |
| VICReg | ✓ | 71 | 48 | 65 | 50 | 83 | 74 | 76 | 33 | 42 | 72 | 63 | 38 | 16 | 22 | 63 | 17 | 39 | 8 | 8 | 12 | 23 | 51 | 68 | 69 | 71 | 5 |
| **ViT-B** — Rand. | | 3 | 1 | 9 | 9 | 5 | 2 | 1 | 2 | 7 | 6 | 8 | 6 | 3 | 3 | 19 | 13 | 4 | 0 | 3 | 2 | 13 | 8 | 18 | 9 | 20 | 0 |
| X-Ent. | | 79 | 59 | 83 | 66 | 73 | 70 | 58 | 39 | 46 | 72 | 67 | 39 | 18 | 25 | 68 | 21 | 39 | 9 | 8 | 11 | 26 | 46 | 67 | 84 | 71 | 2 |
| MoCo-v3 | | 61 | 37 | 80 | 62 | 30 | 53 | 41 | 25 | 39 | 38 | 62 | 40 | 15 | 12 | 81 | 35 | 31 | 10 | 12 | 14 | 40 | 56 | 79 | 84 | 73 | 1 |
| DINO | | 68 | 46 | 75 | 62 | 78 | 78 | 72 | 33 | 45 | 46 | 67 | 44 | 22 | 14 | 90 | 38 | 47 | 15 | 14 | 12 | 44 | 60 | 84 | 83 | 69 | 1 |
| MAE (CLS) | | 28 | 15 | 29 | 26 | 38 | 44 | 20 | 10 | 28 | 23 | 37 | 24 | 11 | 6 | 49 | 23 | 18 | 5 | 9 | 8 | 27 | 34 | 52 | 60 | 51 | 1 |
| MAE (avg) | | 29 | 14 | 30 | 29 | 42 | 42 | 6 | 10 | 26 | 23 | 36 | 26 | 8 | 6 | 55 | 16 | 16 | 4 | 11 | 9 | 25 | 39 | 47 | 44 | 55 | 1 |
| MoCo-v3 | ✓ | 81 | 62 | 84 | 69 | 70 | 53 | 61 | 44 | 50 | 65 | 67 | 41 | 20 | 27 | 72 | 20 | 43 | 10 | 9 | 11 | 25 | 48 | 65 | 79 | 71 | 1 |
| DINO | ✓ | 82 | 63 | 80 | 62 | 69 | 54 | 60 | 43 | 50 | 63 | 69 | 40 | 18 | 26 | 67 | 18 | 41 | 10 | 8 | 11 | 25 | 46 | 64 | 80 | 70 | 2 |
| MAE (avg) | ✓ | 82 | 63 | 78 | 63 | 68 | 59 | 63 | 45 | 52 | 69 | 69 | 41 | 19 | 28 | 77 | 22 | 45 | 11 | 9 | 11 | 26 | 50 | 71 | 82 | 71 | 2 |

Table 12: **AMI score (%) using Agglomerative Clustering with number of clusters unknown (AC w/o C).** In this configuration, the number of clusters is determined automatically using a distance threshold tuned on a subset of IN-1k training data (see Appendix E for details).

| Encoder | FT | In-domain | | | | | Domain-shift | | | | Near-OOD | | | Fine-grained | | | | | | Far-OOD | | | | | | | |
|---|---|---|---|---|---|---|---|---|---|---|---|---|---|---|---|---|---|---|---|---|---|---|---|---|---|---|---|
| | | IN1k | INv2 | C10 | C100 | IN9 | 9-FG | 9-MR | IN-R | IN-S | IN-O | LSU | P365 | Air | Cars | F102 | Bio | Birds | iNat | CelA | UTKF | BHis | DTD | ESAT | MNST | Fash | SVHN |
| Raw image | | 4 | 3 | 11 | 10 | 5 | 6 | 1 | 2 | 7 | 5 | 4 | 5 | 4 | 3 | 17 | 12 | 3 | 3 | 6 | 3 | 11 | 4 | 16 | 54 | 54 | 2 |
| **RN50** — Rand. | | 3 | 2 | 4 | 6 | 4 | 6 | 2 | 2 | 7 | 5 | 4 | 4 | 3 | 2 | 12 | 13 | 3 | 1 | 3 | 1 | 15 | 4 | 21 | 37 | 42 | 1 |
| X-Ent. | | 68 | 44 | 54 | 52 | 68 | 67 | 59 | 35 | 42 | 58 | 62 | 40 | 17 | 26 | 64 | 29 | 46 | 17 | 10 | 12 | 29 | 48 | 60 | 59 | 58 | 8 |
| MoCo-v3 | | 47 | 30 | 64 | 47 | 83 | 67 | 55 | 24 | 35 | 30 | 52 | 37 | 14 | 12 | 65 | 35 | 33 | 14 | 18 | 14 | 31 | 41 | 73 | 85 | 70 | 14 |
| DINO | | 47 | 30 | 45 | 42 | 75 | 66 | 43 | 18 | 28 | 37 | 46 | 39 | 9 | 15 | 72 | 41 | 25 | 14 | 15 | 14 | 47 | 52 | 76 | 70 | 65 | 5 |
| VICReg | | 45 | 30 | 47 | 46 | 76 | 71 | 43 | 20 | 31 | 39 | 45 | 37 | 8 | 13 | 73 | 39 | 26 | 14 | 15 | 13 | 38 | 52 | 78 | 77 | 69 | 7 |
| MoCo-v3 | ✓ | 68 | 45 | 54 | 52 | 73 | 67 | 63 | 35 | 45 | 62 | 60 | 39 | 19 | 26 | 61 | 23 | 47 | 16 | 10 | 12 | 23 | 50 | 56 | 51 | 58 | 8 |
| DINO | ✓ | 68 | 45 | 54 | 51 | 72 | 68 | 63 | 34 | 43 | 63 | 57 | 39 | 17 | 25 | 61 | 23 | 46 | 15 | 10 | 11 | 23 | 48 | 57 | 57 | 56 | 6 |
| VICReg | ✓ | 67 | 44 | 52 | 50 | 72 | 67 | 63 | 33 | 42 | 61 | 59 | 39 | 18 | 24 | 64 | 24 | 45 | 15 | 11 | 12 | 23 | 51 | 56 | 53 | 58 | 7 |
| **ViT-B** — Rand. | | 5 | 3 | 9 | 9 | 6 | 3 | 1 | 3 | 3 | 8 | 9 | 7 | 3 | 3 | 18 | 13 | 5 | 3 | 4 | 3 | 7 | 7 | 18 | 9 | 17 | 0 |
| X-Ent. | | 72 | 50 | 68 | 66 | 66 | 64 | 52 | 39 | 45 | 64 | 62 | 41 | 20 | 28 | 69 | 29 | 44 | 17 | 11 | 11 | 27 | 46 | 57 | 66 | 60 | 4 |
| MoCo-v3 | | 55 | 37 | 67 | 62 | 73 | 65 | 47 | 26 | 38 | 37 | 62 | 38 | 12 | 10 | 78 | 39 | 33 | 15 | 11 | 13 | 36 | 56 | 63 | 67 | 68 | 4 |
| DINO | | 63 | 43 | 56 | 59 | 74 | 65 | 59 | 34 | 45 | 46 | 63 | 42 | 19 | 13 | 90 | 43 | 47 | 20 | 12 | 10 | 37 | 58 | 65 | 60 | 61 | 3 |
| MAE (CLS) | | 28 | 16 | 30 | 27 | 49 | 48 | 25 | 11 | 27 | 22 | 33 | 24 | 10 | 6 | 51 | 21 | 19 | 9 | 9 | 7 | 32 | 34 | 53 | 62 | 53 | 2 |
| MAE (avg) | | 28 | 15 | 30 | 30 | 44 | 43 | 23 | 12 | 24 | 23 | 34 | 24 | 8 | 6 | 59 | 17 | 17 | 8 | 12 | 9 | 30 | 42 | 53 | 52 | 56 | 2 |
| MoCo-v3 | ✓ | 73 | 53 | 70 | 69 | 67 | 61 | 55 | 44 | 48 | 59 | 61 | 42 | 21 | 29 | 71 | 29 | 49 | 19 | 12 | 12 | 26 | 48 | 55 | 61 | 59 | 3 |
| DINO | ✓ | 75 | 53 | 63 | 62 | 68 | 62 | 56 | 43 | 49 | 56 | 63 | 41 | 21 | 29 | 67 | 26 | 47 | 19 | 11 | 11 | 26 | 45 | 54 | 59 | 57 | 3 |
| MAE (avg) | ✓ | 74 | 54 | 64 | 63 | 68 | 63 | 57 | 45 | 50 | 62 | 65 | 42 | 20 | 31 | 76 | 31 | 51 | 20 | 13 | 12 | 29 | 49 | 58 | 63 | 61 | 5 |

Table 13: **AMI score (%) using Affinity Propagation.**

| Encoder | FT | In-domain | | | | | Domain-shift | | | | Near-OOD | | | Fine-grained | | | | | | Far-OOD | | | | | | | |
| | | IN1k | INv2 | C10 | C100 | IN9 | 9-FG | 9-MR | IN-R | IN-S | IN-O | LSU | P365 | Air | Cars | F102 | Bio | Birds | iNat | CelA | UTKF | BHis | DTD | ESAT | MNST | Fash | SVHN |
|---|---|---|---|---|---|---|---|---|---|---|---|---|---|---|---|---|---|---|---|---|---|---|---|---|---|---|---|
| Raw image | | 2 | 1 | 11 | 10 | 6 | 6 | 1 | 3 | 10 | 6 | 7 | 3 | 3 | 2 | 16 | 8 | 2 | 0 | 4 | 3 | 16 | 5 | 26 | 43 | 43 | 4 |
| **RN50** — Rand. | | 2 | 1 | 7 | 7 | 5 | 7 | 2 | 2 | 10 | 5 | 5 | 3 | 2 | 1 | 14 | 11 | 2 | 0 | 3 | 2 | 18 | 5 | 24 | 40 | 40 | 1 |
| X-Ent. | | 67 | 44 | 54 | 52 | 70 | 71 | 60 | 34 | 42 | 56 | 57 | 42 | 18 | 27 | 64 | 31 | 48 | 20 | 13 | 12 | 28 | 49 | 63 | 63 | 65 | 7 |
| MoCo-v3 | | 50 | 30 | 54 | 51 | 68 | 60 | 46 | 27 | 38 | 39 | 53 | 35 | 22 | 12 | 80 | 33 | 33 | 11 | 14 | 13 | 32 | 52 | 64 | 76 | 61 | 15 |
| DINO | | 48 | 28 | 43 | 42 | 68 | 63 | 41 | 19 | 29 | 36 | 53 | 37 | 19 | 14 | 80 | 40 | 24 | 10 | 14 | 13 | 45 | 52 | 70 | 62 | 59 | 4 |
| VICReg | | 49 | 29 | 45 | 45 | 70 | 64 | 42 | 21 | 32 | 37 | 54 | 35 | 16 | 13 | 80 | 37 | 25 | 10 | 14 | 13 | 38 | 52 | 70 | 66 | 55 | 6 |
| MoCo-v3 | ✓ | 65 | 44 | 57 | 52 | 79 | 70 | 66 | 35 | 44 | 61 | 58 | 42 | 19 | 27 | 61 | 26 | 48 | 19 | 12 | 12 | 23 | 50 | 60 | 51 | 65 | 8 |
| DINO | ✓ | 66 | 44 | 55 | 52 | 79 | 73 | 64 | 34 | 43 | 61 | 58 | 41 | 18 | 26 | 61 | 26 | 47 | 18 | 13 | 12 | 24 | 48 | 60 | 61 | 60 | 6 |
| VICReg | ✓ | 65 | 44 | 54 | 51 | 79 | 71 | 64 | 33 | 41 | 61 | 57 | 41 | 18 | 27 | 63 | 27 | 47 | 18 | 13 | 13 | 24 | 51 | 60 | 58 | 64 | 7 |
| **ViT-B** — Rand. | | 3 | 1 | 9 | 9 | 6 | 5 | 1 | 2 | 10 | 7 | 9 | 4 | 3 | 2 | 18 | 17 | 3 | 0 | 3 | 3 | 18 | 8 | 31 | 20 | 20 | 0 |
| X-Ent. | | 71 | 50 | 72 | 65 | 66 | 64 | 53 | 39 | 44 | 64 | 62 | 42 | 20 | 30 | 68 | 31 | 45 | 19 | 13 | 11 | 26 | 45 | 59 | 71 | 68 | 4 |
| MoCo-v3 | | 57 | 36 | 68 | 63 | 75 | 68 | 46 | 26 | 37 | 36 | 63 | 42 | 15 | 12 | 78 | 41 | 34 | 14 | 15 | 13 | 41 | 55 | 69 | 72 | 20 | 4 |
| DINO | | 62 | 41 | 63 | 62 | 84 | 75 | 66 | 33 | 43 | 44 | 62 | 44 | 22 | 15 | 87 | 44 | 45 | 19 | 16 | 12 | 44 | 60 | 74 | 67 | 65 | 2 |
| MAE (CLS) | | 18 | 10 | 26 | 23 | 40 | 38 | 19 | 9 | 24 | 18 | 28 | 15 | 10 | 4 | 45 | 17 | 9 | 2 | 6 | 5 | 33 | 30 | 48 | 47 | 43 | 6 |
| MAE (avg) | | 23 | 12 | 30 | 25 | 47 | 45 | 17 | 9 | 23 | 18 | 35 | 23 | 8 | 6 | 52 | 19 | 13 | 3 | 8 | 9 | 36 | 35 | 58 | 64 | 11 | 1 |
| MoCo-v3 | ✓ | 73 | 52 | 74 | 68 | 68 | 61 | 55 | 44 | 48 | 59 | 61 | 44 | 21 | 32 | 70 | 32 | 49 | 22 | 14 | 12 | 26 | 50 | 57 | 64 | 67 | 3 |
| DINO | ✓ | 74 | 52 | 68 | 62 | 70 | 61 | 56 | 43 | 48 | 56 | 63 | 43 | 21 | 33 | 67 | 29 | 48 | 21 | 14 | 11 | 26 | 45 | 57 | 63 | 65 | 3 |
| MAE (avg) | ✓ | 73 | 53 | 68 | 62 | 70 | 63 | 57 | 45 | 50 | 62 | 63 | 44 | 19 | 34 | 76 | 33 | 51 | 23 | 15 | 12 | 28 | 49 | 63 | 68 | 64 | 4 |

Table 14: **AMI score (%) using HDBSCAN.** In this analysis, we evaluate the HDBSCAN output by counting all the samples labelled as "noise" as being combined together into their own cluster. Such analysis is contrary to the intended usage of HDBSCAN and will negatively impact the measured performance of HDBSCAN, but is the fairest comparison available. For an evaluation of HDBSCAN excluding rejected samples, see Table 15.

| Encoder | FT | In-domain | | | | | Domain-shift | | | | Near-OOD | | | Fine-grained | | | | | | Far-OOD | | | | | | | |
| | | IN1k | INv2 | C10 | C100 | IN9 | 9-FG | 9-MR | IN-R | IN-S | IN-O | LSU | P365 | Air | Cars | F102 | Bio | Birds | iNat | CelA | UTKF | BHis | DTD | ESAT | MNST | Fash | SVHN |
|---|---|---|---|---|---|---|---|---|---|---|---|---|---|---|---|---|---|---|---|---|---|---|---|---|---|---|---|
| Raw image | | 1 | 1 | 7 | 6 | 6 | 5 | 1 | 1 | 6 | 7 | 5 | 1 | 3 | 1 | 9 | 1 | 1 | 0 | 1 | 1 | 12 | 4 | 21 | 65 | 38 | 3 |
| **RN50** — Rand. | | 0 | 0 | 4 | 4 | 3 | 5 | 1 | 1 | 5 | 2 | 3 | 1 | 2 | 0 | 8 | 2 | 1 | 0 | 0 | 0 | 11 | 3 | 17 | 36 | 27 | 1 |
| X-Ent. | | 64 | 38 | 37 | 43 | 50 | 48 | 47 | 32 | 30 | 52 | 39 | 25 | 10 | 13 | 56 | 6 | 28 | 8 | 3 | 3 | 20 | 42 | 58 | 70 | 49 | 6 |
| MoCo-v3 | | 34 | 18 | 37 | 38 | 46 | 42 | 35 | 22 | 25 | 30 | 39 | 20 | 14 | 7 | 76 | 8 | 26 | 5 | 4 | 2 | 23 | 47 | 64 | 77 | 45 | 11 |
| DINO | | 29 | 15 | 28 | 28 | 49 | 47 | 33 | 13 | 19 | 26 | 34 | 20 | 13 | 7 | 78 | 9 | 16 | 4 | 5 | 2 | 37 | 43 | 72 | 68 | 44 | 4 |
| VICReg | | 32 | 18 | 27 | 32 | 42 | 45 | 32 | 16 | 22 | 29 | 37 | 19 | 12 | 8 | 77 | 9 | 13 | 4 | 4 | 3 | 28 | 48 | 72 | 72 | 51 | 5 |
| MoCo-v3 | ✓ | 61 | 39 | 43 | 42 | 50 | 49 | 39 | 31 | 30 | 62 | 51 | 24 | 15 | 14 | 51 | 5 | 27 | 7 | 2 | 3 | 16 | 45 | 52 | 59 | 44 | 5 |
| DINO | ✓ | 61 | 38 | 35 | 39 | 49 | 46 | 38 | 28 | 30 | 63 | 36 | 23 | 16 | 12 | 54 | 5 | 29 | 6 | 3 | 3 | 17 | 45 | 53 | 66 | 47 | 4 |
| VICReg | ✓ | 60 | 37 | 38 | 40 | 50 | 45 | 44 | 28 | 29 | 61 | 43 | 22 | 10 | 13 | 58 | 5 | 31 | 6 | 3 | 2 | 15 | 45 | 57 | 54 | 52 | 5 |
| **ViT-B** — Rand. | | 1 | 0 | 7 | 5 | 5 | 4 | 1 | 1 | 6 | 5 | 7 | 1 | 2 | 1 | 11 | 4 | 1 | 0 | 1 | 1 | 15 | 6 | 26 | 15 | 18 | 0 |
| X-Ent. | | 72 | 51 | 70 | 54 | 51 | 49 | 43 | 37 | 34 | 64 | 53 | 30 | 14 | 13 | 64 | 7 | 28 | 9 | 2 | 2 | 19 | 41 | 46 | 71 | 46 | 3 |
| MoCo-v3 | | 49 | 27 | 62 | 50 | 49 | 48 | 35 | 23 | 27 | 28 | 46 | 26 | 11 | 6 | 75 | 10 | 22 | 7 | 4 | 3 | 32 | 52 | 56 | 76 | 48 | 3 |
| DINO | | 56 | 34 | 56 | 51 | 52 | 50 | 49 | 28 | 31 | 36 | 59 | 26 | 15 | 8 | 84 | 11 | 34 | 9 | 4 | 3 | 36 | 53 | 67 | 74 | 45 | 2 |
| MAE (CLS) | | 2 | 1 | 4 | 5 | 9 | 7 | 1 | 1 | 9 | 4 | 6 | 2 | 3 | 2 | 24 | 2 | 5 | 3 | 4 | 1 | 11 | 11 | 32 | 40 | 31 | 0 |
| MAE (avg) | | 11 | 6 | 18 | 16 | 33 | 38 | 16 | 6 | 16 | 12 | 26 | 10 | 6 | 4 | 44 | 5 | 5 | 1 | 3 | 2 | 29 | 29 | 45 | 68 | 46 | 1 |
| MoCo-v3 | ✓ | 75 | 55 | 76 | 61 | 52 | 49 | 47 | 43 | 39 | 56 | 63 | 31 | 15 | 22 | 66 | 6 | 36 | 10 | 3 | 2 | 17 | 44 | 44 | 71 | 45 | 2 |
| DINO | ✓ | 77 | 55 | 64 | 52 | 51 | 49 | 45 | 41 | 39 | 53 | 48 | 30 | 18 | 17 | 64 | 5 | 32 | 10 | 3 | 2 | 18 | 42 | 46 | 61 | 51 | 2 |
| MAE (avg) | ✓ | 76 | 56 | 75 | 53 | 52 | 49 | 48 | 43 | 39 | 61 | 48 | 30 | 16 | 21 | 71 | 7 | 38 | 11 | 4 | 3 | 21 | 43 | 47 | 75 | 49 | 3 |

Table 15: **AMI score (%) using HDBSCAN, excluding samples rejected by the clusterer as background noise.** *Caution:* These scores are highly inflated because HDBSCAN will reject the samples which are hardest to cluster, and is only being evaluated here on the samples it was confident in clustering—we found HDBSCAN frequently rejected half the samples in a dataset. For the rate at which samples were accepted by HDBSCAN, see Table 16.

| Encoder | FT | In-domain | | | | | Domain-shift | | | | Near-OOD | | | Fine-grained | | | | | | Far-OOD | | | | | | | |
| | | IN1k | INv2 | C10 | C100 | IN9 | 9-FG | 9-MR | IN-R | IN-S | IN-O | LSU | P365 | Air | Cars | F102 | Bio | Birds | iNat | CelA | UTKF | BHis | DTD | ESAT | MNST | Fash | SVHN |
|---|---|---|---|---|---|---|---|---|---|---|---|---|---|---|---|---|---|---|---|---|---|---|---|---|---|---|---|
| Raw image | | 3 | 2 | 12 | 15 | 10 | 8 | 1 | 3 | 16 | 10 | 10 | 3 | 5 | 2 | 20 | 3 | 3 | 1 | 2 | 1 | 18 | 6 | 29 | 75 | 49 | 6 |
| **RN50** — Rand. | | 1 | 1 | 6 | 6 | 4 | 6 | 2 | 1 | 10 | 2 | 4 | 1 | 2 | 1 | 13 | 3 | 2 | 0 | 1 | 1 | 14 | 4 | 21 | 50 | 34 | 1 |
| X-Ent. | | 82 | 61 | 55 | 62 | 55 | 54 | 55 | 53 | 57 | 73 | 57 | 47 | 19 | 29 | 76 | 12 | 54 | 25 | 5 | 4 | 33 | 59 | 67 | 75 | 60 | 10 |
| MoCo-v3 | | 67 | 41 | 56 | 64 | 57 | 55 | 49 | 45 | 52 | 53 | 54 | 43 | 26 | 16 | 89 | 15 | 39 | 16 | 7 | 4 | 38 | 60 | 74 | 82 | 58 | 21 |
| DINO | | 62 | 38 | 46 | 54 | 62 | 60 | 47 | 27 | 39 | 51 | 49 | 46 | 26 | 12 | 90 | 17 | 30 | 13 | 8 | 3 | 51 | 64 | 78 | 74 | 57 | 7 |
| VICReg | | 65 | 40 | 46 | 57 | 55 | 57 | 46 | 31 | 45 | 51 | 52 | 43 | 21 | 14 | 89 | 17 | 28 | 15 | 8 | 4 | 42 | 64 | 76 | 78 | 64 | 11 |
| MoCo-v3 | ✓ | 82 | 62 | 57 | 63 | 55 | 55 | 50 | 51 | 59 | 80 | 64 | 47 | 25 | 29 | 72 | 9 | 55 | 23 | 4 | 4 | 26 | 59 | 67 | 70 | 57 | 11 |
| DINO | ✓ | 82 | 62 | 54 | 62 | 55 | 54 | 50 | 50 | 56 | 80 | 54 | 46 | 26 | 27 | 74 | 9 | 55 | 22 | 5 | 3 | 27 | 58 | 67 | 74 | 60 | 8 |
| VICReg | ✓ | 82 | 59 | 54 | 61 | 55 | 53 | 55 | 49 | 53 | 79 | 56 | 45 | 17 | 26 | 77 | 9 | 55 | 22 | 5 | 3 | 26 | 61 | 70 | 65 | 63 | 9 |
| **ViT-B** — Rand. | | 2 | 1 | 10 | 8 | 7 | 6 | 1 | 2 | 14 | 7 | 9 | 3 | 3 | 2 | 17 | 6 | 2 | 0 | 1 | 1 | 18 | 8 | 31 | 21 | 24 | 0 |
| X-Ent. | | 81 | 65 | 76 | 72 | 53 | 52 | 48 | 53 | 58 | 78 | 63 | 46 | 24 | 29 | 79 | 12 | 51 | 24 | 5 | 3 | 31 | 55 | 59 | 78 | 59 | 6 |
| MoCo-v3 | | 74 | 51 | 72 | 71 | 56 | 56 | 45 | 41 | 50 | 49 | 58 | 48 | 19 | 11 | 88 | 17 | 39 | 19 | 8 | 4 | 49 | 64 | 72 | 81 | 59 | 5 |
| DINO | | 77 | 56 | 69 | 72 | 57 | 57 | 57 | 50 | 57 | 56 | 68 | 49 | 28 | 12 | 93 | 20 | 53 | 25 | 8 | 3 | 51 | 65 | 76 | 79 | 58 | 5 |
| MAE (CLS) | | 38 | 4 | 66 | 38 | 29 | 19 | 6 | 4 | 57 | 19 | 41 | 5 | 6 | 1 | 92 | 18 | 18 | 2 | 22 | 3 | 62 | 33 | 71 | 95 | 73 | 2 |
| MAE (avg) | | 34 | 16 | 32 | 33 | 48 | 52 | 24 | 14 | 32 | 25 | 39 | 30 | 11 | 7 | 66 | 9 | 13 | 4 | 5 | 3 | 42 | 45 | 57 | 76 | 58 | 2 |
| MoCo-v3 | ✓ | 84 | 68 | 79 | 75 | 54 | 54 | 52 | 60 | 64 | 74 | 69 | 48 | 29 | 39 | 81 | 12 | 54 | 27 | 6 | 4 | 32 | 58 | 62 | 79 | 59 | 4 |
| DINO | ✓ | 84 | 69 | 73 | 71 | 53 | 53 | 50 | 57 | 63 | 72 | 62 | 46 | 27 | 33 | 81 | 10 | 56 | 26 | 6 | 3 | 31 | 57 | 60 | 71 | 62 | 4 |
| MAE (avg) | ✓ | 84 | 68 | 77 | 69 | 54 | 53 | 52 | 59 | 64 | 75 | 62 | 47 | 24 | 37 | 87 | 12 | 57 | 29 | 7 | 4 | 36 | 57 | 62 | 80 | 61 | 6 |

Table 16: **Fraction of samples clustered by HDBSCAN.** We indicate the fraction of samples (%) which were placed into a cluster by HDBSCAN—the remaining samples were rejected and placed in the "noise" category. Since every sample in each of the datasets is labelled, such rejections are likely incorrect, sampling a larger fraction of samples clustered is likely to indicate a better clustering attempt. When dealing with curated datasets, it is only plausible that a minority of the samples can truly be outliers.

| Encoder | FT | In-domain | | | | | Domain-shift | | | | Near-OOD | | | Fine-grained | | | | | | Far-OOD | | | | | | | |
| --- | --- | --- | --- | --- | --- | --- | --- | --- | --- | --- | --- | --- | --- | --- | --- | --- | --- | --- | --- | --- | --- | --- | --- | --- | --- | --- | --- |
| | | IN1k | INv2 | C10 | C100 | IN9 | 9-FG | 9-MR | IN-R | IN-S | IN-O | LSU | P365 | Air | Cars | F102 | Bio | Birds | iNat | CelA | UTKF | BHis | DTD | ESAT | MNST | Fash | SVHN |
| Raw image | | 26 | 39 | 37 | 40 | 44 | 48 | 45 | 32 | 39 | 68 | 42 | 29 | 55 | 45 | 41 | 76 | 32 | 22 | 76 | 80 | 59 | 57 | 58 | 79 | 67 | 39 |
| **RN50** — Rand. | | 40 | 54 | 56 | 57 | 58 | 64 | 56 | 50 | 49 | 77 | 63 | 41 | 70 | 50 | 60 | 78 | 49 | 36 | 79 | 82 | 76 | 71 | 71 | 56 | 69 | 53 |
| X-Ent. | | 80 | 71 | 52 | 64 | 90 | 84 | 81 | 55 | 57 | 75 | 58 | 50 | 53 | 46 | 68 | 73 | 51 | 40 | 74 | 78 | 53 | 69 | 79 | 86 | 70 | 33 |
| MoCo-v3 | | 55 | 53 | 51 | 54 | 73 | 66 | 59 | 44 | 53 | 62 | 65 | 45 | 54 | 46 | 84 | 72 | 59 | 39 | 74 | 78 | 77 | 66 | 81 | 84 | 64 | 31 |
| DINO | | 51 | 50 | 44 | 48 | 71 | 70 | 58 | 45 | 53 | 57 | 58 | 42 | 51 | 52 | 84 | 71 | 50 | 36 | 74 | 78 | 67 | 66 | 86 | 84 | 65 | 32 |
| VICReg | | 53 | 54 | 41 | 51 | 66 | 69 | 59 | 47 | 54 | 62 | 60 | 42 | 55 | 59 | 85 | 72 | 47 | 37 | 73 | 76 | 59 | 73 | 88 | 86 | 67 | 30 |
| MoCo-v3 | ✓ | 77 | 71 | 61 | 62 | 89 | 86 | 70 | 55 | 56 | 81 | 71 | 49 | 57 | 48 | 66 | 73 | 49 | 39 | 73 | 77 | 54 | 74 | 68 | 76 | 60 | 29 |
| DINO | ✓ | 77 | 71 | 49 | 58 | 86 | 81 | 66 | 52 | 58 | 82 | 56 | 48 | 58 | 46 | 69 | 74 | 51 | 38 | 73 | 78 | 55 | 74 | 68 | 79 | 64 | 29 |
| VICReg | ✓ | 76 | 72 | 57 | 60 | 89 | 81 | 74 | 51 | 57 | 81 | 67 | 48 | 59 | 50 | 72 | 76 | 54 | 38 | 75 | 79 | 52 | 72 | 73 | 72 | 72 | 32 |
| **ViT-B** — Rand. | | 45 | 58 | 58 | 60 | 66 | 66 | 64 | 52 | 45 | 77 | 67 | 45 | 61 | 57 | 64 | 80 | 53 | 44 | 79 | 80 | 77 | 73 | 75 | 57 | 62 | 56 |
| X-Ent. | | 90 | 85 | 85 | 71 | 95 | 92 | 84 | 65 | 61 | 85 | 78 | 62 | 57 | 44 | 78 | 73 | 53 | 45 | 71 | 76 | 52 | 73 | 70 | 81 | 64 | 33 |
| MoCo-v3 | | 69 | 62 | 78 | 66 | 82 | 82 | 68 | 53 | 58 | 63 | 71 | 51 | 57 | 59 | 83 | 71 | 55 | 42 | 73 | 76 | 59 | 80 | 69 | 86 | 69 | 34 |
| DINO | | 74 | 69 | 72 | 66 | 87 | 84 | 82 | 52 | 59 | 68 | 83 | 51 | 53 | 61 | 89 | 71 | 61 | 43 | 70 | 77 | 65 | 80 | 82 | 87 | 63 | 31 |
| MAE (CLS) | | 6 | 13 | 3 | 10 | 21 | 16 | 7 | 15 | 18 | 12 | 11 | 14 | 28 | 22 | 20 | 40 | 20 | 19 | 33 | 37 | 14 | 25 | 30 | 31 | 35 | 13 |
| MAE (avg) | | 36 | 45 | 40 | 43 | 57 | 62 | 53 | 40 | 52 | 54 | 57 | 33 | 59 | 56 | 62 | 73 | 42 | 28 | 75 | 79 | 64 | 62 | 68 | 82 | 68 | 39 |
| MoCo-v3 | ✓ | 90 | 86 | 93 | 77 | 95 | 88 | 86 | 67 | 64 | 79 | 91 | 61 | 48 | 53 | 76 | 73 | 61 | 44 | 71 | 75 | 46 | 74 | 61 | 81 | 61 | 31 |
| DINO | ✓ | 92 | 86 | 81 | 70 | 95 | 90 | 85 | 68 | 64 | 78 | 70 | 62 | 58 | 51 | 74 | 73 | 55 | 45 | 71 | 76 | 50 | 72 | 67 | 75 | 70 | 28 |
| MAE (avg) | ✓ | 91 | 87 | 97 | 72 | 95 | 90 | 89 | 69 | 64 | 84 | 67 | 59 | 60 | 53 | 79 | 72 | 63 | 44 | 71 | 75 | 50 | 73 | 65 | 87 | 67 | 30 |

# I Predicted Number of Clusters

We report the predicted number of clusters for the three clusterers which do not require a number of clusters to be provided to the clusterer.

As shown in Tables 17–19, the number of clusters predicted varies greatly. We found HDBSCAN usually generated the largest number of clusters, and AC w/o C generated the fewest. The number of clusters predicted was often biased toward the average magnitude (in the order of 100), such that datasets which had fewer GT clusters (in the order of 10) were more likely to be clustered with more clusters than were annotated, and datasets which had more GT clusterers (in the order of 1000) were more likely to be clustered with fewer clusters than were annotated. However, we note that for many datasets the number of classes is ambiguous as the GT categories are hierarchical, and the clustered embeddings may correspond to a coarser granularity than the finest-grained annotations, as discussed in §4.3. Similarly, for datasets which have few annotated classes, it may be feasible to break the data down further into subclasses.

Table 17: **Number of clusters generated using AC w/o C.** Underlined (Bold): encoder which generated clusters with numerosity closest to the GT per dataset (across all clusterers). Background colour scale: logarithmic from smallest underestimate (red) to largest overestimate (blue), centered around the GT number of clusters (white).

| Encoder | FT | In-domain | | | | | Domain-shift | | | | Near-OOD | | | Fine-grained | | | | | | Far-OOD | | | | | | | |
|---|---|---|---|---|---|---|---|---|---|---|---|---|---|---|---|---|---|---|---|---|---|---|---|---|---|---|---|
| | | IN1k | INv2 | C10 | C100 | IN9 | 9-FG | 9-MR | IN-R | IN-S | IN-O | LSU | P365 | Air | Cars | F102 | Bio | Birds | iNat | CelA | UTKF | BHis | DTD | ESAT | MNST | Fash | SVHN |
| *# GT classes* | | 1000 | 1000 | 10 | 100 | 9 | 9 | 9 | 200 | 1000 | 200 | 10 | 365 | 100 | 196 | 102 | 2688 | 555 | 10000 | 1000 | 101 | 32 | 47 | 10 | 10 | 10 | 10 |
| Raw image | | 1543 | 533 | 411 | 398 | 277 | 126 | 344 | 595 | 1221 | 137 | 240 | 1199 | 101 | 340 | 393 | 316 | 914 | 3797 | 324 | 124 | 250 | 159 | 179 | 170 | 47 | 227 |
| **RN50** — Rand. | | 475 | 213 | 254 | 228 | 142 | 198 | 184 | 342 | 513 | 98 | 119 | 351 | 119 | 176 | 156 | 460 | 290 | 735 | 274 | 198 | 111 | 87 | 161 | 55 | 144 | 317 |
| X-Ent. | | 257 | 104 | 54 | 111 | 57 | 52 | 56 | 262 | 275 | 58 | 15 | 321 | 39 | 126 | 79 | 296 | 130 | 436 | 294 | 71 | 76 | 41 | 51 | 73 | 39 | 748 |
| MoCo-v3 | | 50 | 20 | 14 | 22 | 11 | 13 | 11 | 54 | 76 | 9 | 4 | 58 | 6 | 29 | 18 | 37 | 29 | 85 | 46 | 11 | 9 | 9 | 12 | 11 | 11 | 79 |
| DINO | | 90 | 72 | 66 | 89 | 26 | 39 | 68 | 145 | 141 | 86 | 3 | 160 | 4 | 149 | 32 | 124 | 72 | 216 | 269 | 81 | 54 | 53 | 20 | 27 | 13 | 1227 |
| VICReg | | 72 | 43 | 93 | 96 | 26 | 22 | 56 | 263 | 132 | 73 | 3 | 169 | 4 | 184 | 37 | 129 | 61 | 154 | 344 | 119 | 113 | 37 | 17 | 21 | 10 | 548 |
| MoCo-v3 | ✓ | 245 | 95 | 74 | 123 | 45 | 46 | 44 | 228 | 306 | 46 | 20 | 315 | 48 | 112 | 81 | 436 | 131 | 426 | 271 | 65 | 106 | 43 | 58 | 81 | 38 | 938 |
| DINO | ✓ | 241 | 101 | 60 | 114 | 48 | 42 | 47 | 251 | 264 | 50 | 20 | 311 | 49 | 125 | 74 | 463 | 133 | 430 | 274 | 91 | 76 | 40 | 58 | 75 | 45 | 786 |
| VICReg | ✓ | 232 | 96 | 73 | 116 | 47 | 44 | 44 | 253 | 326 | 46 | 17 | 313 | 49 | 151 | 79 | 414 | 138 | 455 | 275 | 96 | 89 | 42 | 61 | 101 | 37 | 876 |
| **ViT-B** — Rand. | | 65 | 56 | 29 | 35 | 42 | 62 | 64 | 57 | 42 | 36 | 35 | 108 | 45 | 95 | 46 | 28 | 22 | 45 | 33 | 21 | 14 | 34 | 10 | 6 | 3 | 15 |
| X-Ent. | | 287 | 143 | 38 | 89 | 56 | 60 | 72 | 183 | 317 | 79 | 19 | 250 | 34 | 108 | 74 | 298 | 143 | 378 | 219 | 64 | 76 | 42 | 51 | 48 | 33 | 399 |
| MoCo-v3 | | 164 | 382 | 64 | 249 | 48 | 61 | 97 | 755 | 355 | 76 | 9 | 864 | 9 | 898 | 58 | 957 | 161 | 525 | 1652 | 197 | 454 | 117 | 85 | 67 | 15 | 4729 |
| DINO | | 213 | 138 | 204 | 698 | 48 | 70 | 102 | 688 | 634 | 320 | 26 | 805 | 19 | 622 | 81 | 671 | 183 | 614 | 2275 | 447 | 694 | 173 | 74 | 103 | 36 | 3804 |
| MAE (CLS) | | 670 | 340 | 244 | 254 | 175 | 130 | 236 | 596 | 375 | 161 | 106 | 495 | 49 | 161 | 176 | 286 | 204 | 548 | 351 | 141 | 70 | 125 | 31 | 52 | 36 | 57 |
| MAE (avg) | | 1906 | 822 | 333 | 361 | 317 | 154 | 266 | 1235 | 459 | 268 | 187 | 1063 | 73 | 199 | 247 | 311 | 198 | 823 | 481 | 105 | 69 | 127 | 25 | 44 | 18 | 57 |
| MoCo-v3 | ✓ | 280 | 147 | 33 | 88 | 50 | 63 | 72 | 192 | 307 | 68 | 19 | 241 | 17 | 121 | 72 | 321 | 118 | 315 | 253 | 60 | 66 | 45 | 53 | 63 | 38 | 737 |
| DINO | ✓ | 299 | 138 | 49 | 100 | 56 | 61 | 69 | 196 | 328 | 65 | 20 | 264 | 23 | 119 | 75 | 308 | 122 | 339 | 250 | 61 | 73 | 44 | 49 | 67 | 42 | 755 |
| MAE (avg) | ✓ | 290 | 152 | 46 | 98 | 53 | 60 | 67 | 197 | 319 | 72 | 15 | 263 | 11 | 130 | 71 | 244 | 122 | 301 | 227 | 64 | 80 | 40 | 52 | 55 | 35 | 683 |

Table 18: **Number of clusters generated using Affinity Prop.**

| Encoder | FT | In-domain | | | | | Domain-shift | | | | Near-OOD | | | Fine-grained | | | | | | Far-OOD | | | | | | | |
|---|---|---|---|---|---|---|---|---|---|---|---|---|---|---|---|---|---|---|---|---|---|---|---|---|---|---|---|
| | | IN1k | INv2 | C10 | C100 | IN9 | 9-FG | 9-MR | IN-R | IN-S | IN-O | LSU | P365 | Air | Cars | F102 | Bio | Birds | iNat | CelA | UTKF | BHis | DTD | ESAT | MNST | Fash | SVHN |
| *# GT classes* | | 1000 | 1000 | 10 | 100 | 9 | 9 | 9 | 200 | 1000 | 200 | 10 | 365 | 100 | 196 | 102 | 2688 | 555 | 10000 | 1000 | 101 | 32 | 47 | 10 | 10 | 10 | 10 |
| Raw image | | 1151 | 299 | 317 | 317 | 153 | 249 | 143 | 572 | 2116 | 73 | 111 | 828 | 120 | 310 | 220 | 913 | 632 | 1969 | 582 | 212 | 137 | 66 | 108 | 784 | 315 | 428 |
| **RN50** — Rand. | | 804 | 220 | 244 | 227 | 113 | 169 | 119 | 443 | 818 | 64 | 103 | 646 | 110 | 234 | 145 | 562 | 423 | 1238 | 424 | 173 | 80 | 48 | 68 | 452 | 198 | 316 |
| X-Ent. | | 227 | 108 | 54 | 59 | 47 | 33 | 46 | 179 | 293 | 65 | 23 | 167 | 33 | 90 | 70 | 121 | 78 | 179 | 106 | 64 | 61 | 43 | 32 | 40 | 17 | 202 |
| MoCo-v3 | | 232 | 102 | 57 | 87 | 44 | 51 | 75 | 241 | 437 | 67 | 23 | 219 | 29 | 96 | 62 | 129 | 106 | 313 | 148 | 75 | 64 | 41 | 29 | 20 | 538 | 194 |
| DINO | | 210 | 86 | 72 | 96 | 35 | 39 | 78 | 205 | 369 | 59 | 19 | 204 | 40 | 101 | 58 | 98 | 93 | 295 | 140 | 77 | 58 | 51 | 25 | 40 | 31 | 210 |
| VICReg | | 205 | 81 | 68 | 86 | 33 | 39 | 73 | 209 | 364 | 56 | 13 | 191 | 30 | 82 | 64 | 112 | 96 | 293 | 147 | 83 | 60 | 40 | 22 | 33 | 977 | 175 |
| MoCo-v3 | ✓ | 169 | 61 | 41 | 62 | 28 | 31 | 34 | 181 | 278 | 46 | 23 | 161 | 48 | 88 | 68 | 154 | 79 | 172 | 114 | 66 | 73 | 35 | 35 | 74 | 17 | 265 |
| DINO | ✓ | 184 | 64 | 52 | 62 | 27 | 27 | 40 | 184 | 265 | 49 | 21 | 164 | 37 | 87 | 67 | 155 | 81 | 179 | 120 | 67 | 67 | 39 | 33 | 43 | 26 | 238 |
| VICReg | ✓ | 172 | 72 | 54 | 60 | 29 | 31 | 39 | 188 | 298 | 51 | 23 | 164 | 35 | 91 | 64 | 145 | 86 | 182 | 110 | 73 | 70 | 38 | 31 | 53 | 19 | 251 |
| **ViT-B** — Rand. | | 616 | 203 | 192 | 199 | 108 | 177 | 104 | 397 | 1252 | 68 | 79 | 520 | 94 | 174 | 159 | 439 | 308 | 963 | 293 | 132 | 96 | 65 | 85 | 184 | 79 | 254 |
| X-Ent. | | 263 | 141 | 24 | 52 | 47 | 53 | 67 | 164 | 303 | 83 | 19 | 157 | 28 | 62 | 65 | 144 | 78 | 176 | 101 | 55 | 72 | 41 | 33 | 28 | 15 | 171 |
| MoCo-v3 | | 1256 | 99 | 31 | 60 | 37 | 43 | 75 | 187 | 460 | 46 | 13 | 221 | 31 | 76 | 62 | 107 | 70 | 208 | 104 | 66 | 60 | 36 | 24 | 24 | 4951 | 169 |
| DINO | | 363 | 44 | 43 | 81 | 24 | 27 | 40 | 165 | 322 | 46 | 19 | 151 | 33 | 60 | 58 | 79 | 57 | 166 | 91 | 71 | 63 | 37 | 20 | 33 | 17 | 155 |
| MAE (CLS) | | 2001 | 505 | 426 | 462 | 254 | 256 | 248 | 1220 | 3670 | 132 | 159 | 1350 | 172 | 390 | 357 | 975 | 943 | 2996 | 861 | 302 | 206 | 143 | 160 | 397 | 297 | 872 |
| MAE (avg) | | 428 | 112 | 99 | 127 | 57 | 49 | 67 | 404 | 943 | 49 | 64 | 231 | 35 | 67 | 103 | 168 | 1676 | 1060 | 208 | 88 | 58 | 47 | 38 | 43 | 7124 | 182 |
| MoCo-v3 | ✓ | 255 | 144 | 23 | 57 | 42 | 65 | 73 | 188 | 307 | 76 | 19 | 165 | 15 | 65 | 68 | 117 | 67 | 138 | 88 | 55 | 67 | 46 | 35 | 39 | 17 | 235 |
| DINO | ✓ | 281 | 129 | 29 | 52 | 46 | 63 | 64 | 184 | 333 | 69 | 18 | 158 | 19 | 58 | 67 | 126 | 65 | 158 | 105 | 58 | 67 | 44 | 35 | 40 | 16 | 259 |
| MAE (avg) | ✓ | 270 | 155 | 29 | 52 | 45 | 58 | 65 | 184 | 300 | 81 | 18 | 161 | 7 | 67 | 63 | 109 | 62 | 141 | 96 | 58 | 74 | 41 | 26 | 34 | 21 | 241 |

Table 19: **Number of clusters generated using HDBSCAN.**

| Encoder | FT | In-domain | | | | | Domain-shift | | | | Near-OOD | | | Fine-grained | | | | | | Far-OOD | | | | | | | |
|---|---|---|---|---|---|---|---|---|---|---|---|---|---|---|---|---|---|---|---|---|---|---|---|---|---|---|---|
| | | IN1k | INv2 | C10 | C100 | IN9 | 9-FG | 9-MR | IN-R | IN-S | IN-O | LSU | P365 | Air | Cars | F102 | Bio | Birds | iNat | CelA | UTKF | BHis | DTD | ESAT | MNST | Fash | SVHN |
| *# GT classes* | | 1000 | 1000 | 10 | 100 | 9 | 9 | 9 | 200 | 1000 | 200 | 10 | 365 | 100 | 196 | 102 | 2688 | 555 | 10000 | 1000 | 101 | 32 | 47 | 10 | 10 | 10 | 10 |
| Raw image | | 882 | 226 | 243 | 245 | 107 | 118 | 98 | 572 | 1237 | 43 | 88 | 694 | 96 | 213 | 156 | 6154 | 483 | 1490 | 5137 | 1569 | 82 | 79 | 95 | 168 | 284 | 602 |
| **RN50** — Rand. | | 1310 | 333 | 322 | 325 | 166 | 148 | 150 | 929 | 1586 | 69 | 121 | 1066 | 123 | 274 | 207 | 6374 | 735 | 2529 | 5176 | 1547 | 120 | 90 | 168 | 274 | 448 | 809 |
| X-Ent. | | 1181 | 481 | 228 | 196 | 227 | 214 | 167 | 533 | 1714 | 119 | 77 | 740 | 98 | 230 | 180 | 6007 | 526 | 1617 | 4939 | 1503 | 134 | 76 | 80 | 81 | 178 | 617 |
| MoCo-v3 | | 1302 | 337 | 222 | 236 | 172 | 165 | 141 | 605 | 2002 | 104 | 59 | 728 | 114 | 242 | 138 | 5905 | 414 | 1685 | 4954 | 1515 | 121 | 57 | 56 | 81 | 214 | 544 |
| DINO | | 1160 | 329 | 241 | 251 | 138 | 130 | 118 | 578 | 1967 | 81 | 76 | 697 | 111 | 167 | 143 | 5791 | 456 | 1616 | 4873 | 1540 | 140 | 72 | 49 | 97 | 183 | 618 |
| VICReg | | 1224 | 344 | 256 | 240 | 179 | 150 | 145 | 582 | 2020 | 86 | 71 | 724 | 115 | 190 | 158 | 5822 | 594 | 1659 | 4901 | 1493 | 128 | 69 | 33 | 90 | 170 | 575 |
| MoCo-v3 | ✓ | 1174 | 474 | 177 | 190 | 225 | 193 | 188 | 517 | 1592 | 132 | 60 | 728 | 86 | 209 | 182 | 6062 | **561** | 1653 | 4830 | 1483 | 119 | 62 | 81 | 127 | 215 | 545 |
| DINO | ✓ | 1230 | 482 | 234 | 236 | 229 | 198 | 193 | 608 | 1615 | 133 | 95 | 765 | 94 | 231 | 161 | 6067 | 575 | 1736 | 4880 | 1544 | 118 | 71 | 92 | 128 | 188 | 533 |
| VICReg | ✓ | 1247 | 483 | 224 | 232 | 217 | 211 | 194 | 599 | 1621 | 133 | 75 | 805 | 114 | 235 | 166 | 6305 | 493 | 1842 | 4980 | 1519 | 115 | 67 | 79 | 190 | 170 | 599 |
| **ViT-B** — Rand. | | 1454 | 368 | 383 | 401 | 159 | 161 | 172 | 946 | 1570 | 77 | 118 | 1116 | 136 | 264 | 256 | 6498 | 835 | 3061 | 5184 | 1511 | 141 | 74 | 208 | 356 | 394 | 956 |
| X-Ent. | | 1102 | 640 | 88 | 210 | 254 | 232 | 211 | 463 | 1571 | 145 | 51 | 569 | 86 | 233 | 162 | 6026 | 519 | 1333 | 4778 | 1489 | 104 | 65 | 96 | 104 | 207 | 573 |
| MoCo-v3 | | 1145 | 416 | 105 | 235 | 196 | 180 | 179 | 567 | 1873 | 99 | 60 | 685 | 97 | 164 | 162 | 5927 | 456 | 1592 | 4884 | 1469 | 139 | 68 | 93 | 85 | 171 | 548 |
| DINO | | 1131 | 416 | 154 | 224 | 191 | 173 | 161 | 583 | 1912 | 89 | 40 | 726 | 114 | 167 | 150 | 5786 | 452 | 1548 | 4681 | 1484 | 141 | 60 | 57 | 78 | 205 | 576 |
| MAE (CLS) | | 133 | 19 | 19 | 21 | 15 | 14 | 5 | 48 | 1095 | 6 | 6 | 31 | 10 | 3 | 52 | **3671** | 50 | 40 | 2712 | 845 | **50** | 12 | 20 | 17 | 14 | 51 |
| MAE (avg) | | **1029** | 249 | 274 | 285 | 137 | 136 | 130 | 584 | 1947 | 70 | 76 | 698 | 91 | 154 | 195 | 5898 | 614 | 1984 | 5000 | 1476 | 137 | 78 | 130 | 131 | 212 | 720 |
| MoCo-v3 | ✓ | 1103 | 651 | 54 | 181 | 250 | 239 | 208 | 464 | 1446 | 130 | 26 | 555 | 89 | 190 | 172 | 6006 | 398 | 1271 | 4750 | 1502 | 96 | 61 | 91 | 104 | 204 | 458 |
| DINO | ✓ | 1102 | 677 | 99 | 202 | 253 | 252 | 215 | 436 | 1478 | 128 | 65 | 579 | 70 | 212 | 155 | 6054 | 464 | 1323 | 4670 | 1507 | 110 | 64 | 93 | 147 | 177 | 514 |
| MAE (avg) | ✓ | 1092 | 649 | 34 | 197 | 244 | 237 | 198 | 423 | 1514 | 129 | 69 | 621 | 70 | 221 | 157 | 6074 | 395 | 1357 | 4728 | 1461 | 124 | 66 | 85 | 81 | 183 | 490 |

## J Silhouette Scores

Our results on the silhouette score are broadly in line with our main finding on the AMI between clusterings and annotation targets, reported in §4. For both the ResNet-50 and ViT-B encoders, the supervised model has the highest silhouette score by a large margin of 0.25–0.3, but otherwise the clustering quality across the encoders is very similar, achieving similar silhouette scores to each other. There are some exceptions to this, such as the silhouette scores for MAE which are near 0, illustrating the intrinsically-poor quality of the clusters it exhibited and hence it is not well-suited to this task.

Despite the very low AMI scores, we observe the silhouette scores for SVHN are generally comparable to the silhouette scores of the other datasets. We believe this is due to the heterogeneity within the classes in SVHN, where house-numbers can be written in different formats, colours, etc., and thus the encoded images can be appropriately grouped together, even if the semantic meaning of the clusters does not correspond to the identity of the digit in the center of the image.

Between the clusterers, K-Means and AC typically achieve the highest silhouette scores. For HDBSCAN, the silhouette scores were often significantly negative. This is because HDBSCAN builds clusters based on transitions in density, and the non-convex clusters that result from this can score poor silhouette scores (a known caveat to this evaluation metric). For Affinity Propagation, we observe silhouette scores near 0, indicating the clusters it discovered have high overlap with each other and are of low quality, corresponding to its poor AMI performance.

## K Effect of Fine-tuning SSL Encoders for IN-1k Classification

In §4.4, we showed the performance clusterings created using SSL-pretrained encoders after fine-tuning on IN-1k to better align them with a categorical classification task similar to that of clustering images.

Our graphs of the fine-tuned model performance (Figure 3) used the same supervised baseline as illustrated for the SSL-only (Figure 6), making the bars comparable. But for further clarity, we show in Figure 6 the direct comparison of the SSL+FT models against the SSL-only models. This makes the direction of change in the performance—and its consistency across models and architectures—clearer.

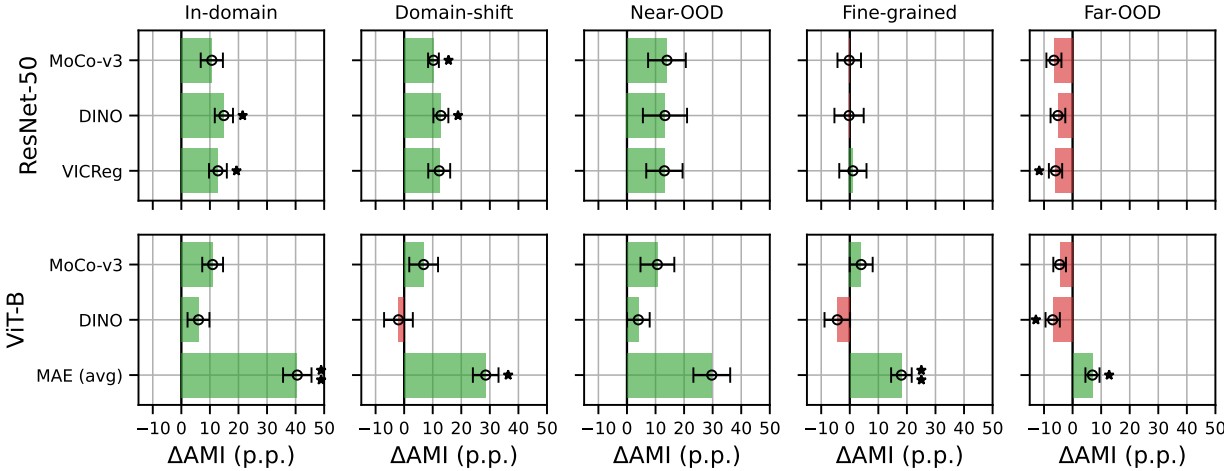

Figure 6: **Percentage-point (p.p.) difference in AMI between clusters formed from embeddings of SSL-pretrained networks after fine-tunings on IN-1k versus only self-supervision.** We measure the difference in AMI (mean over 6 clusterers) with SSL encoders after fine-tuning with cross-entropy on IN-1k compared to those encoders with SSL pretraining on IN-1k only (error bars: $\pm 1$ stderr; $3 \leq N \leq 8$ datasets; $\star$: $p < 0.05$; $\star\star$: $p < 0.01$). Complements Figure 1 and Figure 3 from the main text.

## L    Detailed Comparison of Performances Across Clustering Methods

We sought to evaluate the clusterers to see which clustering methodology produced the best results when using a pretrained encoder. For each set of embeddings, created by passing one of the datasets listed in Appendix H through one of the pretrained ResNet-50 or ViT-B encoders (X-Ent., MoCo-v3, DINO, VICReg, or MAE), we compared the results of clustering that set of embeddings with each of the clusterers (tabulated in Table 9–14).

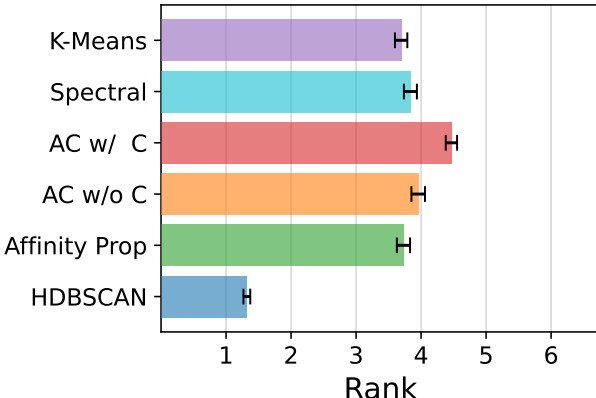

Figure 7: **Average clusterer rank** (higher is better). For each set of embeddings we apply each clusterer, compare the AMI of their clusters, and rank them against each other (lowest AMI $\rightarrow$ rank 1, highest AMI $\rightarrow$ rank 6). Error bars: $\pm 1$ stderr; $N = 225$.

We compared the performance of the clustering methods by ranking each clusterer for each combination of pretrained encoder and dataset, shown in Figure 7. The results show that AC w/ C performs best most often ($p < 0.05$; Wilcoxon signed-rank test versus each other clusterer). Spectral, K-Means, AC w/o C, and AP all perform similarly. HDBSCAN frequently performed worst ($p < 10^{-33}$).

We note that HDBSCAN identifies samples which belong to *no* cluster (noise/background samples). Unless stated otherwise, we consider the noise class to be its own class when computing the AMI. This sets HDBSCAN at a disadvantage, since the samples it identifies as noise are typically distributed across all GT classes, but is fairer than ignoring samples it identifies as noise since that would evaluate it only on easier samples. If we instead exclude the noise class and only evaluate the AMI on samples which HDBSCAN placed in a real cluster, we find it yields the best performance of all clusterers, shown in Table 15, suggesting that HDBSCAN can provide value depending on the goals of the user performing the clustering. However, we note that on the well-labelled datasets we considered, HDBSCAN frequently rejected half of the samples (see Table 16).

Table 20: **Pearson correlation coefficient between clusterers.** For each pair of clustering methods, we measure the Pearson correlation coefficient (%) between the AMI each attained when clustering the embeddings of a given dataset with a given encoder. We utilize datapoints across all datasets and all encoders, including fine-tuned, randomized (untrained), and raw pixels. **Bold**: for a given clustering method (column), the clustering method (row) that it is most correlated with.

|  | K-Means | Spectral | AC w/ C | AC w/o C | Affinity Prop | HDBSCAN |
|---|---|---|---|---|---|---|
| K-Means | – | **97.6** | **99.0** | 97.4 | 97.4 | **94.8** |
| Spectral | 97.6 | – | 97.1 | 97.2 | 96.0 | 94.3 |
| AC w/ C | **99.0** | 97.1 | – | 97.5 | 96.9 | 94.5 |
| AC w/o C | 97.4 | 97.2 | 97.5 | – | **97.7** | 93.1 |
| Affinity Prop | 97.4 | 96.0 | 96.9 | **97.7** | – | 93.6 |
| HDBSCAN | 94.8 | 94.3 | 94.5 | 93.1 | 93.6 | – |

We investigated the correlation between the AMI for each pair of clustering methods, shown in Table 20 and illustrated in Figure 8. We found the correlation between clusterers was generally high ($0.931 \leq r \leq 0.990$). The performance of HDBSCAN was less correlated with the other clusterers ($r \leq 0.948$ vs $r \geq 0.960$).

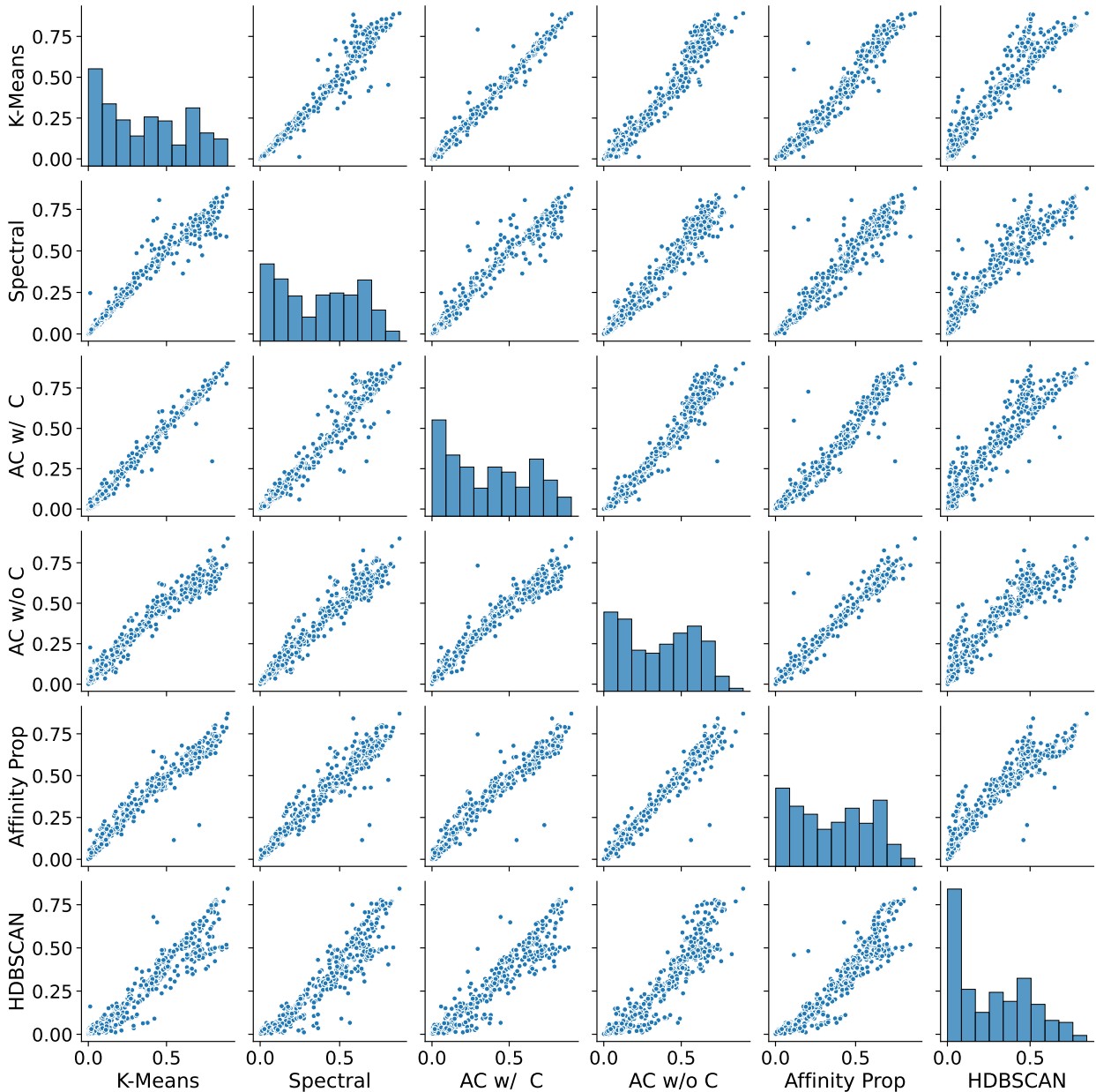

Figure 8: **Correlation of AMI between clustering methods.** For each pair of clustering methods, we show a scatter plot of the AMI each attained when clustering the embeddings of a given dataset with a given encoder. We show all datasets and all encoders, including fine-tuned, randomized (untrained), and raw pixels. Along the diagonal, the distribution of AMI values is shown for each clusterer.

## M   Detailed Comparison Between Encoders

We computed and evaluated the Pearson correlation coefficient between the clusterings of pairs of encoders.

Looking across model architectures (Table 21) we find the performance of SSL encoders are typically more correlated with other SSL models of the same architecture than with the same pretraining loss but a different architecture.

As shown in Table 22 for ResNet-50 models, and Table 23 for ViT-B models, the performance of the fine-tuned models ([FT]) were well correlated with each other ($r \geq 0.989$) and with the supervised trained model (X-Ent.; $r \geq 0.978$). We also observed the performance of the whole-image SSL models were highly correlated with each other ($r \geq 0.946$, and the two read-outs of the MAE model were strongly correlated with each other ($r = 0.912$). Outside of these blocks, correlation scores were lower. In particular, we note the performance of FT encoders was much more correlated with the X-Entropy models than that of their original SSL-only pretrained encoder.

Table 21: **Pearson correlation coefficient between initial encoders.** For each pair of pretrained encoders (without fine-tuning), we measure the Pearson correlation coefficient (%) between the AMI each attained when clustering the embeddings of a given dataset with a given clusterer. **Bold**: for a given encoder (column), the other encoder (row) that it is most correlated with.

| | Raw image | ResNet-50 | | | | | ViT-B | | | | | |
| | | Rand. | X-Ent. | MoCo-v3 | DINO | VICReg | Rand. | X-Ent. | MoCo-v3 | DINO | MAE (CLS) | MAE (avg) |
| --- | --- | --- | --- | --- | --- | --- | --- | --- | --- | --- | --- | --- |
| Raw image | – | **96.1** | 39.8 | 58.6 | 54.8 | 57.7 | 71.0 | 39.0 | 48.8 | 43.0 | 69.6 | 66.8 |
| **RN50** — Rand. | 96.1 | – | 39.3 | 58.4 | 58.0 | 59.6 | **78.5** | 36.5 | 48.1 | 44.4 | 72.7 | 68.1 |
| X-Ent. | 39.8 | 39.3 | – | 89.8 | 86.2 | 87.8 | 41.2 | **95.2** | 87.5 | 93.6 | 74.5 | 73.2 |
| MoCo-v3 | 58.6 | 58.4 | 89.8 | – | 96.1 | 97.6 | 60.4 | 84.5 | 93.1 | 94.2 | 88.5 | 88.6 |
| DINO | 54.8 | 58.0 | 86.2 | 96.1 | – | **99.1** | 67.3 | 77.7 | 91.1 | 92.9 | 90.1 | 91.6 |
| VICReg | 57.7 | 59.6 | 87.8 | **97.6** | **99.1** | – | 67.0 | 80.9 | 92.5 | 93.5 | 90.3 | **92.1** |
| **ViT-B** — Rand. | 71.0 | 78.5 | 41.2 | 60.4 | 67.3 | 67.0 | – | 40.0 | 59.5 | 55.7 | 71.8 | 74.7 |
| X-Ent. | 39.0 | 36.5 | **95.2** | 84.5 | 77.7 | 80.9 | 40.0 | – | 86.7 | 90.4 | 65.7 | 67.3 |
| MoCo-v3 | 48.8 | 48.1 | 87.5 | 93.1 | 91.1 | 92.5 | 59.5 | 86.7 | – | **94.6** | 81.8 | 87.1 |
| DINO | 43.0 | 44.4 | 93.6 | 94.2 | 92.9 | 93.5 | 55.7 | 90.4 | **94.6** | – | 79.7 | 81.5 |
| MAE (CLS) | 69.6 | 72.7 | 74.5 | 88.5 | 90.1 | 90.3 | 71.8 | 65.7 | 81.8 | 79.7 | – | 91.2 |
| MAE (avg) | 66.8 | 68.1 | 73.2 | 88.6 | 91.6 | 92.1 | 74.7 | 67.3 | 87.1 | 81.5 | **91.2** | – |

Table 22: **Pearson correlation coefficient between ResNet-50 encoders.** For each pair of pretrained encoders, we measure the Pearson correlation coefficient (%) between the AMI each attained when clustering the embeddings of a given dataset with a given clusterer. [FT]: fine-tuned with cross-entropy on IN-1k. **Bold**: for a given encoder (column), the other encoder (row) that it is most correlated with.

| | Rand. | X-Ent. | MoCo-v3 | DINO | VICReg | MoCo-v3 [FT] | DINO [FT] | VICReg [FT] |
|---|---|---|---|---|---|---|---|---|
| Rand. | – | 39.3 | 58.4 | 58.0 | 59.6 | 28.5 | 34.1 | 32.0 |
| X-Ent. | 39.3 | – | 89.8 | 86.2 | 87.8 | 97.8 | 98.9 | 98.4 |
| MoCo-v3 | 58.4 | 89.8 | – | 96.1 | 97.6 | 84.7 | 87.0 | 86.9 |
| DINO | 58.0 | 86.2 | 96.1 | – | **99.1** | 80.8 | 82.7 | 83.3 |
| VICReg | **59.6** | 87.8 | **97.6** | **99.1** | – | 82.0 | 84.4 | 84.5 |
| MoCo-v3 [FT] | 28.5 | 97.8 | 84.7 | 80.8 | 82.0 | – | 99.2 | **99.5** |
| DINO [FT] | 34.1 | **98.9** | 87.0 | 82.7 | 84.4 | 99.2 | – | 99.5 |
| VICReg [FT] | 32.0 | 98.4 | 86.9 | 83.3 | 84.5 | **99.5** | **99.5** | – |

Table 23: **Pearson correlation coefficient between ViT-B encoders.** For each pair of pretrained encoders, we measure the Pearson correlation coefficient (%) between the AMI each attained when clustering the embeddings of a given dataset with a given clusterer. [FT]: fine-tuned with cross-entropy on IN-1k. **Bold**: for a given encoder (column), the other encoder (row) that it is most correlated with.

| | Rand. | X-Ent. | MoCo-v3 | DINO | MAE (CLS) | MAE (avg) | MoCo-v3 [FT] | DINO [FT] | MAE (avg) [FT] |
|---|---|---|---|---|---|---|---|---|---|
| Rand. | – | 40.0 | 59.5 | 55.7 | 71.8 | 74.7 | 37.2 | 35.5 | 39.9 |
| X-Ent. | 40.0 | – | 86.7 | 90.4 | 65.7 | 67.3 | 97.9 | 97.9 | 98.4 |
| MoCo-v3 | 59.5 | 86.7 | – | **94.6** | 81.8 | 87.1 | 85.8 | 85.5 | 87.2 |
| DINO | 55.7 | 90.4 | **94.6** | – | 79.7 | 81.5 | 88.8 | 89.1 | 90.8 |
| MAE (CLS) | 71.8 | 65.7 | 81.8 | 79.7 | – | **91.2** | 62.4 | 63.1 | 65.2 |
| MAE (avg) | **74.7** | 67.3 | 87.1 | 81.5 | **91.2** | – | 65.4 | 65.0 | 67.9 |
| MoCo-v3 [FT] | 37.2 | 97.9 | 85.8 | 88.8 | 62.4 | 65.4 | – | 99.2 | 98.9 |
| DINO [FT] | 35.5 | 97.9 | 85.5 | 89.1 | 63.1 | 65.0 | **99.2** | – | **99.2** |
| MAE (avg) [FT] | 39.9 | **98.4** | 87.2 | 90.8 | 65.2 | 67.9 | 98.9 | **99.2** | – |

## N ImageNet-9 Examples

As described in §4.5, we used the variants of the ImageNet-9 backgrounds challenge dataset (Xiao et al., 2020) to evaluate whether SSL-encoded clusters prioritized foreground and background components of the stimulus differently to clusters using embeddings from supervised models. In Figure 9, we provide illustrative example stimuli from the variants of this dataset.

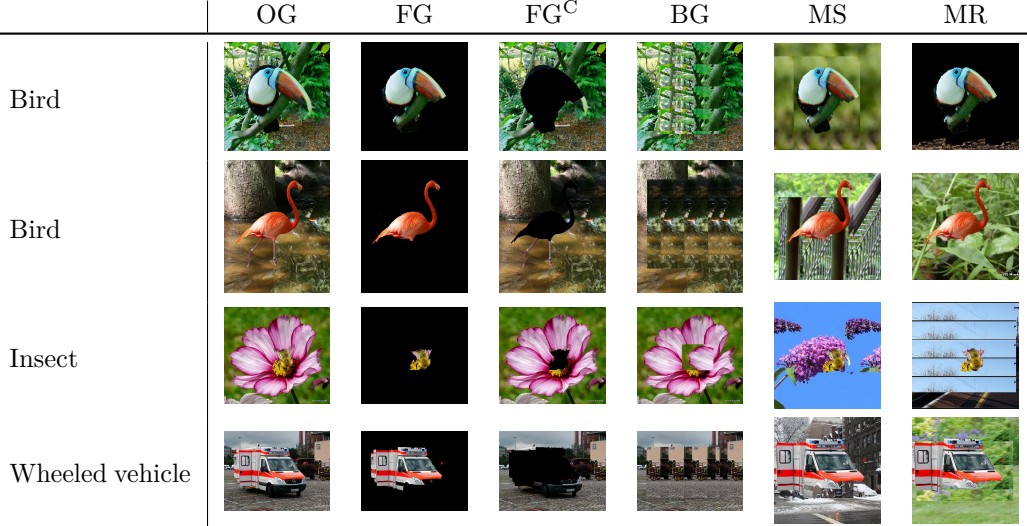

Figure 9: **Example images from the ImageNet-9 dataset.** For three classes (bird, insect, and wheeled vehicle) we show a sample from each of the variant datasets: original images (OG), foreground only (FG), foreground removed and replaced with black (FG$^C$), background only (bounding box replaced with background texture; BG), mixed-same (foreground overlaid on the background of a sample of the same class; MS), and mixed-random (foreground overlaid on the background of a random sample; MR). We note that MS places the foreground object on an appropriate background, whereas MR places the foreground on a background which may be out-of-context for the foreground. ImageNet-9 labels are coarse-grained superclasses, each spanning multiple IN-1k classes, hence images of toucan and flamingo are both labelled "bird".

## O ImageNet-Rendition Information Breakdown

We sought to better understand what information about the stimulus is being captured in the clusters. By using a dataset which possesses more than one annotation per image, we can investigate the agreement between the clusterings and each annotation type. The ImageNet-Rendition dataset in particular has primary annotations for the object class represented in the image (goldfish, great white shark, cowboy hat, volcano, etc.), but also annotations for the style of rendition (cartoon, graffiti, embroidery, origami, etc.), see Figure 10. We compute the AMI for each annotation stream, see Table 24.

Our results indicate there is generally a trade-off between the two: embeddings which are grouped according to object class identities are not grouped according to the artform, and vice versa. This trend is true across all ResNet-50 encoders and supervised ViT-B, but MoCo-v3 and DINO ViT-B embeddings can capture information about both aspects.

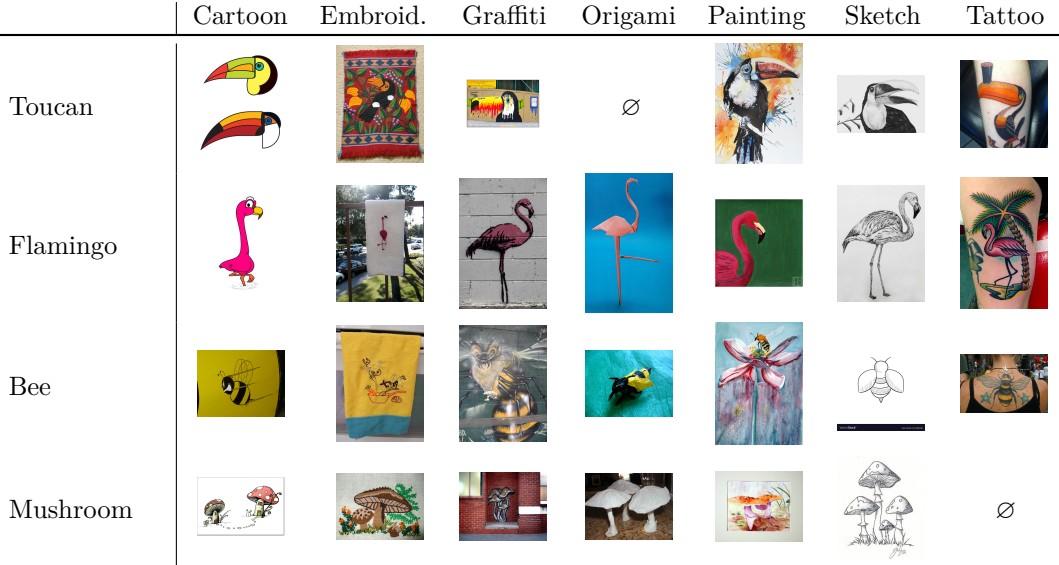

Figure 10: **Example images from ImageNet-R by both class and artform style.** ∅ indicates no images in that artform for this class. In our experiments, we measure the AMI between the clusterings and the labels pooled across each row (Object), each column (Artform), or using only the labels per row–column combination/cell (Both).

Table 24: **ImageNet-Rendition Breakdown.** Information (AMI, %) about different aspects of images in ImageNet-R: the IN-1k object class represented, the style of rendition, and their combinations. **Bold**: best encoder per aspect. Underlined: best encoder per arch. Background: from median AMI (white) to max (blue) per aspect. FT: fine-tuned with x-ent. on IN-1k.

| Arch. | Encoder | FT | Class | Artform | Both |
|-------|---------|----|-------|---------|------|
| **RN50** | X-Ent. | | 34 | 19 | 29 |
| | MoCo-v3 | | 26 | 19 | 23 |
| | DINO | | 18 | **24** | 20 |
| | VICReg | | 20 | 23 | 21 |
| | MoCo-v3 | ✓ | 35 | 18 | 29 |
| | DINO | ✓ | 34 | 18 | 28 |
| | VICReg | ✓ | 33 | 19 | 28 |
| **ViT-B** | X-Ent. | | 38 | 19 | 32 |
| | MoCo-v3 | | 26 | **25** | 26 |
| | DINO | | 33 | 23 | 30 |
| | MAE (CLS) | | 10 | 16 | 11 |
| | MAE (avg) | | 10 | 19 | 13 |
| | MoCo-v3 | ✓ | **44** | 18 | 36 |
| | DINO | ✓ | 43 | 18 | 35 |
| | MAE (avg) | ✓ | **44** | 18 | **36** |

## P    BreakHis Information Breakdown

BreakHis (Spanhol et al., 2016) is a medical dataset containing images of microscopic images of breast tumor tissue collected from 81 patients. At a coarse level, the tumor can be malignant (cancerous) or benign (normal cells). Within each of these categories, the dataset contains samples for four distinct types of benign tumor and four types of malignant tumor. Images were taken for each slide (one slide per subject) at varying zoom levels (40x, 100x, 200x, 400x).

We investigated how much information the clustered embeddings contained about each of these labels, shown in Table 25. We found that SSL pretrained encoders were much better at encoding the medically relevant information about the tumor's malignancy and specific type, with up to twice as much AMI than the supervised and fine-tuned models. The embeddings from the SSL encoders were generally also superior for encoding the magnification level, and the slide ID. However, the MAE model's clusters were worst at encoding the magnification, and the MoCo-v3 model worst at encoding the slide ID. We hypothesize that MoCo-v3's poor performance on slide ID may be because the types of differences between subjects may be comparable to the augmentations it is tasked with being *robust* to during training.

Across all label types for this dataset, SSL pretrained models produced the best clusters. Within these, DINO was the best performing model with either ResNet-50 or ViT-B architecture. The DINO training paradigm features multi-crop training, which may have helped the encoder to produce encodings which work well on this dataset which includes images at a variety of zoom levels and hence features at varying apparent scales.

Table 25: **BreakHis Breakdown.** Information (AMI, %) about different aspects of images in BreakHis: **Bold**: best encoder per aspect. Underlined: best encoder per arch. Background: from median AMI (white) to max (blue) per aspect. FT: fine-tuned with x-ent. on IN-1k.

| Arch. | Encoder | FT | Malignancy | Tumor type | Magnification | Tumor type x Magnifn. | Slide ID |
|---|---|---|---|---|---|---|---|
| **RN50** | X-Ent. | | 7 | 12 | 23 | 26 | 23 |
| | MoCo-v3 | | 5 | 10 | 31 | 30 | 18 |
| | DINO | | 14 | 22 | 35 | 43 | 35 |
| | VICReg | | 9 | 15 | 33 | 36 | 25 |
| | MoCo-v3 | ✓ | 6 | 10 | 19 | 22 | 18 |
| | DINO | ✓ | 7 | 11 | 18 | 22 | 20 |
| | VICReg | ✓ | 6 | 10 | 19 | 22 | 19 |
| **ViT-B** | X-Ent. | | 7 | 13 | 20 | 25 | 23 |
| | MoCo-v3 | | 12 | 19 | 30 | 37 | 31 |
| | DINO | | **14** | **22** | 32 | 40 | **36** |
| | MAE (CLS) | | **14** | 19 | 19 | 28 | 32 |
| | MAE (avg) | | 11 | 16 | 25 | 30 | 26 |
| | MoCo-v3 | ✓ | 6 | 10 | 22 | 24 | 19 |
| | DINO | ✓ | 7 | 12 | 21 | 24 | 21 |
| | MAE (avg) | ✓ | 8 | 13 | 22 | 26 | 22 |

# Q   Correlation Between AMI and Silhouette Score

In addition to the scatter plot of the ranked AMI and silhouette scores shown in the main paper (Figure 4), we also provide the scatter plot of the actual AMI and $S$ values in Figure 11. We observe that in the UMAP-reduced embedding space a larger extent of the silhouette score's range is used, making the correlation between AMI and $S$ more clear. This increases the usability of the silhouette score as a proxy.

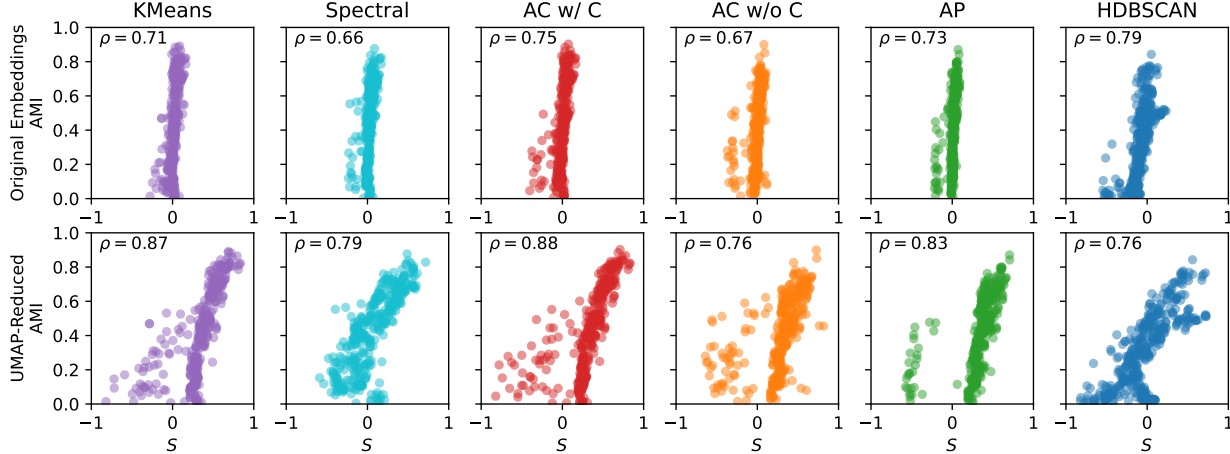

Figure 11: **AMI–Silhouette scatter plots.** The AMI and silhouette score ($S$) per clusterer, across datasets and encoders. The silhouette scores are measured in the original (top) and UMAP-reduced 50-d (bottom) feature spaces. We indicate the per-clustering-method Spearman's rank correlation ($\rho$).

