# OpenReview forum: "An Empirical Study into Clustering of Unseen Datasets with Self-Supervised Encoders"
_TMLR — Decision pending for TMLR_

### Review · Reviewer_z7Md · 2026-05-29

**Summary Of Contributions:**

The paper studies zero-shot clustering: taking image encoders pretrained only on ImageNet-1k (one supervised X-Ent baseline; four SSL paradigms — MoCo-v3, DINO, VICReg, MAE — on ResNet-50 and ViT-B
  backbones) and clustering their frozen embeddings on 26 unseen datasets with six classical clusterers, without any per-dataset tuning. Main findings: (i) supervised encoders cluster better near the training
  domain, SSL (and fine-tuned SSL) better far out-of-domain; (ii) manifold reduction (UMAP) before clustering is essential; (iii) Agglomerative Clustering (Ward, L2) is the best clusterer, though by a small
  margin; (iv) SSL embeddings are more disrupted by foreground/background swaps than supervised ones; (v) silhouette score in UMAP-reduced space correlates strongly with AMI and can serve as a label-free proxy.

  Strengths.
  - Genuinely broad and careful empirical sweep: 18 encoders × 6 clusterers × 26 datasets = 2,808 results, organized along a sensible ID→Far-OOD axis. Scale is a real contribution.
  - Methodological rigor in the parameter search: staggered line-search on held-in data only, honest treatment of HDBSCAN's noise-class disadvantage, statistical testing (Wilcoxon, Bonferroni-corrected t-tests,
  error bars), and a clean separation of train vs. transfer data.
  - Strong reproducibility detail (Appendices D–G: exact sklearn params, versions, compute budget, memory tables).
  - The ImageNet-9 background analysis (Table 2) and the multi-label granularity analyses (iNaturalist/BIOSCAN taxonomy, ImageNet-R class-vs-artform, BreakHis) yield interesting interpretability results beyond raw
  benchmarking.
  - The silhouette-as-proxy finding is practically useful and contradicts prior work (Xu et al., 2022) with evidence.

  Weaknesses.
  - The headline ID-vs-OOD crossover relies on differences the authors themselves repeatedly note are not statistically significant (Figure 1: only MoCo-v3/DINO on domain-shift reach p<0.05; Far-OOD gains "not
  significant"). This undercuts the central narrative.
  - No modern SSL clustering baselines (SCAN, DeepCluster, TEMI, or even DINOv2) are compared, even as reference points; the paper benchmarks classical clusterers on 2020–2022-era encoders.
  - Limited novelty in method — the contribution is empirical characterization, not a new technique. That is acceptable for TMLR but should be framed honestly (see below).
  - Some claims rest on small per-group dataset counts (3≤N≤8), so group-level means have wide error bars and weak power.

**Audience:**

Yes

**Audience Explanation:**

Researchers in self-supervised learning, representation evaluation, category discovery, and OOD generalization would find value in: (a) the systematic ID→OOD characterization, (b) the practical recipe (UMAP-50d + AC/Ward), and (c) the label-free silhouette proxy. The interpretability analyses (background sensitivity, taxonomy granularity, medical/BreakHis) extend appeal to applied subfields.

**Broader Impact Concerns:**

I suggest the statement be expanded to (a) caution explicitly against deploying unsupervised face/demographic clustering, and (b) note that the medical clustering results are characterization-only and not clinically validated. No blocking ethical concern; the additions are straightforward.

**Claims And Evidence:**

Yes

**Claims Explanation:**

Most of the claims are supported, and here are some claims that I think over-stated:
- The central "supervised wins ID, SSL wins OOD" framing is presented as a clean trend, but the abstract and intro do not foreground that most of the OOD differences are not significant (this caveat is buried in Sec. 4.2). The bar charts (Figure 1) make the trend look stronger than the statistics warrant. This is the main reason claims feel under-supported.
  - "DINO produces the best SSL encoder (ViT-B) but the worst (ResNet-50)" is asserted with a mechanistic story (attention focusing on subject) that is plausible but only loosely evidenced.

**Requested Changes:**

1. Reframe the central claim around statistical significance. In the abstract and contributions list, explicitly state which ID/OOD differences are significant and which are not. Currently the abstract asserts the crossover as fact while Sec.4.2 admits most differences are not significant — these must be reconciled.
  2. Add at least one modern reference point. Include DINOv2 and/or a dedicated deep-clustering method (e.g., SCAN/TEMI) as a comparator, or — if out of scope — add an explicit paragraph in Sec.2 or Limitations justifying why no learned-clustering or post-2022 encoder baseline is included.
  3. Figure 1/3: report effect sizes and significance markers directly on the bars (e.g., asterisks), so readers do not over-read non-significant trends.
  4. Clarify the contribution claim "first in-depth investigation of clustering of SSL encoders outside their training domain" (p.2) against Lu et al. (2023) and Cole et al. (2022), which are closely related; sharpen what is genuinely new.
  5. The fine-tuning result that Far-OOD performance declines post-FT (Sec.4.4) is interesting and deserves more prominence/discussion than it currently gets — it is arguably a stronger finding than the headline.

---

> ### Author Response · Authors · 2026-06-20
> **Response**
>
> *Reframe the central claim around statistical significance*
>
> We have updated our abstract to better reflect the nature of our results and not overstate our findings. As described in the overall response, we have updated the statistical tests used for our overall claims to pool data from all SSL models (instead of only performing tests for individual models), which increases the statistical power of our comparisons by a factor of 5 or 6 (depending on the test) and better aligns the tests with the claims made about the overall trends.
>
> *Comparison with DINOv2/SCAN/TEMI*
>
> In Section 2 we have added an explicit delimitation of which SSL methods are used, where we argue that a comparison with e.g. DINOv2 is infeasible due to the pretraining dataset domain(s) overlapping with our evaluation datasets. We have also added further reasoning as to why deep clustering methods such as SCAN/TEMI are not included, as we explicitly focus on classical lightweight clustering methods which can be deployed on arbitrary pretrained networks, leaving a similar study into deep clustering methods for future works.
>
> *Significance of results in Figure 1/3*
>
> As per your suggestion we have added stars to the error bars in Figure 1/3 to indicate significance, making it much easier to tell which individual model results are significant.
>
> *Comparison to Cole et al. and Lu et al.*
>
> We have expanded on the connection to Cole et al. and Lu et al., especially how findings of Cole et al. are corroborated when changing from the classification setting to zero-shot clustering and across more SSL methods.
>
> *Emphasis on results showing Far-OOD declines post-FT*
>
> As per your recommendation, we have expanded on our description of this effect in the abstract and added it as a bullet to the contributions section in the introduction. Additionally, we added explicit statistical tests on the effect of FT on SSL clustering performances (rather than only testing SSL+FT against the supervised baseline). Through this analysis we found the AMI increase from SSL to SSL+FT to be significant on the in-domain, domain-shifted, near-OOD and datasets, in addition to the significant decrease on far-OOD. This is discussed in Section 4.4 and illustrated in a newly added Appendix (Appendix K in the revised paper) which demonstrates the effect sizes graphically.

---

### Review · Reviewer_eE7G · 2026-06-09

**Summary Of Contributions:**

This paper studies zero-shot clustering, where the authors apply self-supervised encoders pretrained only on ImageNet-1k to unseen datasets using conventional clustering algorithms.

Strengths:
- The paper provides a systematic empirical study of clustering with frozen SSL encoders. The empirical results in Figures 1 and 3 suggest that clustering based on SSL encoder embeddings can achieve performance comparable to clustering based on supervised encoder embeddings.
- The correlation between AMI and silhouette score is well presented, and offers a useful proxy for evaluating the clustering performance of SSL encoders when ground-truth labels are unavailable.

Weaknesses:
- The comparison between SSL encoders and supervised encoders is not fully convincing. In Figure 1, SSL encoders perform substantially worse than supervised encoders on in-domain, domain-shifted, and near-ood datasets, while they are only slightly better on far-ood datasets. In Figure 3, the performance of SSL and supervised encoders appears very similar. Also, the error bars in Figures 1 and 3 are computed over only $3 \le N \le 8$ datasets, which does not provide strong evidence for the claimed comparison.
- The main baseline is a supervised encoder trained on Imagenet-1k. While this is a reasonable baseline for comparing frozen pretrained models, the practical implication of this comparison is unclear. The paper does not assess the absolute quality of SSL-based clustering, for example by comparing against task-specific supervised models trained on each dataset, or by providing an oracle/upper-bound reference.

**Audience:**

Yes

**Audience Explanation:**

The studied problem is within the scope of TMLR.

**Claims And Evidence:**

No

**Claims Explanation:**

As mentioned in the weaknesses, the error bars in Figures 1 and 3 are computed over only $3 \le N \le 8$ datasets, which does not provide strong evidence for the claimed comparison.

**Requested Changes:**

- It would be helpful if the authors could provide comparisons with supervised models pretrained on different datasets.
- Please consider using additional baselines or metrics to assess the absolute utility of clustering SSL encoder embeddings.

---

> ### Author Response · Authors · 2026-06-20
> **Response**
>
> *Limited statistical power in figures 1 and 3*
>
> As suggested by z7Md, we have added stars next to the error bars to indicate which comparisons are statistically significant within the graphs. As detailed in the overall response, we have updated the statistical tests used for our overall claims to pool data from all SSL models (instead of only performing tests for individual models), which increases the statistical power of our comparisons by a factor of 5 or 6 (depending on the test).
>
> *Additional comparisons against models supervised on the test domain*
>
> While we recognize that providing upper-bounds based on per-dataset supervised performances could be interesting, we do not believe that the additional insights would balance out the necessary resources required to ensure a fair baseline is established per-dataset. First, we would need to conduct a hyperparameter search on how to train each supervised model backbone on its own training data (which differ greatly in data complexity and size). Second, in order to compare the performance of the domain-supervised model against the zero-shot clustering models we would need to evaluate it with clustering, necessitating a sweep on the clustering hyperparameters. Furthermore, some datasets do not have a large enough training dataset to establish a performant supervised model if trained from scratch (e.g. Oxford Flowers), while others suffer from incomplete labels (e.g. BIOSCAN-1M), creating non-trivial differences which need to be overcome: fine-tune a pretrained model, and if so, which? and how to chose which granularity(ies) of labels to train on? To address this, we have updated our Limitations section (Appendix B) to acknowledge that we report relative rather than absolute utility: the absence of per-dataset reference points for intrinsic task difficulty means absolute AMI values cannot be calibrated, which we leave to future work. We considered citing published per-dataset results as loose reference points, but several of our evaluation datasets are niche and lack a canonical benchmarked result, and differences in metric and protocol across sources could make such a column more misleading than informative.

---

### Review · Reviewer_Wvca · 2026-06-10

**Summary Of Contributions:**

The paper’s main contribution is a thorough empirical investigation of whether pretrained self-supervised visual representations can be clustered meaningfully on datasets they were never trained on. Instead of treating SSL features only as inputs for downstream classifiers or linear probes, the authors ask a different question: do the embeddings themselves already organize images into semantically useful groups when paired with standard clustering algorithms ? To answer that, they run a large benchmark across many datasets spanning in-domain, domain-shifted, near out-of-domain, fine-grained, and far out-of-domain settings, and compare multiple pretraining paradigms, including supervised, contrastive, distillation-based, and masked-image-modeling approaches. They also vary backbone architecture, using both ResNet-50 and ViT-B, which lets them compare how representation structure changes with model family. A second important contribution is methodological: they show that clustering quality depends strongly on the embedding preprocessing step, and that manifold-based reductions such as UMAP often work better than PCA for trained SSL features, while the opposite is often true for raw images and untrained networks. A third contribution is the evidence that clustering can serve as a complementary evaluation lens for SSL, since the paper finds strong agreement between clustering quality and downstream semantic alignment, and shows that silhouette score in UMAP-reduced space can act as a practical label-free proxy when ground truth is unavailable. The paper also adds interpretability value by analyzing what SSL encoders attend to in settings like ImageNet-9, showing that these models can be more sensitive to background-foreground interactions than supervised models.

In terms of strengths, the work is broad, carefully controlled, and unusually detailed for an empirical paper of this type. The experimental matrix is large enough that the observed patterns feel robust rather than anecdotal, and the appendix provides substantial support for the chosen parameters and design decisions. The paper is also useful because it does not just report which encoder “wins,” but explains why certain combinations of encoder, dimensionality reduction, and clusterer behave better on different kinds of datasets. One weakness is that the conclusions are still bounded by a specific experimental scope: ImageNet-1k-pretrained vision models, classical clustering methods, and the selected datasets. Another limitation is that some of the strongest claims are empirical heuristics rather than general theory, so they are persuasive within the tested setup but not necessarily universal

**Additional Comments:**

Overall, this is a solid and well-executed empirical paper with clear utility for the community. The strongest part is the breadth of the evaluation; the weakest part is that the conclusions are necessarily tied to a fairly specific vision pretraining and clustering pipeline

**Audience:**

Yes

**Audience Explanation:**

Yes. The findings should interest researchers working on representation learning, self-supervised learning, clustering, and evaluation of pretrained models, because the paper offers a useful zero-shot perspective on how embeddings behave on unseen datasets and suggests a practical proxy for cluster quality when labels are unavailable. It is also the kind of study that TMLR’s criteria explicitly consider interesting: it surfaces generalizable lessons rather than just benchmark chasing.

**Broader Impact Concerns:**

.

**Claims And Evidence:**

Yes

**Claims Explanation:**

Yes. The paper backs its main claims with a large empirical study spanning many datasets, two backbone architectures, multiple SSL paradigms, and several clustering methods, and it uses both label-based and label-free metrics to support the conclusions. The evidence is generally convincing because the trends are consistent across many settings, and the authors also include detailed parameter searches and ablations in the appendix to justify key design choices. That said, some claims are strongest within the exact experimental scope tested here, so the paper would be even more convincing if it toned down any broader generalization beyond ImageNet-1k vision encoders.

**Requested Changes:**

The main change I would ask for is a modest tightening of claims so they stay clearly within the experimental setting: ImageNet-1k-pretrained vision encoders, classical clustering methods, and the tested datasets. I would also ask the authors to make the practical guidance more explicit in the main paper, especially the recommended UMAP + agglomerative clustering setup and when silhouette score is a reliable proxy. If space allows, a slightly clearer discussion of when the conclusions may fail, especially for other modalities or pretraining regimes, would strengthen the paper.

---

> ### Author Response · Authors · 2026-06-20
> **Response**
>
> *Tightening claims*
>
> We have updated the manuscript (abstract, introduction, background, conclusions) so our claims are better contextualized. This makes it clearer that this study is only on ImageNet-1K pretrained vision models and classical clustering methods, thus we can not make broader conclusions.
>
> *Practical Guidance*
>
> We have added a new section (new Section 5) which explicitly details our practical recommendations for zero-shot clustering (Agglomerative Clustering together with UMAP reduction to 50-dimensions) with caveats such as how to adapt this advice for memory-constrained settings, and details when the silhouette score can be used as a proxy for clustering performance on unlabelled data.

---

### Author Response · Authors · 2026-06-20
**General response from Authors**

We thank all reviewers for their thoughtful reviews and constructive feedback. We appreciate that all reviewers found the paper of high interest for the SSL and clustering communities, and recognized the experimental design and scale of our analysis. We have updated the manuscript to incorporate the reviewer feedback, with changes highlighted in magenta.

Reviewers eE7G and z7Md noted that our statistical tests had limited statistical power, and our claimed results were in places simplified into an overall picture which was not fully consistent with the statistical tests we performed. On reflection, as our claims were primarily made about SSL models overall, our tests should reflect these statements rather than comparisons for individual models. We pooled the results across SSL models, increasing the number of datapoints used in the tests by a factor of 5 or 6 (depending on test). This change means our overall statements (which were previously a bit "hand-wavey" summaries) are now robustly supported by direct evaluations. The new methodology for our statistical claims is shown in the revised manuscript.